# Stress-induced red nucleus attenuation induces anxiety-like behavior and lymph node CCL5 secretion

Dong-Dong Shi [1,2], Ying-Dan Zhang[1], Sen Zhang[1], Bing-Bing Liao[1], Min-Yi Chu[1], Shanshan Su[1], Kaiming Zhuo[1], Hao Hu[1], Chen Zhang [1] & Zhen Wang [1,2,3] ✉

Previous studies have speculated that brain activity directly controls immune responses in lymphoid organs. However, the upstream brain regions that control lymphoid organs and how they interface with lymphoid organs to produce stress-induced anxiety-like behavior remain elusive. Using stressed human participants and rat models, we show that CCL5 levels are increased in stressed individuals compared to controls. Stress-inducible CCL5 is mainly produced from cervical lymph nodes (CLN). Retrograde tracing from CLN identifies glutamatergic neurons in the red nucleus (RN), the activities of which are tightly correlated with CCL5 levels and anxiety-like behavior in male rats. Ablation or chemogenetic inhibition of RN glutamatergic neurons increases anxiety levels and CCL5 expression in the serum and CLNs, whereas pharmacogenetic activation of these neurons reduces anxiety levels and CCL5 synthesis after restraint stress exposure. Chemogenetic inhibition of the projection from primary motor cortex to RN elicits anxiety-like behavior and CCL5 synthesis. This brain-lymph node axis provides insights into lymph node tissue as a stress-responsive endocrine organ.

Studies in humans and animals suggest that exposure to stressors can trigger or exacerbate anxiety disorders[1,2]. One biological process that has been increasingly investigated over the last decade is the inflammatory response[3], since it has a major role in the pathophysiology of mental illnesses, such as depression and anxiety[4,5]. Immune signaling contributes to the regulation of neurobiological processes that modulate anxiety- and depression-associated behaviors in the face of stressor exposure[6]. Exposure to traumatic and stressful events results in hypothalamic–pituitary–adrenal (HPA) axis reactivity, the activation of the immune system, and the release of proinflammatory cytokines[7,8]. Although extensive studies have been performed to characterize the role of cytokine/chemokine dysfunction in the pathophysiology and maintenance of stress-related mental health disorders, our understanding of the role of inflammation in the etiology and maintenance of these disorders remains limited. To understand the role of inflammation in stress-related reactions among these stressed people, we measured the levels of multiple immune mediators, cytokines, and chemokines in peripheral blood in the acute phase of stress.

It has been hypothesized that brain activity may directly control adaptive immune responses in lymphoid organs[9]. However, there is little evidence showing that stress-induced brain activity dysfunction can lead to immune responses in lymphoid organs. Previous research showed that splenic denervation specifically impairs plasma cell formation during T cell-dependent immune response in mice[10]. Neurons in the central nucleus of the amygdala (CeA) and the paraventricular nucleus (PVN) that express corticotropin-releasing hormone (CRH) are connected to the splenic nerve[10]. This study suggested that the CeA and PVN control spleen activity in mice; however, splenectomy surgery did not affect behavioral performance in either wild-type mice or stressed mice. In contrast, even though human studies and cross-neuronal viral

[1]Shanghai Mental Health Center, Shanghai Jiao Tong University School of Medicine, Shanghai, China. [2]Shanghai Key Laboratory of Psychotic Disorders, Shanghai Mental Health Center, Shanghai Jiao Tong University School of Medicine, Shanghai, China. [3]Institute of Psychological and Behavioral Science, Shanghai Jiao Tong University, Shanghai, China. ✉e-mail: wangzhen@smhc.org.cn

tracking assays in animal models have hinted that brain activities and inflammatory signaling contributed to stress-induced anxiety[11,12], to the best of our knowledge, the exact location and identity of neurons involved in stress and inflammation has not been causally proven.

To answer this question, adult rats were applied in our study. And a series of neuroscience technology, including electrophysiological techniques, fiber optic recording technology, optogenetics and chemical genetics techniques, were used to identify a group of glutamatergic neurons in RNs that play a vital role in stress-induced anxiety and inflammation by receiving projections from the primary motor cortex (M1). The results of fMRI further confirmed that the functional connectivity of the RN was significantly lower in stress-induced animals than in controls. The discovery provides insights into brain activity that controls anxiety and inflammation, and provides a potential brain target for treating stress-induced anxiety in humans.

## Results

### Levels of CCL5 were increased in individuals exposed to a stressful environment in the first month post stress

To measure the levels of immune indicators of stress in stressed people, we recruited psychiatrists who worked on the front line during the COVID-19 pandemic in Wuhan between February 21 and March 31, 2020 (Fig. 1a). In the study, 47 individuals exposed to a stressful environment (mean age $40.79 \pm 6.144$ years) and 41 controls (mean age $41.05 \pm 6.499$ years) completed the survey. Sex, age, marriage status and education years were not different between these two groups (Table 1). To evaluate the mental health of individuals exposed to a stressful environment and matched controls, we used the Patient Health Questionnaire-9 (PHQ-9) for regular depression screenings[13], the Generalized Anxiety Disorder (GAD-7) to determine generalized anxiety levels[14], the PTSD Checklist for DSM-5 (PCL-5) to determine stress levels[15], and the Self-Regulation Questionnaire-20 (SRQ-20) for identifying mental disorder symptoms[16]. Acute phase refers to within one month after stress according to DSM-5 (The diagnostic and statistical manual of mental disorders, DSM). In the acute phase, the median scores on the PHQ-9, GAD-7, PCL-5, and SRQ-20 for all participants were 2.00, 1.00, 2.00, and 1.00, respectively. There were no differences in PHQ-9 and GAD-7 scores between the stressed group and the control group (Table S1 and Fig. 1b, c). Individuals exposed to a stressful environment had higher scores on the SRQ-20 ($p = 0.001$) and PCL-5 ($p = 0.019$) (Fig. 1d, e). After 6 months, the PCL-5 scores in individuals exposed to a stressful environment reverted to the level in control individuals ($p = 0.164$) (Fig. 1d). The SRQ-20 scores were still higher in the stressed group than in the control group ($p = 0.040$) (Fig. 1e), suggesting that the impact of stress may still be present even after half a year. The scores on the SRQ-20 and PCL-5 reverted to control levels 1 year later ($p \geq 0.976$) (Fig. 1d, e), which suggested that the mental health of stressed individuals returned to normal levels.

To explore the relationship between inflammatory cytokines/chemokines and the stressed state, we measured the levels of 10 immune mediators, cytokines, and chemokines in peripheral blood. Our results showed that the levels of MCP-1, IL-1β, IL-2, IL-4, IL-6, TGF-α, TNF-α, and BDNF in stressed individuals and controls were not different (Fig. 1f, h–n). CCL5, also known as RANTES, was elevated significantly in stressed people in the first month after returning from a stressful environment (Fig. 1g), suggesting that stress can increase the level of CCL5, which can last for at least one month. Surprisingly, the level of cortisol, a naturally occurring steroid hormone that plays a key role in the body's stress response, was lower in stressed people than in controls (Fig. 1o).

Six months and one year later, the concentration of CCL5 was decreased to the level of controls (Fig. 1p). The concentration of cortisol slightly increased to normal levels following 6 months and 12 months of recovery (Fig. 1q). These results suggested that the physiological indicators of stressed individuals recovered 1 year later.

We also analyzed the correlation of these immune mediators and mental health state in the acute phase. Our results showed that several inflammatory cytokines, such as IL2 and TGFα, were negatively correlated with the stress level (Fig. S1a). Both the CCL5 levels and mental states of matched controls were not different from baseline after 1 year (Fig. S1b–h). To further understand the relationship between CCL5 and mental state, we analyzed the correlation between CCL5 levels and questionnaire scores. After taking the follow-up data into account, CCL5 had a significant positive correlation with PCL-5, PHQ-5, and SRQ-20 scores (Fig. S1i–k). Together, these results reveal that stress led to a dramatic increase in CCL5, and we suspect that CCL5 may be a marker reflecting stress levels.

### Acute restraint stress led to anxiety-like behavior and a sharp increase in CCL5 in rats

To verify the fluctuation in CCL5 levels after stress, which was observed in clinically stressed individuals, both male and female rats were exposed to restraint stress for three days (Fig. 2a and Fig. S1). In our study, restraint stress had no effect on the locomotion abilities of rats (Fig. 2b). The male rats spent less time in the central zone in the open field test and less time in the open arms in the elevated plus maze test (Fig. 2c–e). These results reveal that acute restraint stress can lead to anxiety-like behavior in male rats. However, restraint stress had no effect on anxiety-like behavior in female rats (Fig. S2b–e). The different performance of females and males suggested that males should be selected for subsequent mechanistic studies.

To determine the effect of circadian rhythms on CCL5 and corticosterone levels, control rats not exposed to restraint stress underwent continual blood sampling to establish the 24-h CCL5 and corticosterone profiles. For CCL5 levels, no circadian rhythms in control rats not exposed to restraint stress were observed, indicating that the CCL5 concentration was unaffected by circadian rhythms (Fig. S3a). Control rats not exposed to stress exhibited a marked rhythm of corticosterone levels over the 24-h sampling period (Fig. S3b). These results indicated that the CCL5 concentration was unaffected by circadian rhythms. This finding suggests that CCL5 is more readily available than corticosterone.

To investigate the CCL5 concentration after acute stress exposure, we collected the blood of rats at different times before and after restraint stress exposure. To help animals adapt to the blood collection process, we collected blood 3 days before stress exposure. Our results indicated that the blood collection process had little effect on the CCL5 level and corticosterone level in either male or female rats (Fig. S3c–j). After 3 days of acclimatization, rats were subjected to restraint stress. Blood was collected after the first day of restraint stress exposure, and the levels of CCL5 and corticosterone were analyzed. Although female rats showed no anxiety-like behavior, we observed a sharp increase in CCL5 and corticosterone in both female and male rats on the first day of being subjected to restraint stress (Fig. S3g, i). The concentrations of CCL5 and corticosterone were slightly decreased on the third day. On the eleventh day (one week after restraint stress exposure), the CCL5 concentration was maintained at the same level as that on Day 3. Neither CCL5 nor corticosterone levels oscillated in rats not exposed to restraint stress (Fig. S3d–j). To assess whether the upregulation of CCL5 occurred in other stressed anxiety/depression models, we measured the CCL5 levels in rats exposed to early life stress (ELS) or chronic unpredictable mild stress (CUMS). In contrast to control rats, both ELS rats and CUMS rats exhibited reduced interest in exploring the central zones in the open field test (Fig. S4b, g) and spent less time in the open arms in the elevated plus maze test (Fig. S4c, h). Consistent with the results of acute restraint stress, rats exposed to ELS or CUMS exhibited significantly increased CCL5 levels in serum (Fig. S4e, j), suggesting that stress can lead to a sharp increase in CCL5 in rats. Although acute restraint stress exposure increased CCL5 levels in both female and male rats, it failed to induce

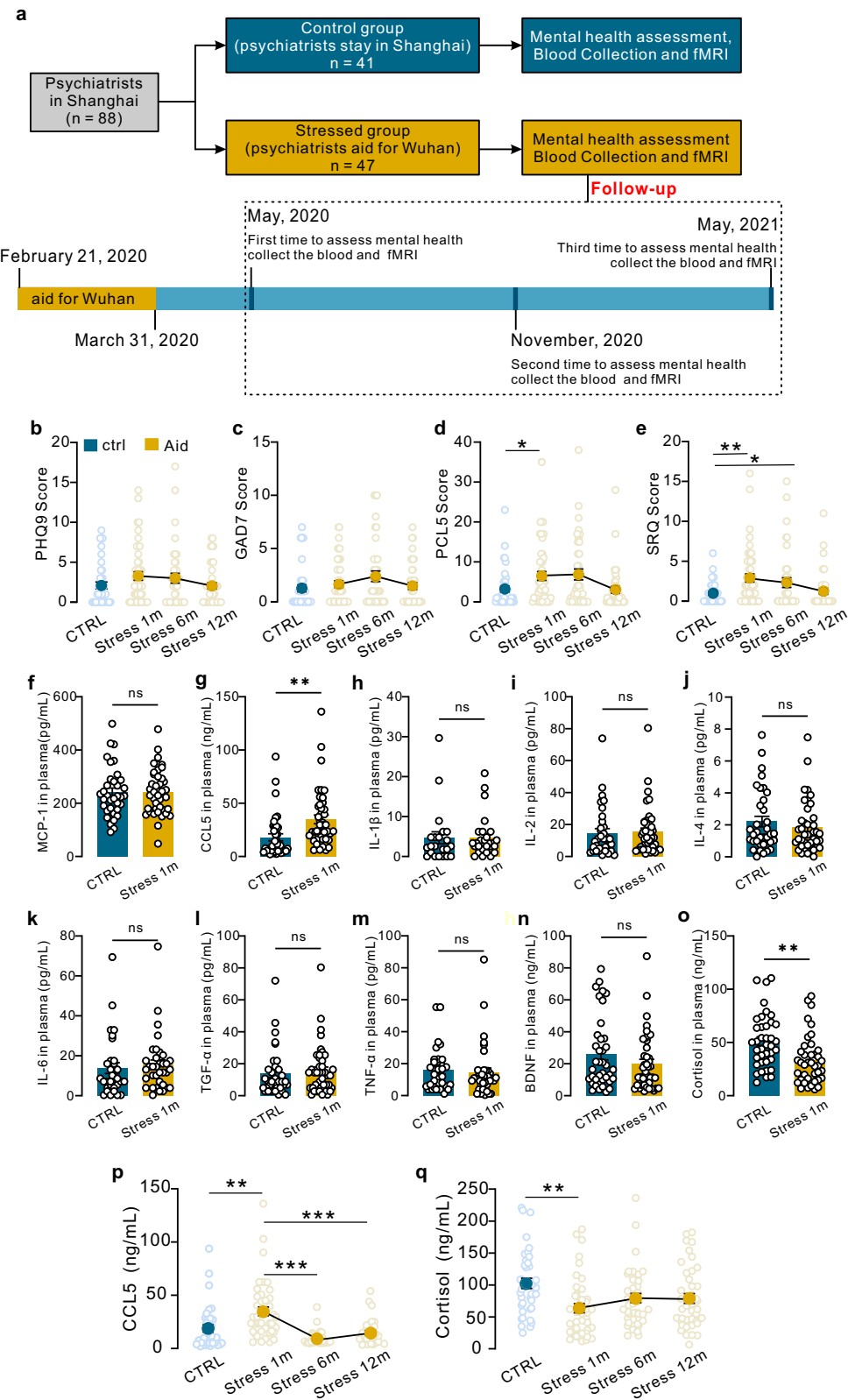

anxiety-like behaviors in female rats. To understand the inflammatory and neural mechanisms of acute stress-induced anxiety, the following experiments were mainly performed using male rats.

**Stress-inducible CCL5 was produced by lymph nodes**

CCL5 is produced by many cell types, including myocytes, endothelial cells and adipocytes[17]. To understand the origin of stress-inducible

CCL5 in serum, we screened multiple tissues for CCL5 induction using restraint stress models. We found that CCL5 was robustly induced in the cervical lymph nodes (CLNs) (Fig. 2f). To test whether CLNs were the main source of stress-inducible CCL5, we surgically excised the CLNs (Fig. 2g). After 3 days of restraint stress exposure, an analysis of the distance moved in the open field test showed that lymphadenectomy had no effect on locomotion activity (Fig. 2h). Rats in which

**Fig. 1 | Mental health outcomes and immune mediators of stressed individuals.** **a** Flowchart of the stress period and blood collection of stressed individuals. **b** PHQ9 scores. **c** GAD7 scores. **d** PCL-5 scores (one-way ANOVA, $F_{3,165} = 3.689$, *$P = 0.0190$). **e** SRQ-20 scores (one-way ANOVA, $F_{3,165} = 3.234$, **$P = 0.0013$, *$P = 0.0404$) of stressed individuals. ($n = 41$ for control, $n = 47$ for stress 1 m, $n = 40$ for stress 6 m, $n = 41$ for stress 12 m). **f**–**o** Plasma levels of MCP-1, CCL5 (two-sided unpaired $t$ test, **$P = 0.0022$, $t = 3.167$, df = 79; $n = 38$ for control, $n = 43$ for stress), IL-1β, IL-2, IL-4, IL-6, TGF-α, TNF-α, BDNF and cortisol (two-sided unpaired $t$ test, **$P = 0.0020$, $t = 3.273$, df = 79; $n = 38$ for control, $n = 43$ for stress) in stressed individuals 1 month after leaving the stressful environment. **p**, **q**, Plasma CCL5

($n = 38$ for control, $n = 43$ for stress 1 m, $n = 39$ for stress 6 m, $n = 33$ for stress 12 m) and cortisol levels ($n = 39$ for control, $n = 43$ for stress 1 m, $n = 41$ for stress 6 m, $n = 41$ for stress 12 m) in stressed individuals 6 months later and 1 year later (one-way ANOVA, $F_{3,165} = 10.800$, **$P = 0.0012$, ***$P < 0.0001$. ctrl control individuals without stressful experience, stress 1 m: stressed individuals 1 month after leaving the stressful environment, stress 6 m: stressed individuals 6 months after leaving the stressful environment, aid 12 m: stressed individuals 12 months after leaving the stressful environment. Data are presented as the mean ± SEM. *$P < 0.05$, **$P < 0.01$, ***$P < 0.001$. Source data are provided as a Source Data file.

the CLNs were removed spent more time in the central zone in the open field test than sham rats (Fig. 2i). There was a trend toward a higher percentage of time spent in the open arms in rats in which the CLNs were removed than in sham rats (Fig. 2j). Lymphadenectomy ablated the CCL5 response to restraint stress (Fig. 2k).

To examine the role of the adaptive immune system in stress-induced behavioral changes, we compared the frequencies and numbers of peripheral blood CD4+ and CD8+ lymphocytes. Rats exposed to acute restraint stress exhibited no changes in the frequencies or numbers of peripheral blood CD4+ and CD8+ lymphocytes compared to nontreated controls (Fig. S5a, b). Furthermore, the frequencies and numbers of CLNs CD4+ and CD8+ lymphocytes were also investigated. Rats exposed to restraint stress showed significantly increased frequencies and numbers of CD8+ lymphocytes compared to controls (Fig. 2l, m). It has long been known that memory cells respond more rapidly to antigens[18], and the level of mRNA for CCL5, which was virtually absent from naïve cells, was over 30-fold higher in both populations of memory phenotype T cells[18]. Compared to control rats, the frequencies and numbers of CLNs memory CD8+ T cells were significantly increased in stressed rats (Fig. 2n, o), suggesting that CD8+ T cells have a broad impact on physical stress-induced anxiety-like behavior. Besides, the CD8+ T cells in CLNs from RS rats exhibited a significant increase in CCL5 expression compared to that in control rats (Fig. S5c).

### Stress-induced translation activation promoted CCL5 protein synthesis in lymph nodes

Given the observed difference in RS-induced changes in cervical lymph nodes, the molecular mechanisms were assessed. To address

transcriptome-wide alterations in CLNs in stressed rats, we performed RNA sequencing (RNA-seq) analysis. Genes in the CLNs of stressed rats showed significant differences compared to those in the control group (Fig. 2p). A total of 4697 specifically differentially expressed genes (DEGs) were identified in the CLNs of RS group rats (Fig. 2q). Gene Ontology (GO) analysis revealed that a large number of these DEGs encoded proteins for the translation and regulation of immune system processes (Fig. 2r). Restraint stress led to an increased level of translation initiation factors and a decreased level of ribosome genes (Fig. 2s). Restraint stress also led to an increase in the mRNA concentrations of cytokines and chemokines, such as IFNα, IL2, IL6, and CCL2 (Fig. S6a). Additionally, the expression of eukaryotic translation initiation factor 4E (eIF4E), which aids in translation initiation by recruiting ribosomes to the 5'-cap structure, was upregulated in the CLNs of RS group rats (Fig. S6b). There were no changes in the expression of interferon regulatory factor 1 (IRF1) in the CLNs of stressed rats, suggesting that RS promotes translation in CLNs, which is consistent with the results of RNA-seq.

To confirm the effect of the translation process on RS-induced anxiety-like behavior, anisomycin, a protein synthesis inhibitor, was used to inhibit the translation process in rats (Fig. S6c). The inhibitory effects were confirmed by immunohistochemistry (Fig. S6d). After the administration of anisomycin, the CCL5 level in CLNs was decreased significantly in RS rats (Fig. S6e). Regarding behavioral results, anisomycin did not affect the locomotion ability of stressed rats (Fig. S6f). Anisomycin increased the time spent in the central zone in the open field test and the open arm ratio in the elevated plus maze in RS rats, suggesting that translation inhibition relieves anxiety levels (Fig. S6g–j). CCL5 levels in serum were also analyzed to detect the effect of translation inhibition. Restraint stress resulted in a sharp increase in CCL5 levels in rats injected with saline, while anisomycin attenuated this increase (Fig. S6k). In summary, stress-induced upregulation of translation triggers CCL5 synthesis in CLNs and the onset of anxiety, although the underlying mechanism remains to be further investigated.

### Identification of RN glutamatergic neuron clusters upstream of the CLNs

To identify brain regions involved in CLN control, we first injected mRFP-encoding pseudorabies virus 152 (PRV) directly into the CLNs of adult rats to retrogradely and transsynaptically label upstream neurons (Fig. 3a). Over 6 days, we observed clusters of mRFP+ neurons in specific regions, including RN and M1 (Fig. 3a and Figs. S7, S8). As we all know, the red nucleus (RN) is part of the midbrain tegmentum and is responsible for motor functions together with brain areas such as the motor cortex, cerebellum, spinal cord and posterior thalamus[19]. We hypothesize that RN neurons activities may also be transmitted as efferent output through lymph node neurotransmission to promote CCL5 synthesis. Two types of neurons were found in RNs: glutamatergic neurons and GABAergic neurons. To investigate which type of neuron plays an important role in regulating CLNs, we stained glutamatergic neurons and GABAergic neurons with a marker for PRV-mRFP (Fig. 3b). Approximately 89.72% of mRFP+ neurons in the RN were costained with vGlut1, suggesting that glutamatergic neurons may be the main neurons in the RN regulating CLNs (Fig. 3c).

**Table 1 | Demographic characteristics**

| Characteristic | Mean ± S.D or No. (%) | | | |
| --- | --- | --- | --- | --- |
| | | Working position | | |
| | | Stressed individuals | Controls | P value |
| Overall | 88 (100) | 47 (52.3) | 41 (47.6) | |
| Sex | | | | |
| Men | 64 (72.7) | 35 (39.8) | 29 (33.0) | 0.903 |
| Women | 24 (27.3) | 12 (13.6) | 12 (13.6) | |
| Age | | | | |
| Mean | 40.90 ± 6.328 | 40.79 ± 6.144 | 41.05 ± 6.499 | 0.345 |
| Median | 40 | 40 | 40 | |
| Marriage status | | | | |
| Unmarried | 9 (10.2) | 4 (4.5) | 5 (5.7) | 0.429 |
| Married | 77 (87.5) | 41 (46.6) | 36 (40.9) | |
| Divorced or widowed | 2 (2.3) | 2 (2.3) | 0 (0) | |
| Education years | | | | |
| Mean | 17.0 ± 1.5 | 20.8 ± 1.3 | 20.7 ± 1.6 | 0.622 |
| Median | 20 | 20 | 20 | |

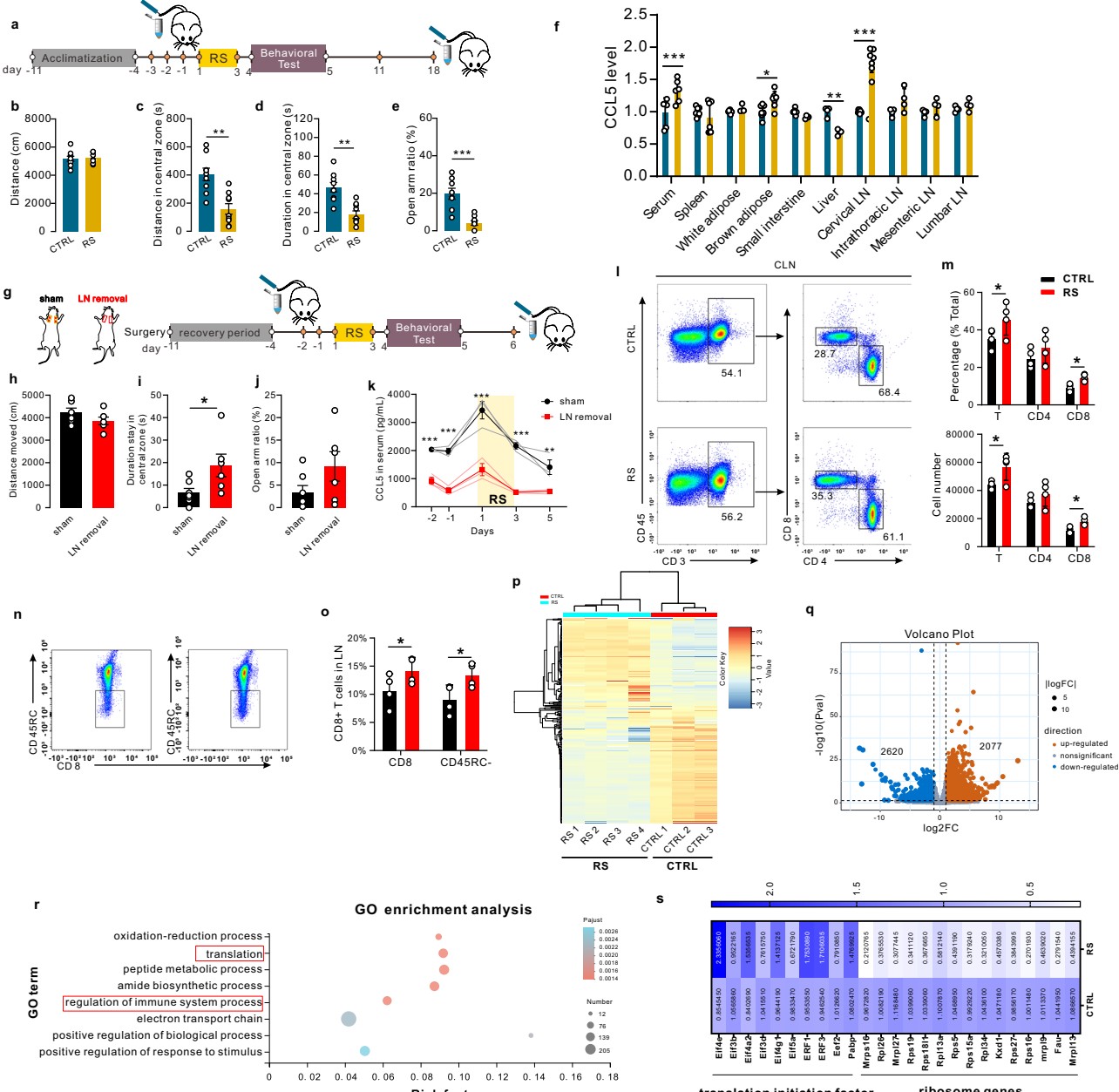

**Fig. 2 | Acute restraint stress led to anxiety-like behavior and a sharp increase in CCL5 in rats. a** Schematic timeline and behavioral paradigm in rats, RS restraint stress. **b** No difference between control rats and RS rats in distance moved in the open field test ($t$ test, $P = 0.5039$, $n = 9$). **c** The distance moved in the central zone was significantly lower in RS rats than in control rats in the open field test ($t$ test, $P = 0.0042$, $n = 9$). **d** The time spent in the central zone in the open field test was significantly lower in RS rats than in control rats ($t$ test, $P = 0.0046$, $n = 9$). **e** There was a lower open arm ratio in the elevated plus maze test in RS rats than in control rats ($t$ test, $P = 0.0002$, $n = 8$ for control and 9 for RS). **f** The fold change in CCL5 in tissues from stressed and control rats (two-way ANOVA, interaction $F_{9,94} = 17.83$, $P < 0.0001$, effect of tissues $F_{9,94} = 17.66$, $P < 0.0001$, effect of restraint stress $F_{1,94} = 24.86$, $P = 0.0969$, $n = 4$). **g** Schematic timeline and behavioral paradigm in rats in which the CLNs were removed. **h** There was no difference in distance moved in the open field test between sham rats and rats in which the CLNs were removed (two-tailed unpaired $t$ test, $t = 1.406$, df = 11, $P = 0.1902$, $n = 6$). **i** Rats in which the CLNs were removed spent significantly more time in the central zone in the open field test than sham rats (two-tailed unpaired $t$ test, t = 2.391, df = 11, $P = 0.0358$, $n = 7$ for control and 6 for LN removal). **j** There was a trend toward a higher open arm ratio in rats in which the CLNs were removed than in sham rats (two-tailed unpaired $t$ test, $t = 1.682$, df = 11, $P = 0.1208$, $n = 6$). **k** Serum CCL5 levels were also measured by ELISA at different times in these two groups. Restraint stress led to a sharp increase in CCL5 expression in both sham rats and rats in which the CLNs were removed. However, the CCL5 levels in sham

rats were significantly higher than those in rats in which the CLNs were removed (two-way ANOVA, interaction $F_{4,20} = 4.44$, $P = 0.0099$, effect of time $F_{4,20} = 21.02$, $P < 0.0001$, effect of lymphadenectomy $F_{1,20} = 191.9$, $P < 0.0001$; n = 4 per group). **l, m** Flow cytometry analysis of the frequency and absolute numbers of different T-cell populations in the CLNs of control and restraint stress-treated rats (two-way ANOVA, interaction $F_{2,18} = 0.4520$, $P = 0.6434$, effect of cell types $F_{2,18} = 47.76$, $P < 0.0001$, effect of stress $F_{1,18} = 9.021$, $P = 0.0076$, $n = 4$ per group; two-way ANOVA, interaction $F_{2,18} = 0.5738$, $P = 0.5734$, effect of cell types $F_{2,18} = 72.87$, $P < 0.0001$, effect of stress $F_{1,18} = 11.80$, $P = 0.0030$, $n = 4$ per group). **n, o** Flow cytometry analysis of CD8 memory cells in the CLNs of control and restraint stress-treated rats (two-way ANOVA, interaction $F_{1,12} = 0.1071$, $P = 0.7491$, effect of cell types $F_{1,12} = 0.9506$, $P = 0.3488$, effect of stress $F_{1,12} = 19.732$, $P = 0.0009$, $n = 4$ per group). CD8 + CD45RC-: memory CD8 T cells; CD8 + CD45RC +: Naïve CD8 T cells. **p** Heatmap showing the differentially expressed genes (DEGs) in the cervical lymph nodes of control and restraint stress-treated rats. The DEGs were identified with a fold change of RS/CTRL > 2.0 or <0.5. **q** Volcano map showing the number of DEGs in cervical lymph nodes in control and RS rats. RS led to the upregulation of 2077 genes and the downregulation of 2620 genes in CLNs. **r** Functional annotation of DEGs in the adult rat cervical lymph nodes. **s** Heatmap showing the expression of DEGs encoding translation initiation factors and ribosome genes. Data are presented as the mean ± SEM. The significance of differences in all two-group comparisons was determined by a two-tailed unpaired $t$ test. *$P < 0.05$, **$P < 0.01$, ***$P < 0.001$. Source data are provided as a Source Data file.

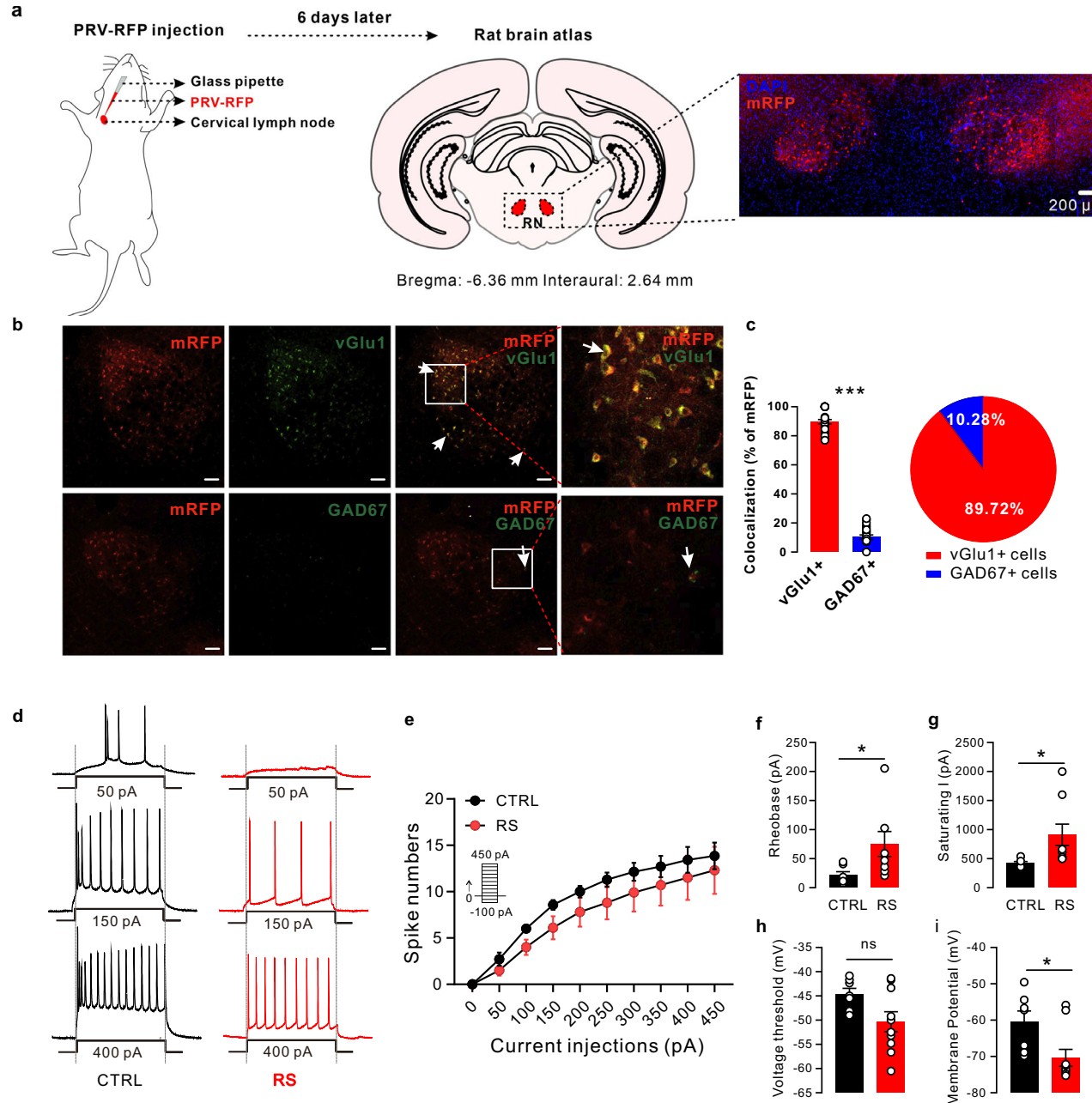

**Fig. 3 | Identification of a cluster of RN glutamatergic neurons upstream of CLNs. a** Schematic timeline and paradigm in rats PRV CLN injections. We first injected pseudorabies virus 152 (PRV), encoding mRFP, directly into the CLNs of adult rats to retrogradely and transsynaptically label upstream neurons. We observed a cluster of mRFP+ neurons bilaterally in specific regions, including the RN, $n = 3$. **b** Identification of a cluster of RN glutamatergic neurons upstream of CLNs. **c** 89.7% of cells projecting to the CLNs were vGlut1+ cells (two-tailed unpaired $t$ test, $P < 0.0001$, scale bar: 100 μm, $n = 3$). **d** Current steps of increasing amplitude applied during whole-cell recordings from RN glutamatergic cells in control (black) and RS (red) rats. Scale bar, 300 mV and 100 ms. **e** Depolarizing current steps evoked fewer action potentials in neurons recorded from RS animals than control animals (two-way ANOVA, interaction $F_{9,150} = 0.1149$, $P > 0.99$, effect of current steps $F_{9,150} = 16.63$, $P < 0.001$, effect of restraint stress $F_{1,150} = 6.761$, $P = 0.0100$; control: $n = 7$, RS: $n = 10$ cells for 3 rats). **f–i** Glutamatergic cells exhibited different intrinsic properties in control and RS rats. **f** Rheobase, $t = 2.235$, $P = 0.0436$, $n = 7$ for control and 8 for RS. **g** Saturating I, $t = 2.184$, $P = 0.0453$, $n = 7$ for control and 10 for RS. **h** Voltage threshold, $t = 2.127$, $P = 0.0504$, $n = 7$–10 per group. **i** Membrane potential, $t = 2.630$, $P = 0.0189$, $n = 7$ for control and 10 for RS. Data are presented as the mean ± SEM. *$P < 0.05$, **$P < 0.01$, ***$P < 0.001$. Source data are provided as a Source Data file.

To test the effect of restraint stress on RN glutamatergic neurons, we measured their frequency of action potentials in response to depolarizing current steps. The current was varied from −100 pA to 400 pA in 50 pA steps and the number of action potentials (induced spikes) elicited in 500 ms intervals was measured (Fig. 3d). The cells that exhibited spike frequency and broad action potentials and lacked spontaneous firing at resting membrane potential were identified as glutamatergic neurons. Compared to control rats, depolarizing currents evoked lower firing in stressed rats (Fig. 3e). Similarly, the cell needed more current to fire spikes at a given frequency (Fig. 3e). In addition, RN glutamatergic neurons in stressed rats displayed a larger rheobase, saturating currents, and a lower membrane potential and had a similar voltage threshold (Fig. 3f–i).

To examine the cell type-specificity and sub-second temporal resolution response patterns of RN glutamatergic neurons, we used fiber photometry to record changes in physiological calcium signaling

of these neurons in freely behaving rats[20] (Fig. 4). We stereotactically infused AAV-CaMKIIa-GCaMP6s into the RN regions in rats.

The calcium indicator GCaMP6s was expressed in the glutamatergic neurons of RNs (Fig. 4a, b). We unilaterally implanted an optical fiber above the RN (200 μm optical density, 0.37 numerical aperture) to record changes in RN neurons activity. During cage exploration, baseline recordings of RN neurons by GCaMP6s showed dynamic fluctuations of −3% ~ 13% (Fig. S9a, b). These dynamics were absent and were not affected by the movement and laser illumination of GFP-expressing neurons, indicating that the fluctuations were calcium-dependent GCaMP6s signaling. Fiber photometry revealed calcium fluxes in RN glutamatergic neurons in the open field test in control and stressed rats. Compared to control rats, restraint stress significantly decreased GCaMP6s fluorescence in RS rats. In the open field test, the $Ca^{2+}$ signals increased once the rats exhibited exploratory behavior. Increased $Ca^{2+}$ signals promoted rats to enter the central zone in the open field test (Fig. 4c, d). Compared to control rats, RS rats showed lower $Ca^{2+}$ signals during exploratory behavior in the open field test (Fig. 4e–i). In summary, the activation of RN glutamatergic neurons was a necessary condition for exploratory behavior in the central zone. However, animals did not necessarily go to the center when RN glutamatergic neurons were activated.

## Chemogenetic manipulation of RN glutamatergic neurons affects anxiety-like behavior and CCL5 secretion

To test the functional impact of RN glutamatergic neuron activity on anxiety-like behavior and CCL5 synthesis, we stereotactically injected recombinant AAV conditionally expressing the inhibitory receptor hM4D(Gi) (Fig. 5a). hM4D(Gi) can inhibit neuronal firing, when activated by the designer drug clozapine-N-oxide (CNO). Inhibition of RN glutamatergic neurons was validated by patch-clamp recordings of brain sections (Fig. 5b). CNO injection reduced the time spent in the central zone in the open field test and the open arm ratio in the elevated plus maze test in hM4D(Gi) rats but not in mCherry control rats (Fig. 5c, d). The administration of CNO increased the CCL5 level in the serum (Fig. 5e) and in the CLNs of hM4D(Gi) rats (Fig. S9c). Next, we sought to ablate the activity of RN glutamatergic neurons by using an AAV that conditionally expresses death-inducing active caspase 3 (AAV-EF1α-DIO-taCasp3-TEVP) and AAV-CaMKII-Cre into the RN of rats, which triggered cell apoptosis through Cre-induced expression of a genetically engineered caspase (taCasp3) in targeted neurons (Fig. 5f). The RN of control rats were injected with AAV-EF1α-DIO-taCasp3-TEVP alone. Immunostaining results showed a significant decrease in vGlut1+ cells at the injection site (Fig. 5f). Behaviorally, taCasp3-ablated rats showed higher anxiety levels, as shown by the reduced time spent in the central zone in the OFT and the diminished open arm ratio in the EPM test compared to controls (Fig. 5g, h). The CCL5 levels in serum and CLNs were significantly higher in taCasp3-ablated rats than in controls (Fig. 5i and Fig. S9d). Together, these loss-of-function experiments all support an essential role of the RN in anxiety-like behavior and CCL5 synthesis.

To determine whether enhanced RN glutamatergic neuronal activity would decrease anxiety levels and CCL5 production after restraint stress exposure, a recombinant hM3D(Gq) adeno-associated virus (AAV) was used; as a control, an AAV with fluorescent protein was applied (Fig. 5j). Action potentials were triggered after CNO activation, and verified by using patch-clamp technique to record the firing of single glutamatergic neurons in brain slices (Fig. 5k). CNO injection decreased the RS-induced anxiety level in hM3D(Gq) rats, as shown by the increased time spent in the central zone in the OFT and the increased open arm ratio in the EPM test compared to controls (Fig. 5l, m). Enhanced RN glutamatergic neuronal activity led to decreased CCL5 levels in the serum and CLNs after restraint stress exposure (Fig. 5n and Fig. S9e). Taken together, these data suggested that the activation of RN glutamatergic neurons could attenuate restraint

stress-induced anxiety-like behaviors and elevate CCL5 levels. Our study demonstrated the function of "brain-lymph nodes" axis in anxiety.

To further understand the role of CCL5 in RN inhibition-induced anxiety, we investigated whether systemic blockade of CCL5 influenced RN inhibition-induced anxiety using a specific neutralizing antibody. The open field test revealed that anti-CCL5 treatment had no effect on the locomotion ability of rats (Fig. S10a). The time spent in the central zone in the OFT and the open arm ratio in the EPM test were significantly increased in anti-CCL5 rats compared to IgG rats, suggesting that systemic neutralization of CCL5 reversed the anxiety induced by RN inhibition (Fig. S10b-d).

Previously, it was thought that circulating hormones and neurotransmitters were the bridge between the central nervous system and the immune system. Circulating hormones (e.g., cortisol) were produced by the neuroendocrine system[21] (e.g., cortisol) and neurotransmitters[22] (e.g., epinephrine and norepinephrine) were released by nerve endings near immune cells, such as epinephrine and norepinephrine in lymphoid organs. The finding that immune cells themselves can secrete neuromodulators and neuroendocrine factors leads to a deeper understanding of neuroimmune interactions[23]. It is well known that the sympathetic and parasympathetic nervous system innervated the body's most peripheral organs. Fluorescent histochemistry and radioenzymatic assay identified and confirmed that there were noradrenergic fibers in the cervical lymph nodes of rat[24]. Norepinephrine (NA) is an important modulator of immune function that is released by the sympathetic nervous system after stress[25]. Thus, we detected norepinephrine (NA) in CLNs. The results showed that NA was significantly increased both in stressed rats and in RN inhibited rats (Fig.S11). However, to understand the specific control of RN neurons to CLNs, further studies should be performed.

## Inhibition of the M1-RN circuit induces anxiety-like behavior

The primary motor cortex (M1) and RN are brain regions involved in limb motor control[26]. We observed mRFP+ neurons bilaterally in specific regions, including the M1 and RN (Fig. S7, S8). To determine whether RN glutamatergic neurons targeted by M1 neurons do indeed participate in anxiety-like behavior and CCL5 synthesis, we inject AAV1-hSyn-Cre, an anterograde transsynaptic virus, locally into M1 and AAV-CaMKII-DIO-GCaMP6s into the RN to allow Cre-dependent expression of GCaMP6s in RN neurons postsynaptic to M1 neurons (Fig. 6a). In these rats, we reliably observed persistent $Ca^{2+}$ transients as soon as exploratory behavior began through a recording optic fiber implanted in the RN. Compared to control rats, RS rats showed lower $Ca^{2+}$ signals during exploratory behavior in both the OFT and EPM test (Fig. 6b–f and Fig. S12).

To further confirm the contribution of the M1-RN pathway to the regulation of anxiety-like behavior, we tested the effects of chemogenetic inhibition of RN glutamatergic neurons receiving direct M1 inputs on rats. To specifically inhibit RN postsynaptic neurons, the monosynaptic anterograde transport virus AAV1-hSyn-Cre was bilaterally delivered to the M1, and AAV-CaMKII-DIO-hM4D(Gi)-mCherry was bilaterally delivered to the RN (Fig. 6g). RN postsynaptic neurons were inhibited by CNO (Fig. 6h). In hM4D(Gi)-mCherry rats, the chemogenetic inhibition of RN postsynaptic neurons induced by i.p. injection of CNO decreased the time spent in the central zone in the OFT and the open arm ratio in the EPM test (Fig. 6i, j). CCL5 levels in serum and CLNs were also increased through the chemogenetic inhibition of RN postsynaptic neurons (Fig. 6k and Fig. S13a). Furthermore, to further confirm that the M1-RN signaling pathway is associated with stress-anxiolytic-like behavior in rats, we retrograde injection AAV-CaMKII-betaglobin-cre-P2A-GFP into RN and inject AAV carrying a double-floxed inverted open reading frame (ORF) (DIO) of the hM4D (Gi) DREADD receptor and mCherry under the control of the CMV promoter to M1 (Fig. S13b). We performed ex vivo patch-clamp

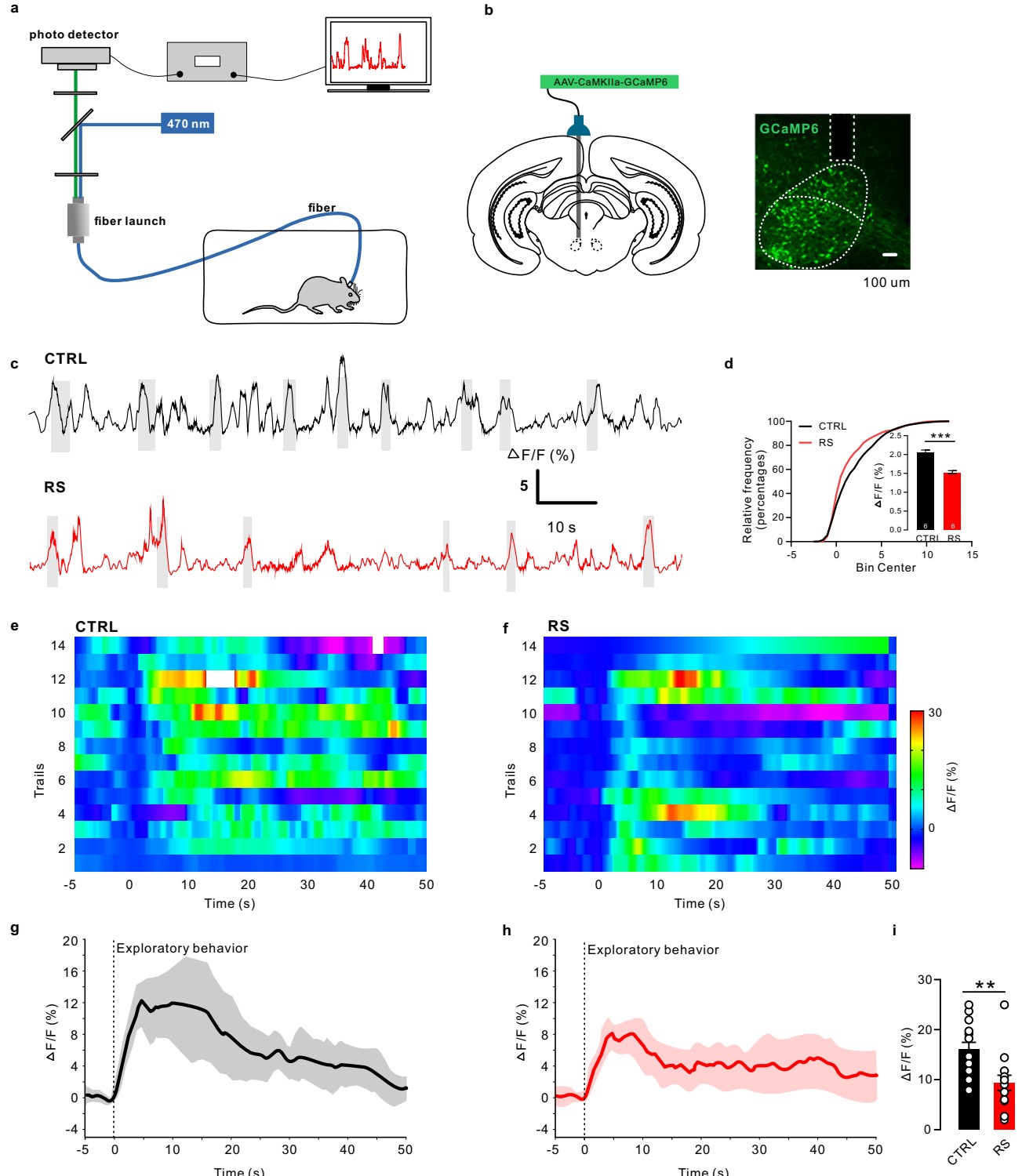

**Fig. 4 | Neuronal activity dysfunction in the RN glutamatergic neurons of control and RS rats. a** Schematic of the fiber photometry setup. Ca²⁺ transients were recorded from RN$^{GCaMP6}$ rats. **b** Schematic of fiber implantation in GCaMP6-expressing RN glutamatergic neurons. Scale bar, 100 μm, $n = 6$. **c** GCaMP6 signals from RN glutamatergic neurons aligned to the moment of exploratory behavior of the central zone in the open field. Control: black; RS: red. Gray areas indicate rats in the central zone. **d** Difference in average fluorescence between control rats and RS-treated rats. The average fluorescence in RS-treated rats was significantly lower than that in control rats (two-tailed unpaired $t$ test, t = 9.082, $P < 0.0001$, $n = 6$ rats

per group). **e**, **f** Heatmap of the RN glutamatergic neuronal response to exploratory behavior in control and RS-treated rats. **g**, **h** Averaged calcium signals in response to exploratory behavior in control and RS-treated rats. Black and red lines, mean calcium signal during exploratory behavior; gray area and red area, SEM. **i** Quantification of the change in calcium signals from exploratory behavior in control and RS-treated rats (two-tailed unpaired $t$ test, t = 3.180, $P = 0.0038$, $n = 6$ rats per group). Data are presented as the mean ± SEM. *$P < 0.05$, **$P < 0.01$, ***$P < 0.001$. Source data are provided as a Source Data file.

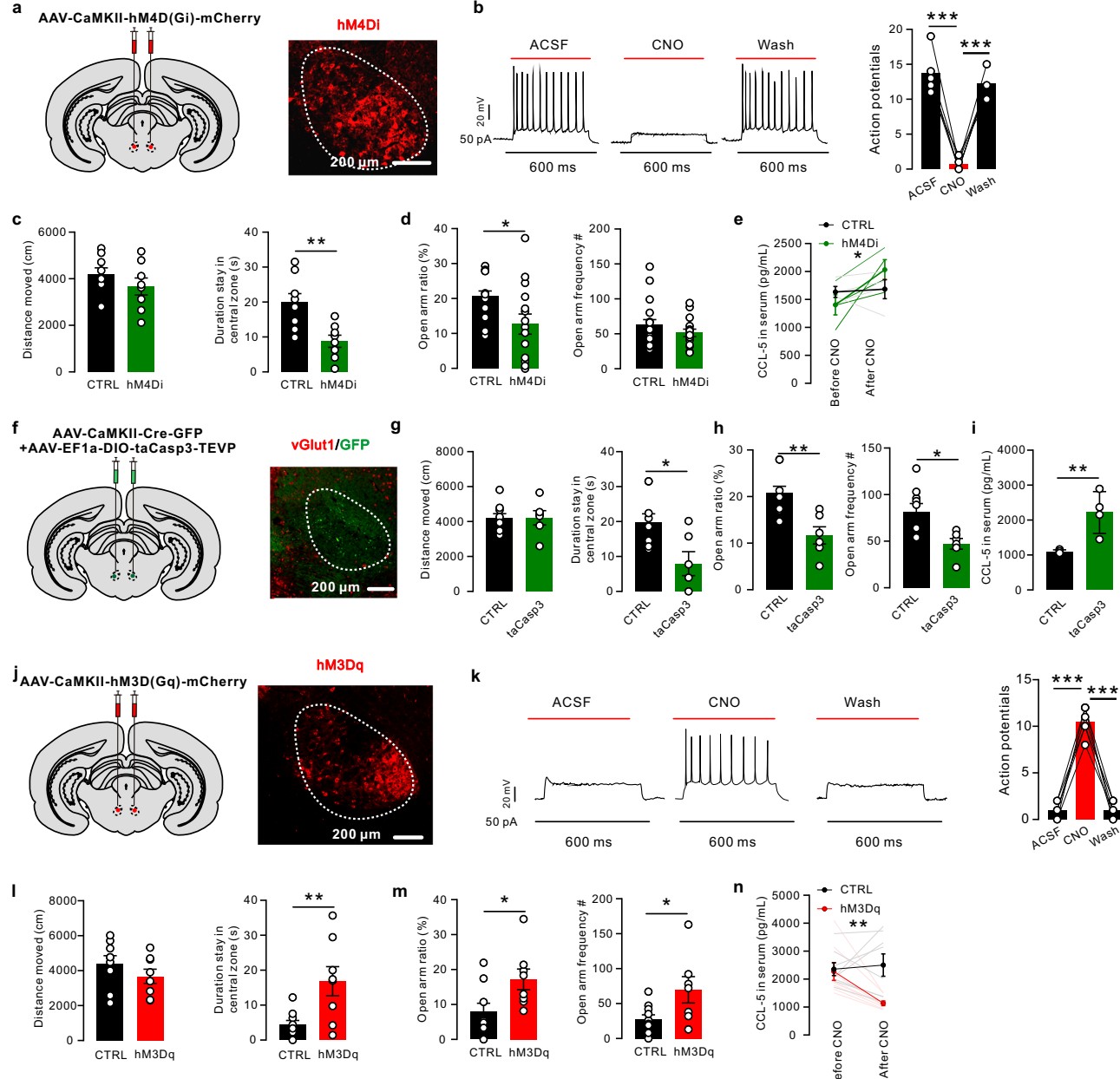

**Fig. 5 | Inhibiting or ablating RN glutamatergic neurons triggered anxiety-like behavior and an increase in CCL5. a** Schematic of AAV-CaMKII-hM4D(Gi)-mCherry injection bilaterally into the RN. Left: confocal image showing CaMKII-hM4D(Gi)-mCherry (red) expression in RN neurons. Right: an enlarged image from the left panel, $n = 3$. **b** Current-evoked action potentials in a representative hM4Di-infected neuron recorded before, during, and after CNO perfusion (10 mM) ($n = 6$ neurons; two-tailed paired $t$ test, t = 10.99, $P = 0.0001$; t = 14.26, $P < 0.0001$, respectively). **c** Inhibiting RN glutamatergic neurons had no effect on the locomotion abilities of adult rats (two-tailed unpaired $t$ test, t = 1.235, $P = 0.2357$, n = 17 rats for control and 15 rats for hM4Di) but decreased the time spent in the central zone in the OFT (two-tailed unpaired $t$ test, t = 3.686, $P = 0.0022$, n = 17 rats for control and 15 rats for hM4Di). **d** Inhibiting RN glutamatergic neurons led to a downward trend in the open arm ratio and frequency (two-tailed unpaired $t$ test, $t = 2.579$, $P = 0.0150$, $n = 15-17$ rats; $t = 1.101$, $P = 0.2797$, $n = 17$ rats for control and 15 rats for hM4Di). **e** CCL5 levels in serum were increased in hM4Di rats after CNO injection but not in control rats (two-tailed paired $t$ test, $t = 2.484$, $P = 0.0475$; $t = 0.2531$, $P = 0.8087$, respectively, $n = 4$). **f** Schematic of the bilateral coinjection of AAV-CaMKII-Cre-GFP and AAV-EF1α-DIO-taCasp3-TEVP into the RN. Confocal image showing reduced vGlu1⁺ signals (red) in the RN, $n = 3$. **g** Ablating RN glutamatergic neurons increased the anxiety level of adult rats, which showed less time spent in the central zone in the OFT than controls (two-tailed unpaired $t$ test, t = 2.783,

$P = 0.0155$, $n = 9$ rats for control and 6 rats for ablate group). **h** DIO-taCasp3 rats spent less time in the open arms than controls (two-tailed unpaired $t$ test, $t = 3.750$, $P = 0.0024$, $n = 9$ rats for control and 6 rats for ablate group). **i** Ablating RN glutamatergic neurons led to an increase in CCL5 levels in serum compared to controls (two-tailed unpaired $t$ test, $t = 3.757$, $P = 0.0094$, $n = 4$ rats). **j** Schematic of AAV-CaMKII-hM3D(Gq)-mCherry injection bilaterally into the RN. Confocal image showing CaMKII-hM3D(Gq)-mCherry (red) expression in RN neurons. **k** Current-evoked action potentials in a representative hM3Dq-infected neuron recorded before, during, and after CNO perfusion (10 mM) ($n = 6$ neurons; two-tailed paired $t$ test, $t = 15.34$, $P < 0.0001$; $t = 13.22$, $P < 0.0001$, respectively). **l** Activating RN glutamatergic neurons had no effect on the locomotion abilities of adult rats (two-tailed unpaired $t$ test, $t = 1.225$, $P = 0.2396$, $n = 8-9$ rats) but increased the time spent in the central zone in the OFT (two-tailed unpaired t test, $t = 3.022$, $P = 0.0086$, $n = 9$ rats for control and 8 rats for hM3Dq group). **m** Activating RN glutamatergic neurons led to an increase in the open arm ratio and frequency in the EPM test (two-tailed unpaired $t$ test, $t = 2.432$, $P = 0.0280$, $n = 8-9$ rats; $t = 2.431$, $P = 0.0281$, $n = 9$ rats for control and 8 rats for hM3Dq group). **n** CCL5 levels in serum were decreased in hM3Dq stress-treated rats after CNO injection but not in RS rats (two-tailed paired $t$ test, $t = 3.461$, $P = 0.0038$; $t = 0.3152$, $P = 0.7573$, respectively, $n = 4$). Data are presented as the mean ± SEM. *$P < 0.05$, **$P < 0.01$, ***$P < 0.001$. Source data are provided as a Source Data file.

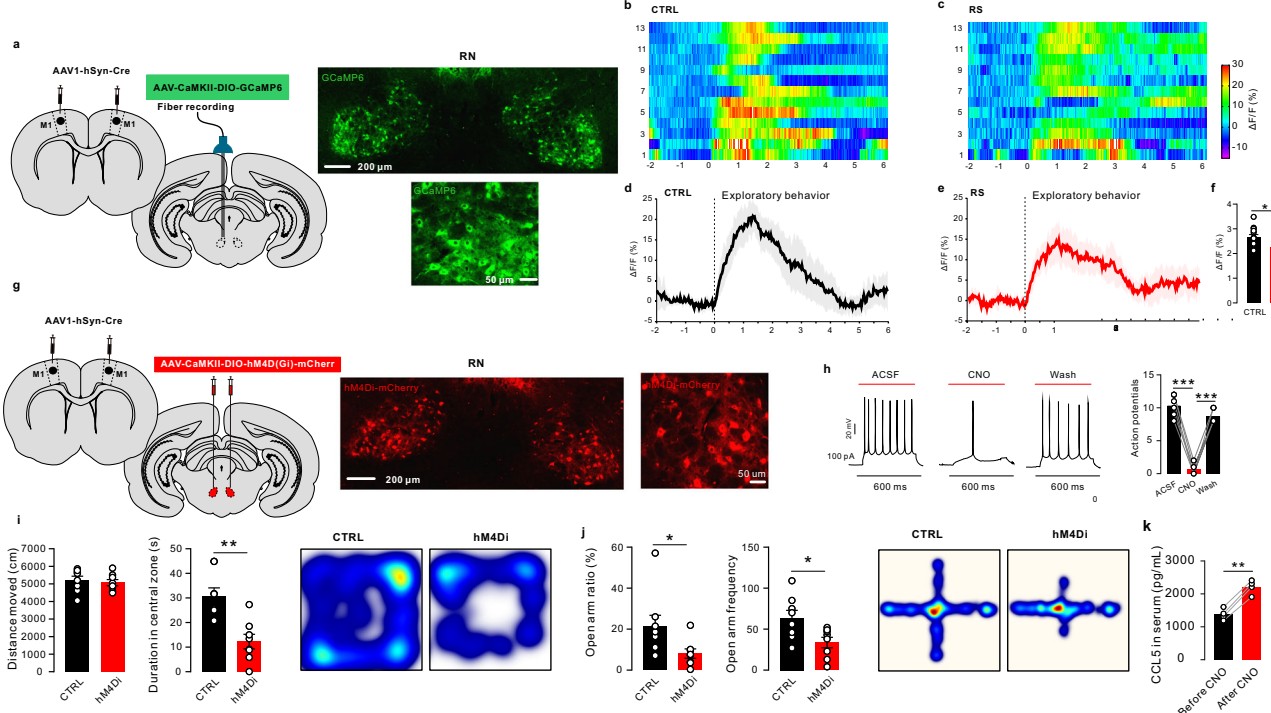

**Fig. 6 | Inhibiting the M1-RN pathway induced anxiety-like behavior. a** Left: injection of AAV1-hSyn-Cre into the M1 and AAV-CaMKII-DIO-GCaMP6 into the RN. Right: a representative confocal image of GCaMP6+ neurons in the RN. **b**, **c** Heatmap showing the Ca²⁺ signals evoked by exploratory behavior in the OFT in RN glutamatergic neurons from control and RS-treated rats. **d**, **e** Averaged responses of control and RS-treated rats (black and red line, mean calcium signal during exploratory behavior; gray area and red area, SEM). **f** Quantification of the change in calcium signals in response to exploratory behavior in control and RS-treated rats (two-tailed unpaired *t* test, *t* = 2.429, *P* = 0.0223, *n* = 6 rats per group). **g** Left: injection of AAV1-hSyn-Cre into the M1 and AAV-CaMKII-DIO-hM4Di-mCherry into the RN. Right: a representative confocal image of hM4Di+ neurons in the RN. **h** Current-evoked action potentials in a representative hM4Di-infected neuron recorded before, during, and after CNO perfusion (10 mM) (*n* = 6 neurons; two-

tailed paired *t* test, *t* = 17.33, *P* < 0.0001; *t* = 17.89, *P* < 0.0001, respectively). **i** Left: locomotion abilities were not affected by inhibiting M1-RN projections (two-tailed unpaired *t* test, *t* = 0.4938, *P* = 0.6291, *n* = 8 rats per group). Inhibiting M1-RN projections decreased the time spent in the central zone in DIO-hM4Di rats compared to mCherry control rats (two-tailed unpaired *t* test, *t* = 4.072, *P* = 0.0011, *n* = 8 rats per group). Right: representative tracks of control and hM4Di rats in the OFT. **j** Both the open arm ratio and frequency decreased significantly in hM4Di rats compared to mCherry control rats (two-tailed unpaired *t* test, *t* = 2.181, 2.649, *P* = 0.0468, 0.0191, *n* = 8 rats per group). Right: representative tracks of control and hM4Di rats in the EPM. **k** CCL5 levels in serum were significantly increased after CNO injection (two-tailed paired *t* test, *t* = 5.984, *P* = 0.0010, *n* = 4 rats). Data are presented as the mean ± SEM. *P* < 0.05, ***P* < 0.01, ****P* < 0.001. Source data are provided as a Source Data file.

electrophysiological experiments on M1 slices to confirm the efficacy of DREADD-mediated inhibition in M1 neurons retrogradely labeled from the RN. After CNO administration, a significant decrease in the number of evoked spikes were found in retrogradely labeled M1 neurons (Fig. S13c). Selective inhibition of M1-RN neuron projections decreased the time spent in the central zone in the OFT and the open arm ratio in the EPM test (Fig. S13d, e). CCL5 levels were increased through selective inhibition of M1-RN neuron projections (Fig. S13f–h). The results of presynaptic and postsynaptic manipulation experiments indicate that the inhibition of the M1-RN pathway leads to anxiety-like behavior and CCL5 synthesis in CLNs.

### Functional connectivity of the RN was weakened in stressed individuals in the first month post stress

To confirm the role of the RN in stress-induced anxiety, we further analyzed the fMRI findings of stressed individuals. Compared with controls, stressed individuals exhibited significantly decreased bilateral RN functional connectivity (FC). Specifically, we found decreased FC between the left RN and the left middle temporal gyrus and left middle frontal gyrus (Fig. S14a, b). We also found decreased FC between the right RN and the left middle temporal gyrus and right middle temporal gyrus. After multiple comparison corrections (*p* = 0.05/2), the decreased FC between the left RN and the mid-temporal gyrus and the right RN and the left mid-temporal gyrus and right mid-temporal gyrus were still significant. The results are

summarized in Fig. S14b. These results confirmed that stress-induced red nucleus dysfunction contributed to anxiety.

## Discussion

This study characterizes a mechanism by which stress impairs glutamatergic neuron functions in the RN to increase inflammation in cervical lymph nodes and anxiety levels. We focused on the chemokine CCL5 based on our clinical findings in stressed individuals, and we identified CLNs as the main immune organs that produce CCL5 in response to stress. We showed that the manipulation of RN glutamatergic neuron activities can alter anxious behaviors and inflammatory responses. These results indicated that the manipulation of peripheral immune organs could modulate stress-induced anxiety-like behavior, which provided a solid foundation and directions for anxiety disorder interventions.

Brain functions can be affected by stressful events, leading to short- and long-term behavioral changes. Our clinical study showed that both PCL-5 and SRQ20 scores indicated that psychiatrists who worked on the front line during the COVID-19 pandemic in Wuhan had a higher stress level than control individuals. The GAD7 scores had no difference between these groups. In humans, anxiety is a normal emotion, but excessive anxiety can develop into anxiety disorder, a chronic condition. The GAD7 is used to identify general anxiety disorder but cannot indicate the absence of anxiety in stressed individuals. Besides, we showed rats that experienced restraint stress spent

less time in the open arm in the EPM test and the central zone in the OFT, behaviors that indicate anxiety (i.e., an expression of their stress level). Anxiety disorders and other mental disorders are distinguished by their symptoms in humans[27]. However, many symptoms are shared across these disorders in animals[28]. The open field test is an experimental test that measures the general motor activity level, anxiety, and willingness to explore in animals (usually rodents) in scientific studies. Animals show an innate aversion to bright, open areas. However, they are also motivated to explore perceived threatening stimuli. Reduced anxiety levels lead to increased exploratory behavior. Increased anxiety leads to reduced exercise and tends to move closer to the walls of the field[29–31]. The open arms of elevated plus maze test are designed based on the natural propensity of rats to explore novel environments and avoid bright, high and unprotected areas. The elevated corticosterone levels of rats in open arms showed that restraint with open arms can lead to stress response[32].

The concept of neuroimmunology was established by Han Selye in the 1950s, which marked the beginning of the exploration of the relationship between the immune system and the nervous system[33,34]. However, the roles of the adaptive immune system in the face of stress and in the control of mood and anxiety have not yet been defined. It has been reported that psychological stress may induce significant changes in the immune system[35,36]. In our studies, we found that the chemokine CCL5 was significantly elevated in stressed individuals in the acute phase both in humans and animals, but MCP-1, IL-1β, IL-2, IL-4, IL-6, TGF-α, TNF-α and BDNF not. CCL5 has been previously reported to be altered in subjects displaying anxiety[37], depression[38], PTSD[39] and chronic stress[40], which further confirmed our findings. Although, many studies have indicated that CCL5 is not altered in mental disorders[41,42]. These results were different from those of many other studies[40,43], possibly because we measured cytokines and chemokines one month after a stressful event. In addition, the literature within the field is inconsistent due to variables such as age, species, and cultural factors, as well as testing conditions such as the time of day and environment. However, these studies did not evaluate in depth the relationship between CCL5 and the brain and anxiety. The phenomenon that CCL5 can trigger chemotactic activity on T lymphocytes and monocytes suggests that CCL5 is associated with chronic inflammatory diseases, such as depression[44] and cancers[45]. The results of the current study are consistent with the higher percentage of total lymphocyte counts during acute laboratory stress in young adults[46]. CCL5 is produced by many cell types, including myocytes, endothelial cells and adipocytes[18]. We screened multiple immune tissues for CCL5 induction, showing that CCL5 was robustly induced in CLNs. Stress-induced anxiety is a complex process evoking systemic immune responses as well as direct responses in brain tissues[23]. Previous study showed that stress can cause multiorgan damage and evoke a systemic immune response, including cytokine and chemokine production, by affecting translational pathways[23]. In this study, we demonstrated that inhibiting the translational pathways using Anisomycin could reduce the stress-induced anxious behavior. Translational regulation of mRNA shows promise as a safe and specific treatment to combat stress-induced anxiety disorders.

While multiple brain regions have previously been implicated in anxiety control, the need for anxiety-like behavior and inflammation in these sites remains an open question, based on the findings of electrical recordings[47], retrograde tracing[48], or functional imaging experiments[49], the necessity of any of these sites for generating anxiety-like behavior and inflammation remains an open question. Here, we demonstrated that the red nucleus (RN), which was identified by PRV-based retrograde tracing from CLNs, constitutes a critical hub in the brain circuits that control anxiety-like behavior and inflammation. It is well known that the prefrontal-thalamic circuits are related to affective functions[50].

RN, a deep gray matter structure composed of highly complex microstructures, is the important node of the prefrontal-thalamic circuits[51,52]. Previous studies showed that the RN is mainly responsible for motor-related behaviors, including motor coordination and control. Besides, the effects of RN on emotional functions, such as the initiation of emotions[53], anticipation of rewards[54] and processing of socioemotional scenes[55], has also been proposed. Besides, the stimulation of the RN in rodents has been found to elicit long-lasting and intense analgesia[56]. Studies have suggested that the molecular regulation of antinociceptive responses in the rat RN may involve a large network of inflammatory mediators and cytokines. Some proinflammatory cytokines, such as TNF-α, IL-1β, and IL-6, could be inhibited through microinjections of monoclonal antibodies into the RN, relieving neuropathic allodynia[57,58]. Although these experiments showed the relationship between the RN and inflammatory mediators, the associated immune organs were not identified. Our results causally demonstrated that stress regulates CLNs through RN glutamatergic neurons. The activation of the RN in rats produced anxiolytic effects, which were probably mediated by the sparse anatomical connections of the RN with the components of the ascending anxiolytic system. In our study, we found that these RN glutamatergic neurons received M1 projections. Primary motor cortex (M1) excitability is correlated with anxiety-related personality traits[59]. M1 stimulation induced a general improvement in anxiety, depression and pain compared to sham stimulation in fibromyalgia patients[60]. In our study, inhibiting the M1-RN pathway induced anxiety-like behavior and CCL5 synthesis, demonstrating a pathway from the M1 directly to the RN for regulating anxiety-like behavior. To confirm the role of the RN in stress-induced anxiety, we further analyzed the fMRI findings of stressed individuals. Compared with controls, stressed individuals exhibited significantly decreased bilateral FC in the RN. These results further confirmed the function of the RN in anxiety and inflammation.

Norepinephrine (NA) can be released by sympathetic nervous system after stressful events, and affects the inflammatory response[25]. Our results demonstrated that NA was significantly increased both in stressed rats and in RN inhibited rats. NA preferentially modulates memory CD8 T-cell function, inducing inflammatory cytokine production[61]. The increase in CCL5 may be fed back to the brain through noradrenergic fibers, thereby affecting anxiety-like behavior. However, this is a very complicated adjustment process, and more in-depth experiments are needed to explore the mechanisms.

This study has some limitations. First, the blood of stressed psychiatrists was collected and followed up over 12 months. The first time that we collected the blood of stressed psychiatrists was one month following their return from stressful environments. This time gap might have caused us to miss some information. Cortisol levels were diminished in individuals from the stressed group, rendering the stress questionnaire the only measure by which we confirmed these individuals' elevated stress levels. Second, restraint stress regulates CLNs through RN glutamatergic neurons, but we have yet to understand the mechanism of communication between the RN and CLNs. Third, both female and male rats were found to have increased CCL5 levels in blood after restraint stress exposure. However, restraint stress did not lead to anxiety-like behavior in female rats. Further exploration and research are needed to determine the effect of stress on RN functions in female rats. We are carrying out follow-up studies to examine the mechanisms of communication between the RN and CLNs in both female and male rats. In addition, none of the controls (neither humans nor rats) experienced any kind of novel nonstressful event; therefore, the possibility that the RN neurons identified in this study were linked to a learning or novel experience cannot be entirely excluded. For fMRI results, the main analysis compared functional connectivity between the red nucleus and every other voxel in the brain in a relatively small sample (~35–40 subjects per group) and then used a cluster threshold

to correct for multiple comparisons and identify significant differences in the stressed group. There are several red flags here: the samples are relatively small for performing tens of thousands of tests, and the reported effects are very small as well, both in magnitude and spatial extent. Further complicating things, subcortical / midbrain signals are notoriously noisy in resting state fMRI datasets.

In summary, we report evidence that supports an essential role for a group of RN neurons in controlling stress-induced anxiety-like behavior and the inflammatory response in CLNs. Our study showed that the RN has been functionally demonstrated to participate in anxiety and inflammation.

## Methods

### Clinical study design and participants

To measure the levels of immune indicators of stress in stressed people, we recruited psychiatrists who worked on the front line during the COVID-19 pandemic in Wuhan between February 21 and March 31, 2020. The acute phase interview was carried out one month after they left the stressful environment, and the two follow-up evaluations were six months and one year later. Psychiatrists who worked in non-COVID-19 second-line care were recruited as controls at the baseline interview. This clinical study was approved by the Ethics Committee of Shanghai Mental Health Center (2020–10). Written informed consent was provided by all participants prior to their enrollment. All human participants in our research underwent COVID-19 PCR/antigen testing, and the results were negative.

### Clinical evaluation

The Generalized Anxiety Disorder 7-item (GAD-7), the Patient Health Questionnaire-9 (PHQ-9), the Self Reporting Questionnaire 20-Item (SRQ-20) and the PTSD Checklist for DSM-5 (PCL-5) were used to evaluate the mental disorder symptoms of all participants. The Generalized Anxiety Disorder 7-item (GAD-7) is an easy-to-use tool for initial screening for generalized anxiety disorder. When screening for anxiety disorders, a score of 8 or greater represents a reasonable cut-point for identifying probable cases of generalized anxiety disorder. The following cutoffs indicate the severity of anxiety: score 0-4, minimal anxiety; score 5–9, mild anxiety; score 10-14, moderate anxiety; and score greater than 15, severe anxiety[62,63]. The Patient Health Questionnaire-9 (PHQ-9) is a multipurpose instrument for screening, diagnosing, monitoring, and measuring the severity of depression. Total scores of 5, 10, 15, and 20 represent cut-points for mild, moderate, moderately severe, and severe depression, respectively[13,64]. The Self Reporting Questionnaire 20-Item (SRQ-20) can be used to detect nonspecific psychological distress; subscales include depression/anxiety, somatic symptoms, reduced vital energy and depressive thoughts. Items are scored as 0 (symptoms absent) or 1 (symptoms present). Scores range from 0 to 20, with scores >10 indicating mental distress. Responses are yes or no[65]. The DSM-5 PTSD Checklist (PCL-5) is a 20-item self-reported measure that assesses 20 DSM-5 PTSD symptoms. When using the PCL-5 severity score for interim diagnosis, the characteristics of the respondent's environment should be considered[66,67].

The objectives of the assessment should also be considered. A lower cut-point score should be considered at screening or when maximum detection of possible cases is required. Higher cut point scores should be considered when attempting to make a provisional diagnosis or minimize false positives. Demographic data was self-reported by participants, including gender (male or female), age, marital status (unmarried, married, divorced, or widowed), and years of education.

### Immune indicator measurement

Peripheral blood samples of the stressed psychiatrist group were collected in the acute phase and at the two follow-up interviews, while those of the control group were collected only at baseline and at the 12-month follow-up. The plasma cytokine and chemokine levels were measured for all psychiatrists. Plasma cytokine and chemokine levels were measured using the ProcartaPlex™ Human Inflammation Panel 10 Plex (Invitrogen) in a Luminex 200 multiplexing instrument (MilliporeSigma). This system allowed us to obtain quantitative measurements for 10 different cytokines and chemokines, including MCP-1, IL-1β, IL-4, IL-10, TNF-α, IFN-γ, IL-2, IL-6, TGF-α and CCL5. A CCL5 (RANTES) ELISA kit (ab174446, human RANTES ELISA Kit (CCL5), Abcam) was used to confirm the Luminex results and measure the CCL5 concentration at follow-ups. In addition, we determined the plasma BDNF and cortisol concentrations using a human BDNF Quantikine ELISA kit (DBD00, R&D Systems) and a human cortisol competitive ELISA kit, (Invitrogen, Thermo Fisher).

### fMRI

Eighty-three participants were recruited, including forty-four stressed psychiatrists and thirty-nine demographically matched controls. Brain scanning was performed using the 3.0 T SIEMENS MAGNETOM Prisma at Renji Hospital. Resting-state functional images were acquired using an acquisition time of 452 s, a repetition time of 2000 ms, an echo time of 30 ms, a flip angle of 90°, a voxel size of 3.3*3.6*2.4 mm, a field of view of 230*230 mm$^2$, a slice number of 70 and 220 time points. Structural MRI images were also acquired using an acquisition time of 221 s, a repetition time of 1800 ms, an echo time of 2.28 ms, a flip angle of 8°, a voxel size of 1*1*1 mm, a field of view of 256*256 mm$^2$, and a slice number of 160. Participants were asked to close their eyes and relax but not fall asleep during scanning.

**Imaging acquisition.** DPABI_V3.0 software (Data Processing & Analysis for Brain Imaging, http://www.restfmri.net) and SPM12 in MATLAB 2019a (Statistical Parametric Mapping Software, http://www.l.ion.ucl.ac.uk/spm) were used to analyze the resting Fmri. We removed the first 10 volumes and left 210 volumes for everyone. We excluded participants with a maximum head movement greater than 2 mm during resting fMRI scans, or participants with an absolute motion rotation greater than 2° based on pretreatment analyses. 4 controls and 7 emergency psychological responders (EPRs) were excluded due to excessive head motion, leaving the final sample of 37 EPRs and 35 controls described above. We regressed the nuisance covariates and examined the structural images. Manual reorientation was performed to align the anterior commissure with the origin, and then co-registered with the mean functional image. DARTEL (Diffeomorphic Anatomical Registration Through Exponentiated Lie Algebra) algorithm was applied to segment the cerebrospinal fluid, white matter, and gray matter. Then, based on the parameters obtained during segmentation, DARTEL was used to normalize the functional images into the standard Montreal Neurological Institute (MNI) space in a $3 \times 3 \times 3$ mm voxel size. It is then smoothed using a Gaussian kernel with a 4 mm full-width at half-maximum value (FWHM). Finally, the execution time bandpass filtering (0.01–0.1 Hz). In the following analyses, average FD (frame displacement) was calculated as a covariate to exclude potential influence of head motion.

**Functional connectivity analysis.** First, we defined two functional regions of interest (ROIs) in the analysis: left and right red nuclei (RNs) and generated ROIs for RN using WFU PickAtlas software[68]. We then used bilateral RN as ROI for voxel rsFC analysis between each seed and whole brain voxels via DPABI. Calculate the average time series of the seed and correlate it with the time series of all other whole voxels. The relevant r-mapping is then converted to a z-map by Fisher's r-to-z transformation. The full factorial model was used to test the rsFC difference of the seven seeds in the three groups by SPM12 in the general linear model, and the mean FD was used as the covariate. Finally, rsFC differences between EPR and control were examined. If

the cluster reaches the voxel level threshold $P < 0.001$ with the cluster level P (FDR correction) <0.05 and the cluster size >30 voxels, the cluster is considered to exhibit significant group differences.

## Rats

All procedures involving rats were approved by the Shanghai Jiao Tong University Institutional Animal Care and Use Committee and in line with the National Institutes of Health (NIH) guidelines. Rats were kept in fully equipped facilities in the laboratory animal center of Shanghai Jiao Tong University. Both male and female rats were used in our study. For restraint stress and unpredictable chronic mild stress (CUMS) models, SD rats (Jackson Labs) at PND 64–90 were used in our experiments. For the early life stress model, as we did before[69]. Simply, two female SD rats in each animal facility mate with one male. Males are removed after a week and females are separated into separate cages 1–3 days before giving birth. We applied an unpredictable maternal separation protocol in which whole litters were moved to other cages apart from their mothers for 4 h/day[69]. The offspring are weaned at PND 21, and the males and females are weaned separately, placed in cages of 3–4 rats.

## Restraint stress

An established restraint stress protocol was used to induce anxiety-like behavior in rats. Adults (PND 63–70) were placed in a ventilated plastic cylinder for 2 h a day for 3 days. Restraint stress is provided by placing rats in restraining cylinders that fit closely with their body size and drilled holes to allow free breathing, and then housed the rats in homecare. The restraint time, duration, and frequency are the same for different batches. Control rats are moved from the feeding chamber to the testing chamber and processed for 2–4 min before returning to the feeding chamber after 2 h.

## Chronic unpredictable mild stress (CUMS)

The animals were exposed to a variable sequence of mild and unpredictable stressors for 5 weeks. A total of nine different stressors were used, with two stressors per day, including cage tilting (45°) for 12 h, paired caging for 12 h, water deprivation for 12 h, wet cages for 12 h, continuous illumination in the dark cycle for 12 h, restraint stress for 2 h, cold swimming for 5 min, tail pinch for 1 min, and food deprivation for 12 h.

## Early-life stress

To evaluate the effects of early-life stress on immune response, we used the maternal separation for 2–4 h/day in rats[70]. Unpredictable maternal separation protocol was used in our study[69]. As shown in our precious study, postnatal stress had no effect on the offspring survival, food or water intake and coat condition or barbering behavior.

## Behavioral testing procedures

We conducted these behavioral tests in the light cycle. Tracking software (Ethovision, Noldus, Netherlands) was used to videotape and analyze the behavioral tests data. During tests, the order of rats was random. Rats had a minimum acclimation period of 3 days prior to the behavioral tests. For pre-experimentation handling, we removed the rat from its cage and either simply held or stroked the animal for a duration of no less than one minute at least once per day for 3 days prior to behavioral assays.

**Open field test**. The open field apparatus is a $60 \times 60 \times 60$ cm plexiglass arena with transparent walls and a blue floor with a central area of $30 \times 30$ cm. The test was conducted in dim light conditions without the presence of experimenters. Test rats are placed in a corner of the open field. Over a 10-min test period, the total distance moved, the speed of movement, the frequency of transfers between the center zone and the surrounding area, and the time spent in the center zone

were recorded. Clean the device with 75% alcohol between tests. No animals were excluded from the open field test.

**Elevated plus maze test**. The apparatus consists of two open arms ($60 \times 15$ cm) and two closed arms ($60 \times 15$ cm), with a gray wall (50 cm) without a roof connected by a central square platform ($15 \times 15$ cm), 60 cm from the ground. Place the test rat in the central area of the EPM, facing the open arm. Allow the rat to move freely in the maze for 5 min. Noldus software automatically analyzes the distance traveled, the time spent on each arm, and the number of times it has entered each arm. Animals that fell from their open arms were excluded from the analysis. Clean the device with 75% alcohol between tests.

## Lymphadenectomy

The rat was anesthetized using oxygen and isoflurane (5%) (R510-22-10, RWD Life Science Co., Ltd., China) with a flow rate of 1.0 L/s in an induction chamber. To verify that the rat was asleep, its tail was pinched and rolled. If the rat did not react, the surgery was performed. Cervical lymph nodes (CLNs) were removed. Ointment was applied on the rat eyes before surgery to avoid eye dryness. A very small incision (approximately 5 mm) in the neck was made with sharp scissors. The incision was stretched with 2 forceps to see the LNs. The incision depth could reach 10 mm. The superficial CLN appeared grayish or darker than the surrounding fat. The fascia (the thin membrane covering the fat and tissue) on top of the LN was pinched with one set of forceps and pulled lightly without breaking the surrounding tissue. The second set of forceps was placed as far as possible underneath the LN. With the set of first forceps, the fascia was broken, and the superficial CLN was removed. The absence of hairs in the wound was verified.

## ELISA

Corticosterone/CCL5/NA concentrations were measured in rat serum. Blood samples were collected from the tail tips of rats into capillary tubes. Blood was allowed to clot for 30 min at room temperature and centrifuged at $3000 \times g$ for 20 min at 4 °C, and serum was collected and stored at −80 °C. Corticosterone was quantified by enzyme-linked immunosorbent assay (ELISA, parameter corticosterone assay KGE009, R&D systems) according to the assay instructions. CCL5 was quantified by ELISA (EK0496, Rat RANTES ELISA Kit, BOSTER) according to the assay instructions. NA was quantified by ELISA (D751020, (NA/NE) ELISA, Sangon Biotech) according to the assay instructions. All samples were run in duplicate and counterbalanced across plates and were within the standard curve.

## Flow cytometry and intracellular cytokine staining

Lymph nodes or blood samples were subjected to flow cytometry using Celesta (BD) and the following fluorescence-labeled antibodies: BV510-conjugated anti-CD45, APC-conjugated anti-CD3, FITC-conjugated anti-CD4, Percp-5.5-conjugated anti-CD8, and PE-conjugated anti-CD45RC. CD8+T cells were isolated from the lympho-nodes (LNs) of adult rats using anti-CD8 magnetic beads (Miltenyi Biotec). For intracellular cytokine staining, T cells were obtained immediately from the CLNs of stressed rats and then permeabilized using intracellular staining permeabilization wash buffer (eBioscience, 88-8824-00). Then, intracellular CCL5 was analyzed by flow cytometry.

## Immunohistochemistry and imaging

Rats were terminally anesthetized with isoflurane and $O_2$ and transcardially perfused with PBS followed by 4% paraformaldehyde (PFA), and brains, lymph nodes and spleens were postfixed for 24 h in 4% PFA at 4 °C. Brains, lymph nodes and spleens were then dehydrated in 30% sucrose at 4 °C until sinking to the bottom of the 50 mL tube. Coronal sections were made on a freezing vibratome at a thickness of 30 μm. Sections of selected areas were blocked by incubation in PBS plus 4%

normal goat serum and 0.1% Triton X-100 for 1 h at room temperature and subsequently incubated with c-fos antibody (1:5000, rabbit, Invitrogen), vGlut1 antibody (1:500, rabbit, Invitrogen), GABA antibody (1:200, rabbit, Invitrogen), or CCL5 antibody (1:500, rabbit, Invitrogen) at 4 °C overnight. Incubated slices were then incubated for 2 h with a 1:1000 dilution of Alexa Fluor 488 goat anti-rabbit IgG (1:1000, Thermo Fisher Scientific) at RT. The sections were mounted on slides and coverslipped. All images were obtained using FLUOVIEW FV1200 confocal microscopes (Olympus) and Olympus VS200. Digitalized images were analyzed using Fuji (NIMH, Bethesda MD, USA).

## RNA isolation and qRT–PCR
Frozen tissues were used to isolate total RNA by QIAzol lysis reagent and purified using the miRNAeasy mini kit (Qiagen). A high-capacity cDNA reverse transcription kit (Life Technologies) was then used to obtain cDNA. PCR experiments were done twice using SYBR Premix Ex Taq (Takara Bio). the CFX96 Touch Real-Time PCR Detection System (Bio-Rad) was applied to detect and analyze the fluorescence signals. The primers are exhibited in Table S5. To normalize the values, the values of GAPDH mRNA were used, and then the levels are normalized to those of controls.

## RNA-seq analysis
The CLNs from rats were dissected for RNA-Seq. Total RNA was extracted using TRIzol Reagent (Invitrogen) according to the manufacturer's protocol. Read count normalization and gene expression estimation were performed by HTSeq. Samples were filtered for protein-coding and long noncoding RNAs, raw counts were summed across all samples (Table S5), and the bottom 25% were removed to eliminate genes with very low expression. Subsequently, pairwise differential expression comparisons were performed using Voom Limma. A nominal significance threshold of $p < 0.05$ and fold change >2.0 was used. Enrichment between gene lists was analyzed using DAVID (2021 Update). Heatmap and volcano plot analyses of expression levels were analyzed using R4.2.0.

## Retrograde transsynaptic PRV tracing
For retrograde transsynaptic tracing experiments, rats were anesthetized with isoflurane and $O_2$ and placed on a heated pad. The lymph node was exposed through an incision in the side of the neck. The injection was performed via the insertion of a glass micropipette into the lymph node to a depth of 0.5–1.0 mm. For each lymph node, 1 μL of PRV-RFP ($1.71 \times 10^8$ PFU/mL) was injected at a speed of 100 nL/min using a pump. The glass micropipette was held in place for 5 min after each injection. After PRV injection, the lymph node was put back into the neck, and the wound was sutured. For histological experiments, rats were sacrificed on Days 2, 3, 4, 6 after PRV injection.

## AAVs
We used the following AAVs in the study: AAV-hSyn-GCaMP6f (AAV2/9, titer: $2.59 \times 10^{12}$ viral particles/mL; for optical fiber-based $Ca^{2+}$ recordings), AAV-CaMKIIα-hM4D(Gi)-mCherry (AAV9, titer: $6.52 \times 10^{13}$ viral particles/mL; for chemogenetic inhibition of RN excitatory neurons), AAV-CaMKIIα -hM3D(Gq)-mCherry (AAV2/9, titer: $5.29 \times 10^{12}$ viral particles/mL; for chemogenetic activation of RN excitatory neurons), AAV-CaMKIIa-Cre-GFP (AAV2/8, titer: $1.14 \times 10^{13}$ viral particles/mL; for ablating RN excitatory neurons), AAV-EF1α-DIO-taCasp3-TEVp (AAV2/9, titer: $1.16 \times 10^{13}$ viral particles/mL; for ablating RN excitatory neurons), AAV-hSyn-Cre (AAV1, titer: $1.52 \times 10^{13}$ viral particles/mL, for anterograde trans-synaptic tracing experiments), AAV-EF1α-DIO-GCaMP6s (AAV9, titer: $1.34 \times 10^{13}$ viral particles/mL, for optical fiber-based $Ca^{2+}$ recordings), AAV-CaMKIIα-DIO-hM4D(Gi)-mCherry (AAV9, $5.29 \times 10^{13}$ viral particles/mL; for chemogenetic inhibition of M1-RN excitatory neurons), and AAV-CMV-beta globin-cre-GFP (AAV2, titer:

$1.36 \times 10^{12}$ viral particles/mL; for chemogenetic inhibition of M1-RN neurons). All AAVs were purchased from Vigene Biosciences (Shandong, China), BrainVTA Technology Co., Ltd.

## Stereotaxic surgery
Rats were anesthetized with isoflurane (R510-22, RWD China, 4–5% induction, 1–2% maintenance), by inhalation of $O_2$, and then placed in a stereotaxic frame. Exposing the skull surface, bilateral AAV infusion was administered bilaterally using a 33-gauge syringe needle (Hamilton) at a rate of 0.067 μL/min. The brain coordinates of the injection were selected according to the rat brain atlas: RN (AP: −6.00 mm; ML: ±0.80 mm; DV: 7.60 mm from dura mater) and M1 (AP: +1.00 mm; ML: ±2.40 mm); DV: 1.60 mm from dura). Rats that had been injected with AAVs were allowed 3 weeks to recover and for the viral transgenes to be adequately expressed before being subjected to behavioral experiments. The viral injection amount was 0.8 μL for M1 and 0.5 μL for RN. Animals where the virus injection space is inaccurate or the virus spreads significantly outside the target area are excluded from the analysis.

To functionally manipulate the RN, AAV-CaMKIIα-hM4D(Gi)-mCherry, AAV-CaMKIIα -hM3D(Gq)-mCherry or a mixture of AAV-CaMKIIα-Cre-GFP and AAV-EF1α-DIO-taCasp3-TEVp was bilaterally injected into the RN. AAV-CaMKIIα-mCherry or AAV-flex-taCasp3-TEVp was bilaterally injected into the same coordinates as the control group. To functionally manipulate the M1-RN neural circuit, AAV-hSyn-Cre was bilaterally injected into the M1, and AAV-CaMKIIα-DIO-hM4D(Gi)-mCherry was bilaterally injected into the RN. Following viral injection, rats were allowed to recover for at least 3 weeks before being subjected to behavioral tests and histological analysis.

## Optical-fiber-based $Ca^{2+}$ recording of freely behaving rats
Optical-fiber-based $Ca^{2+}$ recording was performed using a Multi-Channel Fiber Photometry Device (inper, Hangzhou, China). An optical fiber (200 μm in diameter, NA 0.37, Black Ceramic Ø2.5 mm, inper, China) was slowly inserted into the RN. The optic fiber was glued to a short cannula, and the fiber tip was extended approximately 1 mm out of the cannula. Neuronal $Ca^{2+}$ signals and behavior videos were recorded simultaneously using InperStudio (inper, Hangzhou, China). All fiber photometry data and behavior videos were aligned offline through event markers using Inper Data Process (inper, Hangzhou, China) and analyzed by metlab2022a.

## Drugs
To inhibit mRNA translation in lymph nodes, we intraperitoneally administered anisomycin (S7409, Selleck) dissolved in 2% DMSO + corn oil at a dose of 7 mg/kg before exposure to restraint stress. For control experiments, the same volume of 2% DMSO + corn oil was injected into the control rats.

For chemogenetic inhibition of RN excitatory neurons, at least 3 weeks after viral injection, we intraperitoneally administered clozapine N-oxide (CNO, Selleck) dissolved in physiological saline (0.9% NaCl) at a dose of 3 mg/kg in a volume of 1 mL/kg 30 min before the behavioral test. For chemogenetic activation of RN excitatory neurons, at least 3 weeks after viral injection, we intraperitoneally administered clozapine N-oxide (CNO, Selleck) dissolved in physiological saline (0.9% NaCl) at a dose of 3 mg/kg in a volume of 1 mL/kg 2 h and 30 min before exposure to restraint stress. After 3 days of restraint stress exposure, the rats were subjected to behavioral tests.

## Systemic neutralization of CCL5
Monoclonal antibodies recognizing CCL5 or control IgG were administered intraperitoneally (250 μg/rat in sterile PBS) to rats injected with AAV-CaMKIIα-hM4D(Gi)-mCherry in the RN immediately after CNO injection.

## Brain slice preparation

Rats were anesthetized with isoflurane and perfused with 50 mL of chilled dissection buffer (25.0 mM NaHCO$_3$, 1.25 mM NaH$_2$PO$_4$, 2.5 mM KCl, 0.5 mM CaCl$_2$, 7.0 mM MgCl$_2$, 25.0 mM glucose, 110.0 mM choline chloride, 11.6 mM ascorbic acid and 3.1 mM pyruvic acid, gassed with 95% O$_2$ and 5% CO$_2$). Coronal M1 and RN slices of 300/200 μm thickness were sectioned in chilled dissection buffer with a VT1000s vibratome (Leica). Slices were incubated in oxygenated artificial cerebrospinal fluid (ACSF), left to recover for 90 min at 37 °C, and then transferred to room temperature until the electrophysiological recordings were obtained.

## Electrophysiological recordings

Prepare 200 μm thick brain slices containing RN using a vibratome (Leica) and ice-cold cutting solution (kynurenic acid 3 mM, NaHCO$_3$ 26.2 mM, MgCl$_2$ 4.9 mM, CaCl$_2$ 1.22 mM, glucose 1.25 mM and sucrose 225 mM). Electrophysiological recordings were done as in previous study[69]. Simply, incubate sections containing RN in ACSF at 31 °C and then at room temperature. Then, the sections were transferred to the recording chamber. Perfused ACSF at 2 mL/min and oxygenate with 95% O$_2$ and 5% CO$_2$. Neurons were observed using an IR camera on Olympus U-TLUIR. Whole-cell patch-clamp recordings (multiclamp 700B amplifier) were applied to record the neurons. ACSF were perfused at 4 mL/min during recording. Borosilicate glass pipettes were pulled on the Sutter Instrument P-97 micropipette puller and then used to record. After fire polishing, the resistance range was 2–5 MΩ to improve the quality of the seal. For plasticity experiments and excitatory transmission recordings, the internal solution were as follows: K gluconate 140 mM, KCl 5 mM, creatine phosphate 10 mM, MgCl2 2 mM, Na2ATP 4 mM, Na3GTP 0.3 mM, EGTA 0.2 mM and HEPES 10 mM, pH 7.3, osmolarity 300.

100 μM NASPM was applied. For IPSC recordings, cells with a capacitance higher than 30 pF were screened, and the internal solution contained 30 mM K gluconate, 100 mM KCl, 10 mM creatine phosphate, 4 mM MgCl$_2$, 3.4 mM Na$_2$ATP, 0.1 mM Na$_3$GTP, 1.1 mM EGTA and 5 mM HEPES, pH 7.3, osmolarity 289. The internal solution for electrophysiological profiling experiments contained 130 mM K gluconate, 10 mM creatine phosphate, 4 mM MgCl$_2$, 3.4 mM Na$_2$ATP, 0.1 mM Na$_3$GTP, 1.1 mM EGTA and 5 mM HEPES, pH 7.3, osmolarity 289. Current clamp recordings were filtered at 2.5 kHz and sampled at 5 kHz. Voltage clamp recordings were filtered at 2.5 kHz and sampled at 10 kHz. To examine the effect of RS on the excitability of pyramidal neurons in RN, we injected a continuous current from −100 pA to 450 pA in steps of 50 pA for 500 ms. Current was injected into the current clamp every 60 seconds. Electrophysiological recordings were performed under double-blind conditions.

## Western blotting

Both lymph node and brain tissues were extracted by RIPA buffer (Beyotime, China) containing 1% phenylmethanesulfonyl fluoride (PMSF; Beyotime, China) and 1% protein inhibitor (PI; Beyotime, China). The protein concentrations were determined using the Pierce™ BCA Protein Assay Kit (Thermo Scientific). Proteins were separated by 4–12% SDS–PAGE gel electrophoresis and then transferred electrophoretically onto nitrocellulose membranes (0.45 μm; Bio Basic, Inc.). For immunoblotting, the membrane was blocked with 5% nonfat milk. After incubation with a specific primary antibody, a horseradish peroxidase-conjugated secondary antibody was applied. The positive immune reaction signal was detected by ECL detection kits (GE Healthcare, UK).

## Statistical and reproducibility analyses

Clinical data were analyzed using SPSS statistics 26. Analyzed the statistical significance of all animal data using GraphPad Prism software. To compare multiple groups or multiple measurements, we used one-way analysis of variance (ANOVA). For multigroup or multiple analyses, we first assessed the normality of the distribution using normality and lognormality tests. A $p > 0.05$ meant that the distribution was normal. Instead, nonparametric tests were used. Normally distributed data were tested for homogeneity of variance. The Brown–Forsythe test and the Bartlett test were used to test for homogeneity of variance: $p > 0.05$, homogeneity was met; $p < 0.05$, it was violated, then Brown–Forsythe and Welch ANOVA tests were used. For one-way ANOVAs with multiple comparisons across groups, Tukey's multiple comparisons tests were used. For between-subject comparisons after two-way ANOVA, Bonferroni's multiple comparison tests were used, and for a within-subject comparison Fisher's single comparison tests were used. Littermates are randomly assigned to experimental groups, and animals are tested in random order. The data were analyzed by researchers who were unaware of the experimental conditions and were not informed during data collection.

## Reporting summary

Further information on research design is available in the Nature Portfolio Reporting Summary linked to this article.

## Data availability

Source data are provided with this paper. All the data associated with this study are present in the paper or the Supplementary Materials. The raw sequence data reported in this paper have been deposited in the Genome Sequence Archive[71] in National Genomics Data Center[72], China National Center for Bioinformation / Beijing Institute of Genomics, Chinese Academy of Sciences (GSA: CRA012614) that are publicly accessible at https://ngdc.cncb.ac.cn/gsa. Source data are provided with this paper.

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

## Acknowledgements

This work was supported by grants from the National Natural Science Foundation of China (82230045) (Z.W.), (32271066) (D.-D.S.), Shanghai Science and Technology Committee (20XD1423100) (Z.W.) and (23QA1408300) (D.-D.S.), Shanghai Municipal Education Commission (2021-01-07-00-02-E0086, 20161321) (Z.W.) and Lingang Laboratory (LG202106-03-02) (Z.W.). Thanks to the psychiatrists who worked on the front lines during the COVID-19 pandemic in Wuhan for their participation in this study.

## Author contributions

Z.W. and D.-D.S. designed the study, and D.-D.S. wrote the article. Z.W. and D.-D.S. coordinated the study and collected and analyzed the data. S.S., Z.K., H.H., C.Z. and Z.W. provided psychological support for the patients. D.-D.S., Y.-D.Z., S.Z. and B.-B.L. performed the animal experiments. M.-Y.C. analyzed the fMRI data. All authors critically revised the article or contributed important intellectual content.

## Competing interests
The authors declare no competing interests.
