## [Peer Review File · Nature Communications]

Reviewers' Comments:

Reviewer #1:

Remarks to the Author:

Overall, this is a potentially interesting paper describing a novel circuit from motor cortex (M1) to red nucleus (RN) to cervical lymph nodes (CLN) that controls anxiety behavior, possibly through CLN production of CCL5. While the story is interesting and potentially important, there are several significant weaknesses that should be addressed in a revision. Most critically, the authors need to provide further evidence to support the circuit in question and provide casual evidence to link activation of M1-RN-CLN to downstream effects of CCL5 that promote anxiety.

1. Chemogenetic strategies might not be appropriate for defining this monosynaptic circuits. Do either M1 neurons projecting to RN or RN neurons projecting downstream to CLN contain bifurcating axons targeting multiple downstream brain structures. If true, as is the case with most glutamatergic projections neurons, the authors would need to repeat these studies using optogenetic terminal stimulation studies to target monosynaptic connections between brain regions.

2. Is the connection between RN and lymph nodes mono or polysynaptic? If the latter, please provide a more comprehensive tracing map to show the intermediary connections between RN and lymph nodes.

3. I guess the most important question is how CCL5 being released from CLN is regulating anxiety? Much of the work centers on establishing the circuit from RN to lymph nodes. Indeed, chemogenetic manipulations of RN shows it is necessary and sufficient to induce anxiety, but there are no experiment that prove a causal relationship between RN activation of CCL5 and anxiety. It's quite possible the anxiety phenotype initiated by chemogenetic manipulation of RN is due to it actions on downstream neural circuitry and is unrelated to the RN-CVN. This is particularly important given that circuit specific optogenetic approaches (see point 1) were not used.

4. Related to point 3 above, what and where is CCL5 acting upon to promote anxiety? Does it feed back up and enter the brain to control neural activity in anxiety-provoking brain regions?

5. Data in figure 5 is interesting and shows that deleting or inhibiting vGluT cells in the RN increases anxiety and CCL5. However, it's unclear if the anxiety has anything to do with the increase in CCL5. Can you prevent the anxiety effects elicited by inhibition or ablation of RN neurons by co-administering a monoclonal antibody against CCL5 in the periphery?

6. In Fig 6, the authors show that chemogenetic inhibition of M1 neurons projecting to RN regulates their activity and anxiety behavior. As mentioned above, this is not necessarily circuit specific if those cells in the M1 bifurcate and project elsewhere in the brain. Further, it's not established that activation of this pathway ultimately affects CCL5 to control anxiety.

7. For cytokine and chemokine analysis, some of the values are quite low (ie., <2-3pg/ml). Was this in the range of detection of the ELISA kit used?

Reviewer #2:

Remarks to the Author:

In this study, Shi et al identify CCL5 as an increased cytokine in stressed patients during the acute phase. In a rat model of restrain stress (RS), they demonstrate the increased expression of CCL5 in cLN and by RNAseq identify translation as potential mechanism involved in this increased expression. Then, the authors try to demonstrate that translation is involved in RS dependent induction of CCL5 in the cervical LN. Additionally, the authors show that RS inhibits the excitability of Red Nucleus glutamatergic neurons leading to anxiety-like behaviour. Additionally, they showed that reduced RN firing induces increased CCL5 expression in the cervical LN, and conversely,

increased RN neurons activation reduces anxiety-like behaviour as well CCL5 expression.

While the neuroscience part is overall convincing, the link to immunology should be strengthened. Furthermore, the paper should be thoroughly restructured to make things clearer. While the role of CCL5 in stress and anxiety has been object of study in the past, the discovery of a cortex-midbrain-LN neuronal circuit is intriguing and deserves attention. The brain circuit is analysed in a detailed and thorough manner, however, as acknowledged by the authors, there is no mention of the circuit connecting the brain with cervical LN (PNS).

Major concerns:

CCL5 has been previously reported to be altered in subjects displaying anxiety (<https://doi.org/10.1016/j.pharep.2014.08.006>), chronic stress (<https://doi.org/10.1016/j.ynstr.2018.02.002>), PTSD (<https://doi.org/10.1002/da.20564>), depression during pregnancy (<https://doi.org/10.1186/s12884-021-04225-2>) as well as animal models (<https://doi.org/10.1016/j.biopsycho.2021.02.765>; <https://doi.org/10.1016/j.celrep.2021.108979>) raising concerns about the novelty of the main discovery.

It starts with the title, but throughout the manuscript the grammar needs a lot of improvement. One of many examples is line 65 'litter' instead of 'little'.

The current formatting of the paper and explanations are thoroughly lacking clarity. E.g. Figure 2r exists twice, whereas Fig 2v is missing. This is not acceptable and is a major concern for the overall quality of the data. How can the underlying data be trusted when the presentation lacks these basic elements?

It needs to be explained what precisely is the stress that human subjects were exposed to. Why are glucocorticoids lower in stressed subjects?

The questionnaires need to be explained in much more detail, not even the acronyms are spelled out, nor what they actually measure. What do the scores mean, how are they interpreted?

It is completely unclear what makes the cervical LN special that it is the predominant source of CCL5 compared to other LNs. This is exciting, indicating local neural control of LN functions but the paper lacks any explanation. The tracing in Figure 3a is very nice labeling, however, do other LNs not project to the RN?

Figure 2: Why are female rats not affected? In humans, both sexes are equally affected, correct? Why is the paper making a point of this?

The order of panels in the Figures and the order of Figures mentioned in the text is extremely confusing. For Figure S2a the 24h CCL5 data are shown later. Also, it is not clear why 24h experiments were performed.

Specifically, Figure panels need to be in chronological order. E.g. the authors discuss FigS2a and d, and then later S2b,c. This is difficult to follow. What is the difference between Fig. S3b,e, and Fig. S2f,h (the latter is female, the former is male?) This needs to be mentioned, only females are highlighted. Figure S8 is mentioned after S9. There are additional examples.

In the rat experiments, are other chemokines altered apart from CCL5, different from the human scenario? A link needs to be made between the rat and the human scenario. Indeed, there seem to be additional chemokines upregulated in Fig. S5a. It is not clear why in Fig S5 CCL5 is not upregulated.

CCL5 levels return to baseline in rats within 2 days, whereas in the human scenario this is still seen after 6 months. How do these stress tests (acute/chronic) compare with respect to CCL5 levels?

Figure 3i is very nice, there still seems to be an increase though, just with overall lower CCL5 levels.

Within the cervical LN, which cells are producing the effect? The authors mention that circulating CD8 T cells are not different, does that mean the effect of cervical LN CD8 T cells is due to tissue residency? This indicates that the effect is not cell autonomous, how does the cervical LN then imprint a CCL5 expression phenotype, and on which cells? What happens after CD8/CD4 T cell depletion?

Figure 3g: CCL5 levels in LN are not convincing. Chemokine levels are notoriously difficult to quantify by imaging.

Fig 3: only 1 time point after PRV infection is not ideal as it is not possible to assess sequential jumps and areas not related to cervical LN (injection sites) can potentially be targeted. RFP in the RN seems quite specific, however, close up pictures of other brain areas would be appreciated. Fig S6: following up on the previous comment, quality is low and specific staining cannot be appreciated. This is potentially an issue as it is not possible to clearly discriminate areas with specific mRFP staining

Figure S5c: a translation inhibitor is given in vivo. Anisomycin in the Figure not Anisamycin. This is a general translation inhibitor, the side effects are wide. How specific is the effect for CCL5?

Minor:

Stats are missing in Figure 5e, and 5p.

Fig1a, the color code seems to be inverted compared to the rest of the figure.

Line 150: "circadian" rhythms ?

Fig 2o-r: this part is very confusing both in figure and the text, the paper would benefit from a more clear and logical explanation, as this part feels detached to the rest

Fig 3b: the 2 RN sections presented look very different, with different expression levels and patterns of RFP

Fig 3d: what about other areas?

Fig4 f-j are not properly explained in the text but just mentioned

Fig 5 is crammed and difficult to digest

Line 301: what is a post hoc immunostaining??

Line 302: reduction in the density of vGlut1+ cells in the injection site (Fig. 5j). Figure did not correspond to the line

Line 306-8: how come these are independent experiments? Are they not performed on the same animals?

ELS and CUMS needs to be briefly explained in the text

Reviewer #3:

Remarks to the Author:

This is a potentially high-impact report integrating human and rodent data and defining a specific brain circuit and neuroimmune mechanism that appears to be important in mediating stress effects on exploratory behavior in an anxiogenic context. The authors show that after stress, CCL5 levels

are increased in humans and rodents; that this effect is driven by lymphocytes in cervical lymph nodes; and it's controlled by glutamatergic projections from the midbrain red nucleus to those lymph nodes. Silencing RN cells decreased exploratory behavior in an open field and increased CCL5 levels after restraint stress.

Overall, this is an important paper that is poised to have a significant impact on the field, pending some revisions. The topic (neuroimmune interactions) is widely understood to be important, timely, and broadly interesting but also understudied. The circuit mapping studies, rigorously examining both inputs and outputs in an unbiased way are a major strength, as is the integration of data from humans and rodents in a way that will surely enhance the impact. There is also an abundance of data from various experimental sources supporting the key conclusions.

That said, there are also some significant issues that would need to be addressed in a revision:

1. In my view the major weakness of the paper is the behavioral data. Behavior in the open field test and elevated plus maze are just very hard to interpret, and the tests don't lend themselves well to pairing with photometry. Fundamentally, it's unclear what the red nucleus cell activity during open field exploration really means -- the link to "anxiety" is tenuous. In the context of all the other data in this paper, I don't view this as a fatal weakness, but I do have some suggestions:

First, I am not sure what we are meant to conclude from the photometry experiment in Fig. 4. The authors write, "In summary, RN glutamatergic neurons played an important role in stress-induced anxiety-like behavior." But it's not really clear what role they played, whether that role is important, and how the actual observations of signals time-locked to center zone entry relate to anxiety. If anything, it seems like periods when the mouse is exploring the center zone would be evidence of *reduced* anxiety or maybe increased exploratory drive, not elevated anxiety, and RN glutamatergic neurons may be playing a role in supporting exploratory behavior in potentially anxiogenic contexts, but not in anxiety per se.

The chemogenetic experiments in Figure 5 help to some degree but it's still a bit confusing and hard to interpret: In Figure 4, the authors find that there's an increase in RN cell activity *after* center zone entry. In Figure 5, the authors find that silencing RN cells reduces time spent in the center zone. How can silencing cells that become active *after* the mouse's decision to enter the center zone result in a reduction in center zone exploration time? One possible interpretation is that RN cells become active during center zone exploration and somehow, they reinforce or sustain this behavior, possibly serving as a safety signal or by activating other brain regions that promote exploration or by inhibiting activity in other brain regions that signal threat. It would be interesting to know whether the authors observed a reduction in center zone entries in addition to a reduction in time spent in the center zone as reported in Fig. 5C. If not, that would be consistent with the idea that RN activity isn't actually initiating exploration / center zone entries, but it is important for sustaining exploration over time.

Optogenetic inhibition experiments timed to occur only after center zone entries or only during time in the periphery would also be helpful. As is, the role for RN cell activity remains unclear. I think optogenetic experiments could go a long way toward clarifying conclusions about what these cells are actually doing with respect to behavior. It would also be helpful to include optogenetic and photometry experiments from more interpretable behavioral assays, although this might be outside the scope of what's feasible given all the other experiments in this paper.

2. Some additional minor issues related to the photometry and DREADD experiments:

- Figure 5: The authors claim that "CNO injection reduced the central zone duration in open field test and open arm ratio in elevated plus maze test in hM4D(Gi) rats, but not in mCherry control rats (Fig. 5c-d)." However, the effect in the EPM was not significant, which undermines the claim about the EPM. With N=8-9 mice per group, the experiment is probably underpowered and could benefit from a larger sample.

- Figure 4: instead of or maybe in addition to plotting many sample photometry traces superimposed on top of each other in Fig. 4c-d, it might be more informative to plot a single representative photometry trace and mark periods when the mouse is in the center zone vs. the periphery. That might provide a better illustration of the group mean traces in Fig. 4h-i, and also a better sense of the temporal dynamics, how they vary with repeated entries, etc.

3. The second conceptual problem with this paper is that the links between the human data and rodent data are messy and a little hard to interpret. The rodent data focus on the association between stress, CCL5, and anxiety. The human data show an increase in CCL5 levels after stress but there is only one measure of anxiety in the human data -- GAD7 self-report scores -- and they are not increased in the stressed group. Further undermining the association between CCL5 and anxiety, the authors observed modest correlations between CCL5 levels and a number of psychiatric symptom scores, but zero correlation with anxiety as measured by GAD7. In humans, CCL5 levels seem unrelated to anxiety. This critique also applies to how the authors interpret their rodent data: for all the reasons noted above, and based on their findings in humans, it seems to me that the data are pointing to a role for CCL5 and red nucleus neuronal activity not in anxiety but rather in stress responsiveness and possibly in regulating exploratory behavior and other behavioral responses to stressful or threatening contexts. These points should be addressed in the Discussion and should probably lead the authors to revise their claims about anxiety, including in the title and abstract.

4. I felt that the paper could also be strengthened by expanding the discussion of the current findings in the context of the existing literature, particularly with respect to what is known about immune system responses to psychosocial stress and their role in driving behavior. The authors touch on this briefly in the second paragraph of their discussion, noting that this general concept has been discussed since at least 1950, and they cite two studies observing immune responses to stress but they do not describe what was observed. In general, my impression of the literature is that previously published results are mixed and inconsistent. It might enhance the impact of the current work to approach this literature review with a little more rigor and state clearly whether stress-sensitive changes in CCL5 signaling have been observed before, also noting negative findings, etc. Same goes for the many negative results in Figure 1 (MCP-1, IL-1b, IL-2, TNFa, etc. etc.). Those negative findings enhance the impact of the present work by establishing specificity. But it would also be helpful if the authors could clarify for readers whether previous studies have examined these molecules in the context of stress, noting both positive and negative findings, and perhaps trying to synthesize the results -- how the present work differs from previous studies in ways that might explain what was observed.

5. In the abstract, the authors claim that "Together, our data suggest that CCL5 might be a novel, stable and practicable stress level biomarker." I would eliminate all claims about biomarkers. There is too much overlap in CCL5 levels between stressed participants and unstressed control participants to serve as a biomarker.

6. fMRI data: I would move these results to the supplement or maybe cut them. The methods aren't very well described (they refer to seven seeds but do not define them) but my understanding is that the main analysis in Fig. 6 compared functional connectivity between the red nucleus and every other voxel in the brain in a relatively small sample (~35-40 subjects per group) and then used a cluster threshold to correct for multiple comparisons and identify significant differences in the stressed group. This approach is well known for yielding false positives (<https://www.pnas.org/doi/10.1073/pnas.1602413113>), and there are a lot of red flags here: the samples are relatively small for performing tens of thousands of tests, and the reported effects are very small as well, both in magnitude and spatial extent. Further complicating things, subcortical / midbrain signals are notoriously noisy in resting state fMRI datasets: e.g. <https://pubmed.ncbi.nlm.nih.gov/31836321/>. One option would be to evaluate the stability of these results in bootstrapped subsamples of the data. If the results are stable, they might be more persuasive. But a better option might just be to remove the fMRI data or at least move it to the

supplement and present it as a preliminary finding with a lot of caveats.

Minor issues:

- Fig. S1A: "Our results showed that several inflammatory cytokines, such as IL2 and TGF α , were negatively correlated with the stress level (Fig. S1a)." This analysis needs to be corrected for multiple testing. This claim probably isn't true after making these corrections, since the correlations in question are very weak and just barely exceed a threshold for significance without correction. Same goes for the claims about CCL5 correlating with PCL5, PHQ9, and SRQ scores -- the p values should be corrected. Since those p values are smaller, I would guess they might actually be significant.
- The images in Fig. S6a aren't great in terms of quality. The insets in S6b help to some degree but they're still hard to see. Are better images available? Are the six regions shown in S6b the only regions that showed any labeling? That's very hard to appreciate from Fig. S6a because of the high background. The six named regions appear to have some of the lowest levels of red fluorescence, presumably due to background. But this raises questions about false negatives due to high background. Overall, this is a relatively minor issue since there are any other experiments in this paper implicating RN.
- Minor: there are two panels labeled "r" in Figure 2
- Minor: for the photometry experiments, the authors note that "We implanted an optic fiber (200 μ m optical density, 0.37 numerical aperture) above the unilateral RN to record activity changes of the RN neurons." The scale bar in Fig. 4b makes it look like the diameter of the optical fiber is 100 μ m. Is the scale bar a typo?
- For the RNA seq data in Figure 2, the authors note that "While there are no changes in expression of interferon regulatory factor 1 (IRF1) in stressed CLN, suggesting that RS led to translation dysfunction in CLN, which is consistent with the results of RNA-seq." Why do the results suggest *dysfunction* in translation? Why is this not a normal, healthy, adaptive response, just one component of activating the immune system? How do we know it's evidence of "dysfunction"?
- The "stressed" group of humans in Figure 1 were psychiatrists who traveled to Wuhan to perform frontline work. The effects on CCL5 levels in this group are attributed to psychosocial stress, but it's also possible that they were exposed to COVID or some other infection. Do the authors have any data to speak to this? If not, they could just comment on it in the Discussion.

Reviewer #4:

Remarks to the Author:

In their manuscript, Shi et al. identify CCL5 as a potential marker of stress in stressed individuals. They further support this finding in rats that underwent restraint stress, revealing cervical LN as the main anatomical origin for CCL5 elevation and its underlying cellular mechanisms. Using retrograde neuronal tracing, they identify specific glutamatergic neurons in the Red Nucleus (RN) that innervate the CLNs. They characterize their electrophysiological properties during stress, dissect a neural circuit in the M1 that regulates these neurons' activity and elegantly demonstrate their functional significance to stress-related behavior and CCL5 elevation via chemogenetic tools. Finally, using fMRI, the authors show reduced functional connectivity in RN of stressed human individuals, consistent with the results they showed in rats.

The findings are novel, interesting and potentially impactful, and the authors are providing overall solid experimental evidence for their claims. Yet, there are several issues that I think should be addressed in order to fully support the authors' conclusions.

Major comments:

1. None of the controls (both human and rats) went through any kind of novel non-stressful event, not entirely excluding the possibility that the RN neurons identified in this study are linked to a learning or novel experience process. Please refer to this limitation in the text, or preferentially add such control group, showing this doesn't elevate CCL5 in the serums/CLNs and CLN CD8 T cells (e.g., novel object or a motor learning paradigm).
2. Did all human participants undergo COVID19 PCR/antigen testing? As the "stressed" group was exposed to a more infected area, I would recommend making sure and showing that the differences between groups were not a result of a prior/recent infection.
3. It would be beneficial if the authors show that upon inhibition of RN glutamatergic neurons, there is an increase in noradrenaline (NA) in the CLN (e.g., via Elisa, IHC). It may further strengthen the link between the stress-induced neuronal response, and CLN immune regulation.
4. I would suggest to avoid using the word "Dysfunctions" in the title to describe the reduced activity of RN in times of stress. An editing suggestion for the title: Stress-induced Red Nucleus Attenuation induces Anxiety-like Behavior and Lymph Node CCL5 secretion.
5. Although there is a simultaneous rise in CCL5 and CD8 T cells in CLN of stressed rats, it still doesn't entirely prove that these cells are the subgroup that secretes CCL5. It will be valuable if the authors intracellularly stain CD8 T cells to CCL5.
6. As the authors mentioned, the fact that female rats did not show signs of stress and no cortisol level increase, but still presented elevated CCL5 levels, weakens the claim that stress is related to the chemokine dynamics. Are there any stress tests that are known to be more suitable for female rats other than the open field or elevated maze, or professional literature evidence for poor outcomes of stress test in female rodents?
7. Overall, poorly written (also making some of the experimental explanations not entirely clear): Use of both present and past tenses interchangeably, spelling and grammar mistakes, and questionable choices of words ("besides", "fortunately", "so on", "dramatic"). The discussion is also very repetitive and unorganized. Would highly recommend the text goes through professional proofreading (including Legends and Materials and Methods section).
8. Figure 6: Are the brain regions that are presented in Fig. 6m, the only areas that showed significant differences between the groups? I would expect additional regions to be involved during stress (e.g., hypothalamus, amygdala, prefrontal cortex).
 - fMRI data is not in the scope of my expertise, so couldn't fully assess these results.
9. As the authors stated, it is surprising that cortisol levels were diminished in individuals from the stressed group, rendering the stress questionnaire the only measure by which they confirmed these individuals' elevated stress levels. The authors should address this limitation in the text, and- if possible- measure cortisol levels of these participants at an earlier time point (closer than 1 month to the stressful event).

Minor comments:

1. There are many figure reference mistakes throughout the text, figures and legends. Some of these are listed here:
 - a. Line 150: Fig.S2a - S3a?
 - b. Line 152: Fig. S2d - S3d?
 - c. Lines 211, 213, 214, 216 - Fig.s-v? (Fig.2- "r" appears twice in the figure)
 - d. Line 261: Fig. 3e - 3f?
 - e. Line 264: Fig. 3f - 3e?
 - f. Lines 274-5: Fig. ref?
 - g. Line 279: Fig.S7a?
 - h. Line 302: Fig. 5j - 5g?
 - i. Line 325: Fig. S5a-b - S6a-b?
 - j. Line 599: wrong Table ref.
 - k. Line 680: Table S5 - S4?
2. The figures' panels are not in the same order as their presentation in the text, causing quite a confusion throughout the reading (one example: Fig.1g). Recommend changing the order of the plots to the order they appear in the text.
3. The authors should clarify undefined abbreviations: For example, FC (line 366), EPR (line 531), PND (line 565, where it first appears and not later in the text).
4. The number of human participants is inconsistent throughout the text, figures and tables: Table 1- 88 (n=47, 41), Methods fMRI- 82 (n=44, 39)?
5. How did the authors acclimatized the mice before behavioral tests? The authors should depict

the methodological details.

6. There is no mentioning of CD45RC in the text and Fig.1q-r legend. The authors should state why they measured this marker and what its increase indicates.
7. Line 50: What is the "inflammatory system"? Recommend changing to "inflammatory response".
8. Line 68: Add a reference to the statement.
9. Line 147: add "Circadian" rhythm.
10. Line 226: "The inhibition effects were confirmed by immunohistochemistry." Add Fig. ref.
11. Line 277: Add a reference.
12. Line 290: Why "conditionally"?
13. Fig. 2:
 - a. 2a: add days units to the scheme.
 - b. 2f: Live - Liver?
 - c. 2k: problematic to state that there is a difference between groups while it is not significant... The authors can write "a trend toward higher..."
 - d. 2l: it seems there is a difference in CCL5 also in the sham group, before, during and after stress. Is't it significant? This suggests that there is another possible source of CCL5, bone marrow (BM) for example, which is known to also be innervated and affected by the sympathetic nervous system (and therefore by stress responses). Would be beneficial if the authors will measure levels of CCL5 in BM as well.
14. Fig. 3:
 - a. 3a: Glass pipette instead of electrode?
 - b. 3b: What do the arrows signify?
 - c. The title of the figure is not clear.
15. Fig. 5:
 - a. 5i: Missing units for Y axis
16. Fig.6: no 6l in legends, wrong 6m, n.
17. In Fig. S7, the experimental design and timeline is not clear. The authors should add an experimental scheme.

Point-by-point responses to the reviewers' comments:

Reviewer #1:

Overall, this is a potentially interesting paper describing a novel circuit from motor cortex (M1) to red nucleus (RN) to cervical lymph nodes (CLN) that controls anxiety behavior, possibly through CLN production of CCL5. While the story is interesting and potentially important, there are several significant weaknesses that should be addressed in a revision. Most critically, the authors need to provide further evidence to support the circuit in question and provide casual evidence to link activation of M1-RN-CLN to downstream effects of CCL5 that promote anxiety.

RESPONSE: We thank the reviewer for his or her positive comments about the importance of our work. Below, we address the concerns of the reviewer point by point.

- 1. Chemogenetic strategies might not be appropriate for defining this monosynaptic circuits. Do either M1 neurons projecting to RN or RN neurons projecting downstream to CLN contain bifurcating axons targeting multiple downstream brain structures. If true, as is the case with most glutamatergic projections neurons, the authors would need to repeat these studies using optogenetic terminal stimulation studies to target monosynaptic connections between brain regions.*

RESPONSE: Thank you for your questions. In our research, to specifically inhibit RN postsynaptic neurons, the monosynaptic anterograde transport virus AAV1-hSyn-Cre was bilaterally delivered to the M1, and AAV-CaMKII-DIO-hM4D(Gi)-mCherry was bilaterally delivered to the RN. Based on the principle of the Cre-DIO system, only glutamatergic neurons with Cre projecting from the M1 in the RN can be inhibited by CNO. Other glutamatergic neurons without cre in the RN could not be inhibited by CNO. Thus, only glutamatergic neurons projecting from M1 in the RN were inhibited by CNO. Although M1 neurons projecting to the RN contain bifurcating axons, other brain regions cannot be inhibited by CNO without DIO.

To explore the physiological properties of the M1-RN pathway more specifically, we delivered AAV-CaMKII-hChR2(H134R)-mCherry to the M1. Rats were implanted bilaterally with an optical cannula in the RN immediately after AAV injection. Three weeks after surgery, the rats were subjected to RS for 3 days, and their anxiety-like behavior was measured 24 h after the last restraint. During the behavior tests, the optical cannula was connected to the optical patch cable, and 10 Hz pulses of blue light illumination (5-ms duration) was delivered from a blue laser (470 nm, Inper Inc. China) to the optical cannula through the optical patch cable and optical rotary joint for 10 min. Behavioral testing was performed during stimulation. Our results were consistent with the results of chemogenetic strategies, showing that stimulating the M1 fibers in the RN could prompt the rat to enter the central zone in the OFT and open arms in the EPM (Fig.R1).

Fig. R1 Activation of M1-RN alleviated the increased anxiety-like behavior induced by restraint stress, while inhibition of M1-RN induced anxiety-like behavior. a, Schematic illustrating the injection of Chr2-carrying AAV into the M1 and cannula implantation with optic fibers onto the RN. **b,** Time spent in the central zone in the OFT (two-tailed unpaired t test, $t = 2.968$, $p = 0.0179$). **c-d,** Open arm ratio (two-tailed

unpaired t test, $t = 3.305$, $p = 0.0108$) and open arm frequency (two-tailed unpaired t test, $t = 2.407$, $p = 0.0427$) in the EPM. e, Schematic illustrating the injection of eNpHR3.0-carrying AAV into the M1 and cannula implantation with optic fibers into the RN. f, Time spent in the central zone in the OFT (two-tailed unpaired t test, $t = 3.376$, $p = 0.0097$). g-h, Open arm ratio (two-tailed unpaired t test, $t = 2.964$, $p = 0.0180$) and open arm frequency (two-tailed unpaired t test, $t = 0.3409$, $p = 0.7420$) in EPM. M1, primary motor cortex. Data are presented as the mean \pm SEM. * $P < 0.05$, ** $P < 0.01$, *** $P < 0.001$.

2. *Is the connection between RN and lymph nodes mono or polysynaptic? If the latter, please provide a more comprehensive tracing map to show the intermediary connections between RN and lymph nodes.*

RESPONSE: We thank the reviewer for raising this very important point. The connection between the RN and lymph nodes is polysynaptic. In response, we have provided a more comprehensive tracing map and carried out new sets of PRV tracing experiments on different days. We found that over 120 h, PRV ascended from the cervical lymph nodes to the spinal cord, brain stem, hypothalamus, RN and M1 (Fig. R2). However, over 48 h, PRV ascended from the cervical lymph nodes up into the spinal cord only. Over 72 h, PRV ascended from the cervical lymph nodes up into the spinal cord and RN (Fig. R3). The results are presented in Fig. S7 and Fig. S8 of the revised manuscript. (Lines 1443-1452)

Fig. R2 Retrograde PRV tracing from CLNs (over 120 h). a-d, Selected CNS regions that were labeled by PRV. LC, locus coeruleus; SubCV, subcoeruleus nucleus, ventral part; SubCD, subcoeruleus nucleus, dorsal part; RMg, raphe magnus nucleus; NTS, nucleus tractus solitarius; PVN, paraventricular thalamic nucleus; M1, primary motor cortex.

Fig. R3 Time-dependent infection of different brain areas after PRV injection into CLNs. Representative images showing PRV-infected neurons (red) in different brain areas at different time points after viral injection. Left: 3 days; middle 4 days; right: 5 days. These experiments were performed in 3 male rats for each group..

3. I guess the most important question is how CCL5 being released from CLN is regulating anxiety? Much of the work centers on establishing the circuit from RN to lymph nodes. Indeed, chemokinetic manipulations of RN shows it is necessary and sufficient to induce anxiety, but there are no experiment that prove a causal relationship between RN activation of CCL5 and anxiety. It's quite possible the anxiety phenotype initiated by chemogenetic manipulation of RN is due to its actions on downstream neural circuitry and is unrelated to the RN-CVN. This is particularly important given that circuit specific optogenetic approaches (see point 1) were not used.

RESPONSE: We thank the reviewer for this comment. Our results showed that RN attenuation could induce anxiety-like behavior and lymph node CCL5 secretion. As the reviewer pointed out, there are no experiments that prove a causal relationship between RN activation of CCL5 and anxiety. In fact, we did not point out that CCL5 can promote anxiety. Both elevated CCL5 levels and anxiety result from RN attenuation. CCL5 secretion, as a result of brain regulation, responds to anxiety levels.

Since the connection between the RN and lymph nodes is polysynaptic, it is difficult for us to perform optogenetic experiments to explore the physiological properties of the RN-CLN pathway. To verify the relationship between the RN and CCL5 and anxiety, as suggested by the reviewer in question 5, we administered a monoclonal antibody against CCL5 in the periphery in rats with the inhibition of RN neurons. We investigated whether systemic blockade of CCL5 influenced RN inhibition-induced anxiety using a specific neutralizing antibody. The open field test revealed that anti-CCL5 treatment had no effect on the locomotion ability of rats. The time spent in the central zone in the OFT and the open arm ratio in the EPM test were significantly increased in anti-CCL5 rats compared to IgG rats, suggesting that systemic neutralization of CCL5 reversed the anxiety induced by RN inhibition (Fig. R4).

Fig. R4 CCL5 neutralization reverses anxiety induced by RN inhibition. a, Distance moved in OFT (two-tailed unpaired t test, $t = 0.1369$, $P = 0.8932$, $n = 8$ rats per group). b, The time spent in the central zone in the OFT was significantly increased in anti-CCL5

rats compared to controls (two-tailed unpaired t test, $t = 2.176$, $P = 0.0486$, $n = 8$ rats per). c-d, Both the open arm ratio and frequency were increased in anti-CCL5 rats compared to IgG rats (two-tailed unpaired t test, $t = 3.129$; 2.511 , $P = 0.0087$; 0.0273 , $n = 8$; 8 rats per). Data are presented as the mean \pm SEM. * $P < 0.05$, ** $P < 0.01$, *** $P < 0.001$.

To explore the physiological properties of the M1-RN pathway more specifically, we delivered AAV-CaMKII-hChR2(H134R)-mCherry to the M1. Rats were implanted bilaterally with an optical cannula in the RN immediately after AAV injection. Our results were consistent with the results of chemogenetic strategies, showing that stimulating the M1 fibers in the RN could prompt the rat to enter the central zone in the OFT and the open arms in the EPM test (Fig. R1).

4. *Related to point 3 above, what and where is CCL5 acting upon to promote anxiety? Does it feed back up and enter the brain to control neural activity in anxiety-provoking brain regions?*

RESPONSE: We thank the reviewer for this question. As shown in response 3, our results showed that the time spent in the central zone in the OFT and the open arm ratio in the EPM test were significantly increased in anti-CCL5 rats compared to IgG rats, suggesting that systemic neutralization of CCL5 reversed the anxiety induced by RN inhibition (Fig. R4). It is indeed an important question worthy of further investigation. We speculate that CCL5 mainly affects anxiety through two pathways. First, the overproduced CCL5 circulates to the brain through the blood, affecting the permeability of the blood–brain barrier, causing abnormal activity of neurons in the brain and leading to anxiety. Second, the excessive production of CCL5 causes it to negatively feed back to the brain through the nerves, thereby inhibiting the activity of RN neurons and leading to anxiety.

To verify these two hypotheses, we first detected the expression of CCL5 in the brain and found that the expression of CCL5 in the brain did not significantly increase after stress, indicating that the CCL5 produced in the periphery does not enter the brain.

Noradrenergic fibers were identified in rat cervical lymph nodes by fluorescence histochemistry and confirmed by radioenzymatic determination of norepinephrine (NA)¹. Norepinephrine (NA) is one of the primary catecholamines of the sympathetic nervous system released during a stress response and plays an important role in modulating immune function². We detected norepinephrine (NA) in CLNs. The results showed that NA was significantly increased both in stressed rats and in RN-inhibited rats (Fig. R5). NA preferentially modulates memory CD8 T-cell function, inducing inflammatory cytokine production³. The increase in CCL5 may be fed back to the brain through noradrenergic fibers, thereby affecting anxiety-like behavior. However, this is a very complicated adjustment process, and more in-depth experiments are needed to explore the mechanisms. We discussed this issue in the discussion (Lines 520-533).

Fig. R5 NA increased in both stressed rats and RN inhibition rats. a, The concentration of NA in RS rats was significantly increased compared to that in controls (two-tailed unpaired t test, $t = 2.668$, $p = 0.0257$). b-c, Both the ablation (two-tailed unpaired t test, $t = 8.511$, $p = 0.0001$) and inhibition (two-tailed unpaired t test, $t = 14.83$, $p < 0.0001$) of RN neurons led to an increase in the NA concentration in CLNs. Data are presented as the mean \pm SEM. * $P < 0.05$, ** $P < 0.01$, *** $P < 0.001$.

5. *Data in figure 5 is interesting and shows that deleting or inhibiting vGlut cells in the RN increases anxiety and CCL5. However, it's unclear if the anxiety has anything to do with the increase in CCL5. Can you prevent the anxiety effects elicited by inhibition or ablation of RN neurons by co-administering a monoclonal antibody against CCL5 in the periphery?*

RESPONSE: We agree that coadministering a monoclonal antibody against CCL5 in the periphery would be useful to understand the connection between anxiety and CCL5 levels. To verify the relationship between RN and CCL5 and anxiety, as suggested by the reviewer, we administered a monoclonal antibody against CCL5 in the periphery in rats with the inhibition of RN neurons.

Methods:

Systemic neutralization of CCL5. Monoclonal antibodies recognizing CCL5 or control IgG were administered intraperitoneally (250 μ g/rat in sterile PBS) to rats injected with AAV-CaMKII α -hM4D(Gi)-mCherry in the RN immediately after CNO injection. (**Lines 876-879**)

Results:

We investigated whether systemic blockade of CCL5 influenced RN inhibition-induced anxiety using a specific neutralizing antibody. The open field test revealed that anti-CCL5 treatment had no effect on the locomotion ability of rats. The time spent in the

central zone in the OFT and the open arm ratio in the EPM test were significantly increased in anti-CCL5 rats compared to IgG rats, suggesting that systemic neutralization of CCL5 reversed the anxiety induced by RN inhibition. (Fig. R4 & Fig. S10) (Lines 339-345 & Lines 1464-1472)

6. In Fig 6, the authors show that chemogenetic inhibition of M1 neurons projecting to RN regulates their activity and anxiety behavior. As mentioned above, this is not necessarily circuit specific if those cells in the M1 bifurcate and project elsewhere in the brain. Further, it's not established that activation of this pathway ultimately affects CCL5 to control anxiety.

RESPONSE: We thank the reviewer for this important point, which we respond to in detail in question 1. As described in the answer to question 1, we further delivered AAV-CaMKII-hChR2(H134R)-mCherry into the M1 and bilaterally implanted it with an optical cannula in the RN immediately after AAV injection in rats. The optogenetic experiment confirmed that the M1-RN circuit was responsible for anxiety.

Our results indicated that the activation of this pathway resulted in a decrease in CCL5, while the inhibition of the M1-RN circuit led to an increase in CCL5 (Fig. R1). To further investigate the relationship among the M1-RN circuit, CCL5 and anxiety, we coadministered a monoclonal antibody against CCL5 in the periphery in M1-RN-inhibited rats. Both the open field test and elevated plus maze test showed that anti-CCL5 reversed the anxiety induced by M1-RN inhibition (Fig. R6).

Fig. R6 CCL5 neutralization reverses anxiety induced by M1-RN inhibition. a, Distance moved in OFT (two-tailed unpaired t test, $t = 1.487$, $P = 0.1591$, $n = 8$ rats per group). b, The time spent in the central zone in the OFT was significantly increased in anti-CCL5 rats compared to controls (two-tailed unpaired t test, $t = 2.913$, $P = 0.0113$, $n = 8$ rats per). c-d, Both the open arm ratio and frequency were increased in anti-CCL5 rats compared to IgG rats (two-tailed unpaired t test, $t = 2.456$; 2.228 , $P = 0.0302$;

0.0457, n = 8; 8 rats per). Data are presented as the mean \pm SEM. *P < 0.05, **P < 0.01, ***P < 0.001.

7. *For cytokine and chemokine analysis, some of the values are quite low (ie., <2-3pg/ml). Was this in the range of detection of the ELISA kit used?*

RESPONSE: Thank you for your questions. Plasma cytokine and chemokine levels were measured using ProcartaPlex™ Human Inflammation Panel 10 Plex (Invitrogen) in a Luminex 200 multiplexing instrument (Millipore Sigma). This system allowed us to generate quantitative measurements for 10 different cytokines and chemokines, including MCP-1, IL-1 β , IL-4, IL-10, TNF- α , IFN- γ , IL-2, IL-6, TGF- α and CCL5. A CCL5 (RANTES) ELISA kit (ab174446, human RANTES ELISA Kit (CCL5), Abcam) was used to confirm the results of Luminex and measure the CCL5 concentration in follow-ups. The range of IL-4 detection was 1.0 pg/ml.

Reviewer #2

In this study, Shi et al identify CCL5 as an increased cytokine in stressed patients during the acute phase. In a rat model of restrain stress (RS), they demonstrate the increased expression of CCL5 in cLN and by RNAseq identify translation as potential mechanism involved in this increased expression. Then, the authors try to demonstrate that translation is involved in RS dependent induction of CCL5 in the cervical LN. Additionally, the authors show that RS inhibits the excitability of Red Nucleus glutamatergic neurons leading to anxiety-like behaviour. Additionally, they showed that reduced RN firing induces increased CCL5 expression in the cervical LN, and conversely, increased RN neurons activation reduces anxiety-like behaviour as well CCL5 expression.

While the neuroscience part is overall convincing, the link to immunology should be strengthened. Furthermore, the paper should be thoroughly restructured to make things clearer. While the role of CCL5 in stress and anxiety has been object of study in the past, the discovery of a cortex-midbrain-LN neuronal circuit is intriguing and deserves attention. The brain circuit is analysed in a detailed and thorough manner, however, as acknowledged by the authors, there is no mention of the circuit connecting the brain with cervical LN (PNS).

RESPONSE: We thank the reviewer for the positive evaluation of our work. We have substantially revised the manuscript, added new experiments and added additional required explanations. The changes are described in the cover letter and below.

1. *CCL5 has been previously reported to be altered in subjects displaying anxiety (<https://doi.org/10.1016/j.pharep.2014.08.006>), chronic stress (<https://doi.org/10.1016/j.ynstr.2018.02.002>), PTSD*

*(<https://doi.org/10.1002/da.20564>), depression during pregnancy
(<https://doi.org/10.1186/s12884-021-04225-2>) as well as animal models
(<https://doi.org/10.1016/j.biopsych.2021.02.765>;
<https://doi.org/10.1016/j.celrep.2021.108979>) raising concerns about the novelty
of the main discovery.*

RESPONSE: We thank the reviewer for this comment. We completely agree that CCL5 is altered in different psychiatric disorders. We have added these studies to our discussion (**Lines 439-449**). However, the core concerns of our research are not CCL5 itself but the neural circuit and the peripheral-central relationship mechanisms of CCL5. We focused on the chemokine CCL5 based on our clinical findings in stressed individuals and identified CLNs as the main immune organs that produce CCL5 in response to stress. We show that the manipulation of RN glutamatergic neuron activities can alter anxious behaviors and inflammatory responses. Our research deepened the understanding of the CCL5 mechanism of stress-induced anxiety.

- 2. It starts with the title, but throughout the manuscript the grammar needs a lot of improvement. One of many examples is line 65 'litter' instead of 'little'.*

RESPONSE: Thank you for pointing this out, and we apologize for the mistakes in the manuscript. We have revised the grammar in the manuscript.

- 3. The current formatting of the paper and explanations are thoroughly lacking clarity. E.g. Figure 2r exists twice, whereas Fig 2v is missing. This is not acceptable and is a major concern for the overall quality of the data. How can the underlying data be trusted when the presentation lacks these basic elements?*

RESPONSE: We apologize for the mistake in the figure. We have revised these mistakes in Figure 2. We have also revised the manuscript carefully and thoroughly.

- 4. It needs to be explained what precisely is the stress that human subjects were exposed to. Why are glucocorticoids lower in stressed subjects?*

RESPONSE: Thank you for your suggestions. We have added explanations of stress in human subjects (**Lines 91-93**). To detect the immune indicators in stressed people, we recruited psychiatrists who worked on the front line during the COVID-19 pandemic in Wuhan between February 21 and March 31, 2020. The acute phase interview was carried out 40 days after they left the stressful environment. Blood collection was also performed 40 days after they left the stressful environment. Glucocorticoids are increased during stress, especially in the early phase. Unfortunately, we did not collect blood during or immediately after the subjects left the stressful environment. According to the changes in glucocorticoids after stress, we speculate that glucocorticoids increase first and

compensatorily decrease after stress. Thus, forty days after subjects left the stressful environment, glucocorticoids were lower in stressed subjects than in controls because of a compensatory decrease. Six months later, the glucocorticoid levels returned to normal (Fig. 1).

5. *The questionnaires need to be explained in much more detail, not even the acronyms are spelled out, nor what they actually measure. What do the scores mean, how are they interpreted?*

RESPONSE: We thank the reviewer for this suggestion. We have added the questionnaire information to the manuscript (**Lines 575-594**).

“The Generalized Anxiety Disorder 7-item (GAD-7), the Patient Health Questionnaire-9 (PHQ-9), the Self Reporting Questionnaire 20-Item (SRQ-20) and PTSD Checklist for DSM-5 (PCL-5) were used to evaluate the mental disorder symptoms of all participants. The Generalized Anxiety Disorder 7-item (GAD-7) is an easy-to-use tool for initial screening for generalized anxiety disorder. When screening for anxiety disorders, a score of 8 or greater represents a reasonable cut-point for identifying probable cases of generalized anxiety disorder. The following cutoffs indicate the severity of anxiety: score 0-4, minimal anxiety; score 5-9, mild anxiety; score 10-14, moderate anxiety; and score greater than 15, severe anxiety^{4, 5}. The Patient Health Questionnaire-9 (PHQ-9) is a multipurpose instrument for screening, diagnosing, monitoring and measuring the severity of depression. Total scores of 5, 10, 15, and 20 represent cut points for mild, moderate, moderately severe and severe depression, respectively^{6, 7}. The Self Reporting Questionnaire 20-Item (SRQ-20) can be used to detect nonspecific psychological distress; subscales include depression/anxiety, somatic symptoms, reduced vital energy and depressive thoughts. Items are scored as 0 (symptoms absent) or 1 (symptoms present). Scores range from 0 to 20, with scores >10 indicating mental distress. Responses are yes or no⁸. The PTSD Checklist for DSM-5 (PCL-5) is a 20-item self-report measure that assesses the 20 DSM-5 symptoms of PTSD. The characteristics of a respondent's setting should be considered when using PCL-5 severity scores to make a provisional diagnosis^{9, 10}. The goal of assessment should also be considered. A lower cut-point score should be considered when screening or when it is desirable to maximize the detection of possible cases. A higher cut-point score should be considered when attempting to make a provisional diagnosis or to minimize false-positives.”

6. *It is completely unclear what makes the cervical LN special that it is the predominant source of CCL5 compared to other LNs. This is exciting, indicating local neural control of LN functions but the paper lacks any explanation. The tracing in Figure 3a is very nice labeling, however, do other LNs not project to the RN?*

RESPONSE: We thank the reviewer for this comment. Cervical lymph nodes are those nearest the front of the neck. Previous studies identified noradrenergic fibers in rat cervical lymph nodes by fluorescence histochemistry, and this finding was confirmed by radioenzymatic determination of norepinephrine¹¹. Moreover, Mehmet et al. detected direct connections between lymphatic fluid channels along the cranial nerves and vascular structures and the cervical lymph nodes¹². All these studies indicated that cervical LNs were more closely related to the brain than other LNs. We found that over 120 h, PRV ascended from the cervical lymph nodes to the spinal cord, brain stem, hypothalamus, RN and M1 (Fig. R2). However, over 48 h, PRV ascended from the cervical lymph nodes up into the spinal cord only. Over 72 h, PRV ascended from the cervical lymph nodes up into the spinal cord and RN (Fig. R3).

We also traced inguinal lymph nodes and mesenteric lymph nodes using PRV (Fig. R7). Over 72 h, PRV ascended from the cervical lymph nodes up into the spinal cord. After 6 days, we also observed a cluster of mRFP+ neurons bilaterally in the RN. Although cervical LNs, inguinal LNs and mesenteric LNs all project from the RN, we did not observe an increase in CCL5 in stressed rats and rats in which RN glutamatergic neurons were inhibited in inguinal LNs and mesenteric LNs. This might be because there are direct connections between nerves and cervical LNs.

a **Inguinal lymph nodes**

b **Mesenteric lymph nodes**

Figure R7 Identification of a cluster of RN neurons upstream of inguinal LNs (a) and mesenteric LNs (b).

7. *Figure 2: Why are female rats not affected? In humans, both sexes are equally affected, correct? Why is the paper making a point of this?*

RESPONSE: We thank the reviewer for this comment. Our research shows that in rats exposed to restraint stress, females respond better than males, which is consistent with previous studies¹³. The idea that females are more resilient than males in responding to stress is a popular view^{13, 14, 15}. The resilience of female rats to stress may be because of the protective effect of estrogen¹⁵. The different performance of females and males suggested that males should be selected for subsequent mechanistic studies. Thus, we listed these data to explain why subsequent mechanistic studies have mainly focused on male rats. (**Lines 147-148**)

In our human study, we recruited 47 psychiatrists who worked on the front line during the COVID-19 pandemic in Wuhan (35 male and 12 female) as stressed individuals and 41

psychiatrists as controls in our study. Sex, age, marriage status and education years were not different between these two groups (Table 1). In our human study, GAD-7, PHQ-9, SRQ-20 and PCL-5 scores were not different between males and females. In our animal study, female rats showed better resilience than males. There are several reasons. First, in our human study, the number of females was lower than that of males. Second, the subjects in our research were psychiatrists who received systematic psychological training. In addition, cultural factors have an important influence on psychological conditions. The above factors led female and male subjects to be equally affected.

8. *The order of panels in the Figures and the order of Figures mentioned in the text is extremely confusing. For Figure S2a the 24h CCL5 data are shown later. Also, it is not clear why 24h experiments were performed.*

RESPONSE: We apologize for the mistake in the order of panels in the figures. We have updated the order of panels in the figures and the order of figures mentioned in the text.

To determine the effect of circadian rhythms on CCL5 and corticosterone levels, control rats without restraint stress underwent continual blood sampling to establish the 24-h CCL5 and corticosterone profiles. For CCL5 levels, no effect of circadian rhythms in control rats not exposed to restraint stress was observed, indicating that CCL5 concentration was unaffected by circadian rhythms. This finding suggests that CCL5 is readily available and may be a better stress biomarker than corticosterone. (Lines 156-157).

9. *Specifically, Figure panels need to be in chronological order. E.g. the authors discuss FigS2a and d, and then later S2b,c. This is difficult to follow. What is the difference between Fig. S3b,e, and Fig. S2f,h (the latter is female, the former is male?) This needs to be mentioned, only females are highlighted. Figure S8 is mentioned after S9. There are additional examples.*

RESPONSE: We apologize for the mistake in the order of panels in the figures. We have updated the order of panels in the figures and the order of figures mentioned in the text.

10. *In the rat experiments, are other chemokines altered apart from CCL5, different from the human scenario? A link needs to be made between the rat and the human scenario. Indeed, there seem to be additional chemokines upregulated in Fig. S5a. It is not clear why in Fig S5 CCL5 is not upregulated.*

CCL5 levels return to baseline in rats within 2 days, whereas in the human scenario this is still seen after 6 months. How do these stress tests (acute/chronic) compare with respect to CCL5 levels?

RESPONSE: We thank the reviewer for this point. We also measured the levels of other chemokines and cytokines, such as MCP-1, IL-1 β , IL-2, IL-4, IL-6, TGF- α and TNF- α , in the serum of stressed rats. No difference was found between stressed rats and controls. Only CCL5 was increased significantly in serum in both humans and rats. In Fig. S5a, the expression of CCL5 was measured by RT-PCR in cervical LNs. The results showed that restraint stress also increased the concentrations of cytokines and chemokines, such as IFN α , IL2, IL6, and CCL2, in cervical LNs (Fig. S5a). CCL5 mRNA expression was not upregulated in the cervical LNs of stressed rats. However, we discovered a significant increase in CCL5 levels using ELISA. This may be because stress activates the translation process, resulting in a decrease in mRNA levels.

The concentration of CCL5 were slightly decreased on the third day, which might be because rats had adapted to the stress. But there was still an upward trend in CCL5 levels in stressed rats. CCL5 levels were still high in rats after early life stress and chronic mild unpredictable stress (Fig. S4e & S4j). In humans, CCL5 levels were higher in stressed individuals compared to controls after 1 month and returned to baseline after 6 months (Fig. 1p).

11. Figure 3i is very nice, there still seems to be an increase though, just with overall lower CCL5 levels

RESPONSE: Sorry, we are confused by this question. Fig.3i is the result of voltage threshold.

12. Within the cervical LN, which cells are producing the effect? The authors mention that circulating CD8 T cells are not different, does that mean the effect of cervical LN CD8 T cells is due to tissue residency? This indicates that the effect is not cell autonomous, how does the cervical LN then imprint a CCL5 expression phenotype, and on which cells? What happens after CD8/CD4 T cell depletion?

RESPONSE: We thank the reviewer for this comment. It has long been known that memory cells respond more rapidly to antigen¹⁶, and mRNA for CCL5, which is virtually absent from naïve cells, was over 30-fold higher in both populations of memory phenotype T cells¹⁶. Immunological memory is an important immune mechanism by which the body resists the reinvasion of pathogens. After the body's primary immune response, heterogeneous memory CD8 T cells are generated, including effector memory T (TEM) cells circulating in blood tissues, central memory T (TCM) cells circulating in secondary lymphoid organs, and long-term resident tissue-resident T (TRM) cells circulating in tissues. The body's primary immune response may be to fighting for breast milk or fighting and so on.

Compared to control rats, the frequencies and numbers of CLN memory CD8⁺ T cells were significantly increased in stressed rats (Fig. 2q-r), suggesting that CD8⁺ T cells have

a broad impact on physical stress-induced anxiety-like behavior. Norepinephrine preferentially modulates memory CD8 T-cell function, inducing inflammatory cytokine production³. We measured norepinephrine (NA) in CLNs. The results showed that NA was significantly increased both in stressed rats and in RN inhibition rats (Fig. R4). A previous study showed that noradrenaline induces adrenergic receptor signaling to reduce lymph node blood flow¹⁷.

A previous study showed that adoptive transfer of CD4⁺ T cells into Rag2^{-/-} mice did not change cytokine levels in response to stress, while CD8⁺ T cells resulted in an increase in TNF- α , IL-6 and IFN- γ in stressed Rag2^{-/-} mice. Moreover, the depletion of CD8⁺ T cells in WT mice abolished these cytokine responses to stress. Corticosterone and behavioral stress responsiveness were impaired in Rag2^{-/-} mice reconstituted with CD8⁺ T cells¹⁸. These studies suggested that behaviors could be impaired by CD8⁺ T cells.

13. Figure 3g: CCL5 levels in LN are not convincing. Chemokine levels are notoriously difficult to quantify by imaging.

RESPONSE: Thank you for pointing this out. We quantified CCL5 levels in CLNs by ELISA and RT-PCR. To show the changes in CCL5 levels more clearly, we used images for semiquantification.

14. Fig 3: only 1 time point after PRV infection is not ideal as it is not possible to assess sequential jumps and areas not related to cervical LN (injection sites) can potentially be targeted. RFP in the RN seems quite specific, however, close up pictures of other brain areas would be appreciated. Fig S6: following up on the previous comment, quality is low and specific staining cannot be appreciated. This is potentially an issue as it is not possible to clearly discriminate areas with specific mRFP staining.

RESPONSE: Thank you for pointing this out. In response, we have provided a more comprehensive tracing map and carried out new sets of PRV tracing experiments on different days. We found that over 120 h, PRV ascended from the cervical lymph nodes to the spinal cord, brain stem, hypothalamus, RN and M1 (Fig. R2). However, over 48 h, PRV ascended from the cervical lymph nodes to the spinal cord only. Over 72 h, PRV ascended from the cervical lymph nodes to the spinal cord and RN (Fig. R3). The connection between the RN and CLNs is polysynaptic. The results are presented in Fig. S7 and Fig. S8 of the revised manuscript. (Lines 1443-1452)

15. Figure S5c: a translation inhibitor is given in vivo. Anisomycin in the Figure not Anisamycin. This is a general translation inhibitor, the side effects are wide. How specific is the effect for CCL5?

RESPONSE: We thank the reviewer for this comment. We administered a monoclonal antibody against CCL5 in the periphery in rats in which RN neurons were inhibited.

Methods:

Systemic neutralization of CCL5. Monoclonal antibodies recognizing CCL5 or control IgG were administered intraperitoneally (250 µg/rat in sterile PBS) to rats injected with AAV-CaMKII α -hM4D(Gi)-mCherry in the RN immediately after CNO injection. (**Lines 876-879**)

Results:

We investigated whether systemic blockade of CCL5 influenced RN inhibition-induced anxiety using a specific neutralizing antibody. The open field test revealed that anti-CCL5 treatment had no effect on the locomotion ability of rats. The time spent in the central zone in the OFT and the open arm ratio in the EPM test were significantly increased in anti-CCL5 rats compared to IgG rats, suggesting that systemic neutralization of CCL5 reversed the anxiety induced by RN inhibition. (**Fig. R4 & Fig. S10**) (**Lines 339-345 & Lines 1464-1472**)

Minor:

1. *Stats are missing in Figure 5e, and 5p.*

RESPONSE: We have added the statistics to Figure 5e and 5p.

2. *Fig1a, the color code seems to be inverted compared to the rest of the figure.*

RESPONSE: Thank you for identifying this error. We have revised the color code.

3. *Line 150: “circadian” rhythms ?*

RESPONSE: Thank you for identifying this error. We have updated the sections.

4. *Fig 2o-r: this part is very confusing both in figure and the text, the paper would benefit from a more clear and logical explanation, as this part feels detached to the rest*

RESPONSE: Thanks for figure it out. We have re-organized the Fig 2o-r.

5. *Fig 3b: the 2 RN sections presented look very different, with different expression levels and patterns of RFP*

RESPONSE: Thank you for identifying this error. We have updated Fig. 3b.

6. *Fig 3d: what about other areas?*

RESPONSE: Thank you for identifying this error. We also observed c-fos expression in the mPFC, M1, BLA, CI, Tu, PAG and so on.

7. *Fig4 f-j are not properly explained in the text but just mentioned*

RESPONSE: Thank you for identifying this error. We have explained Fig. 4 f-j specifically.

8. *Fig 5 is crammed and difficult to digest*

RESPONSE: Thank you for identifying this error. We have reorganized Fig. 5.

9. *Line 301: what is a post hoc immunostaining??*

RESPONSE: Thank you for identifying this error. We have revised the description.

10. *Line 302: reduction in the density of vGlut1+ cells in the injection site (Fig. 5j).
Figure did not correspond to the line*

RESPONSE: Thank you for identifying this error. We apologize for the mistake. We have revised Fig. 5.

11. *Line 306-8: how come these are independent experiments? Are they not performed on the same animals?*

RESPONSE: Thank you for identifying this error. We have revised the statement.

12. *ELS and CUMS needs to be briefly explained in the text*

RESPONSE: Thank you for identifying this error. We have explained the ELS and CUMS protocols in the manuscript (**Lines 686-703**).

Reviewer #3 (Remarks to the Author):

This is a potentially high-impact report integrating human and rodent data and defining a specific brain circuit and neuroimmune mechanism that appears to be important in mediating stress effects on exploratory behavior in an anxiogenic context. The authors show that after stress, CCL5 levels are increased in humans and rodents; that this effect is driven by lymphocytes in cervical lymph nodes; and it's controlled by glutamatergic projections from the midbrain red nucleus to those lymph nodes. Silencing RN cells decreased exploratory behavior in an open field and increased CCL5 levels after restraint stress.

Overall, this is an important paper that is poised to have a significant impact on the field, pending some revisions. The topic (neuroimmune interactions) is widely understood to be important, timely, and broadly interesting but also understudied. The circuit mapping studies, rigorously examining both inputs and outputs in an unbiased way are a major strength, as is the integration of data from humans and rodents in a way that will surely enhance the impact. There is also an abundance of data from various experimental sources supporting the key conclusions.

That said, there are also some significant issues that would need to be addressed in a revision:

In my view the major weakness of the paper is the behavioral data. Behavior in the open field test and elevated plus maze are just very hard to interpret, and the tests don't lend themselves well to pairing with photometry. Fundamentally, it's unclear what the red nucleus cell activity during open field exploration really means -- the link to "anxiety" is tenuous. In the context of all the other data in this paper, I don't view this as a fatal weakness, but I do have some suggestions:

- 1. First, I am not sure what we are meant to conclude from the photometry experiment in Fig. 4. The authors write, "In summary, RN glutamatergic neurons played an important role in stress-induced anxiety-like behavior." But it's not really clear what role they played, whether that role is important, and how the actual observations of signals time-locked to center zone entry relate to anxiety. If anything, it seems like periods when the mouse is exploring the center zone would be evidence of **reduced** anxiety or maybe increased exploratory drive, not elevated anxiety, and RN glutamatergic neurons may be playing a role in supporting exploratory behavior in potentially anxiogenic contexts, but not in anxiety per se..*

*The chemogenetic experiments in Figure 5 help to some degree but it's still a bit confusing and hard to interpret: In Figure 4, the authors find that there's an increase in RN cell activity **after** center zone entry. In Figure 5, the authors find that silencing RN cells reduces time spent in the center zone. How can silencing cells that become active **after** the mouse's decision to enter the center zone*

result in a reduction in center zone exploration time? One possible interpretation is that RN cells become active during center zone exploration and somehow, they reinforce or sustain this behavior, possibly serving as a safety signal or by activating other brain regions that promote exploration or by inhibiting activity in other brain regions that signal threat. It would be interesting to know whether the authors observed a reduction in center zone entries in addition to a reduction in time spent in the center zone as reported in Fig. 5C. If not, that would be consistent with the idea that RN activity isn't actually initiating exploration / center zone entries, but it is important for sustaining exploration over time.

RESPONSE: We thank the reviewer for this comment. We analyzed the frequency in the central zone in the open field test as suggested by the reviewer. There was no reduction in central zone entries after inhibiting or silencing RN cells in either the open field test or the elevated plus maze test, indicating that RN activity does not initiate exploration/central zone entries, but it is important for sustaining exploration over time.

Fig. R8 Analysis of the frequency of entering the central zone in the open field test after inhibiting or silencing RN cells. a, Inhibiting RN glutamatergic neurons had no effect on central zone entries in the open field test. b, Ablating RN glutamatergic neurons had no effect on central zone entries. (two-tailed unpaired t test, $t = 0.4492$, $P = 0.6597$, $n = 8-9$ rats; $t = 0.4076$, $P = 0.6902$, $n = 8-9$ rats).

2. *Optogenetic inhibition experiments timed to occur only after center zone entries or only during time in the periphery would also be helpful. As is, the role for RN cell activity remains unclear. I think optogenetic experiments could go a long way toward clarifying conclusions about what these cells are actually doing with respect to behavior. It would also be helpful to include optogenetic and*

photometry experiments from more interpretable behavioral assays, although this might be outside the scope of what's feasible given all the other experiments in this paper.

RESPONSE: We thank the reviewer for this suggestion. To understand the role of RN cell activity, we performed optogenetic inhibition experiments timed to occur only after central zone entries and optogenetic activation experiments timed to occur only during time in the periphery.

Methods:

We delivered AAV-CaMKII-hChR2(H134R)-mCherry to M1 macrophages. Rats were implanted bilaterally with an optical cannula in the RN immediately after AAV injection. Then, the rats were returned to their home cages and allowed to recover for 3 weeks. ChR2 was activated through a solid-state laser (473 nm, Inper, China). NpHR3.0 was activated through a solid-state laser (589 nm, Inper, China).

Results:

Activating M1-RN glutamatergic neurons when the rat was in the peripheral area prompted the rat to explore, but not every activation made the rat go to the central zone in the open field test. In addition, inhibiting RN glutamatergic neurons led to the rat withdrawing toward peripheral regions when the rat was in the central area. Furthermore, activating M1-RN glutamatergic neurons increased the time spent in the central zone, and inhibiting RN neurons decreased the duration in the open field test (Fig. R1). These results confirmed the conclusion that the activation of RN glutamatergic neurons is a necessary condition for exploratory behavior in the central zone. However, animals do not necessarily go to the center when RN glutamatergic neurons are activated.

- 3. Figure 5: The authors claim that "CNO injection reduced the central zone duration in open field test and open arm ratio in elevated plus maze test in hM4D(Gi) rats, but not in mCherry control rats (Fig. 5c-d)." However, the effect in the EPM was not significant, which undermines the claim about the EPM. With N=8-9 mice per group, the experiment is probably underpowered and could benefit from a larger sample.*

RESPONSE: We thank the reviewer for this point and expanded the sample size in this experiment. The results showed that CNO injection reduced the open arm ratio in the elevated plus maze test in hM4D(Gi) rats. However, CNO injection had no effect on the open arm frequency in the EPM test. We also updated the data in Fig. 5. (**Lines 1308-1311**)

Fig. R9 Analysis of the open arm ratio and frequency in the EPM test after inhibiting RN cells. a, CNO injection reduced the open arm ratio in the elevated plus maze test in hM4D(Gi) rats. b, CNO injection had no effect on the open arm frequency in the elevated plus maze test in hM4D(Gi) rats (two-tailed unpaired t test, $t = 2.579$, $P = 0.0150$, $n = 15-17$ rats; $t = 1.101$, $P = 0.2797$, $n = 15-17$ rats).

4. *Figure 4: instead of or maybe in addition to plotting many sample photometry traces superimposed on top of each other in Fig. 4c-d, it might be more informative to plot a single representative photometry trace and mark periods when the mouse is in the center zone vs. the periphery. That might provide a better illustration of the group mean traces in Fig. 4h-i, and also a better sense of the temporal dynamics, how they vary with repeated entries, etc.*

RESPONSE: We thank the reviewer for this suggestion. We plotted a single representative photometry trace and marked periods when the rat was in the central zone (Fig. 4c and Fig. R10).

Fig. R10 GCaMP6 signals from RN glutamatergic neurons aligned to the moment of exploratory behavior to the central zone in the open field. Control: black; RS: red. Gray areas indicate rats in the central zone.

Restraint stress significantly decreased GCaMP6s fluorescence in RS rats compared to control rats. In the open field test, the Ca^{2+} signals increased once the rats exhibited

exploratory behavior. Increased Ca^{2+} signals prompted rats to enter the central zone in the open field test (Fig. 4c-d). Compared to control rats, RS rats showed lower Ca^{2+} signals during exploratory behavior in the open field test (Fig. 4e-i). In summary, the activation of RN glutamatergic neurons is a necessary condition for exploratory behavior in the central zone. However, animals do not necessarily go to the center when RN glutamatergic neurons are activated. (Lines 290-298)

5. *The second conceptual problem with this paper is that the links between the human data and rodent data are messy and a little hard to interpret. The rodent data focus on the association between stress, CCL5, and anxiety. The human data show an increase in CCL5 levels after stress but there is only one measure of anxiety in the human data -- GAD7 self-report scores -- and they are not increased in the stressed group. Further undermining the association between CCL5 and anxiety, the authors observed modest correlations between CCL5 levels and a number of psychiatric symptom scores, but zero correlation with anxiety as measured by GAD7. In humans, CCL5 levels seem unrelated to anxiety. This critique also applies to how the authors interpret their rodent data: for all the reasons noted above, and based on their findings in humans, it seems to me that the data are pointing to a role for CCL5 and red nucleus neuronal activity not in anxiety but rather in stress responsiveness and possibly in regulating exploratory behavior and other behavioral responses to stressful or threatening contexts. These points should be addressed in the Discussion and should probably lead the authors to revise their claims about anxiety, including in the title and abstract.*

RESPONSE: We thank the reviewer for this comment. In humans, CCL5 levels seem to be unrelated to GAD7 scores. Psychiatrists who worked on the front line during the COVID-19 pandemic in Wuhan had higher scores on the SRQ-20 (for identifying mental disorder symptoms, $p = 0.001$) and PCL-5 (for stress levels, $p = 0.019$) (Fig. 1d, e) than controls.

Anxiety disorders and other mental disorders are distinguished by their symptoms in humans¹⁹. However, many symptoms are shared across these disorders in animals²⁰. The open field test is an experimental test used to assay general locomotor activity levels, anxiety, and willingness to explore in animals (usually rodents) in scientific research. Animals display a natural aversion to brightly lit open areas. However, they also have a drive to explore a perceived threatening stimulus. Decreased levels of anxiety lead to increased exploratory behavior. Increased anxiety will result in less locomotion and a preference to stay close to the walls of the field^{21, 22, 23}. The elevated plus maze test is based on the natural tendency of rodents to explore novel environments and their innate avoidance of unprotected, bright, and elevated places (represented by the open arms). Confinement to the open arms induces physiological signs of stress (increased corticosterone levels)²⁴. Stressful experiences have a powerful impact on brain function

and can lead to both short- and long-term behavioral alterations. Thus, both the open field test and elevated plus maze test can assess the effect of stressful experiences and anxiety. Rats that experienced restraint stress spent less time in the open arm in the EPM test and the central zone in the OFT, suggesting that rats showed anxiety-like behavior (an expression of their stress level). In humans, anxiety is a normal emotion, but excessive anxiety can develop into anxiety disorder, a chronic persistent disorder. GAD7 is used to identify general anxiety disorders and cannot represent the absence of anxiety in stressed individuals. Both the PCL-5 and SRQ20 scores indicated that psychiatrists who worked on the front line during the COVID-19 pandemic in Wuhan had a higher stress level than controls. In summary, our data indicate a role for CCL5 and red nucleus neuronal activity both in anxiety emotion and in stress responsiveness. **(Lines 408-430)**

- 6. I felt that the paper could also be strengthened by expanding the discussion of the current findings in the context of the existing literature, particularly with respect to what is known about immune system responses to psychosocial stress and their role in driving behavior. The authors touch on this briefly in the second paragraph of their discussion, noting that this general concept has been discussed since at least 1950, and they cite two studies observing immune responses to stress but they do not describe what was observed. In general, my impression of the literature is that previously published results are mixed and inconsistent. It might enhance the impact of the current work to approach this literature review with a little more rigor and state clearly whether stress-sensitive changes in CCL5 signaling have been observed before, also noting negative findings, etc. Same goes for the many negative results in Figure 1 (MCP-1, IL-1b, IL-2, TNFa, etc. etc.). Those negative findings enhance the impact of the present work by establishing specificity. But it would also be helpful if the authors could clarify for readers whether previous studies have examined these molecules in the context of stress, noting both positive and negative findings, and perhaps trying to synthesize the results -- how the present work differs from previous studies in ways that might explain what was observed.*

RESPONSE: We thank the reviewer for this suggestion. We have expanded the discussion of the current findings in the context of the literature. **(Lines 439-449)**

“In addition, CCL5 has been previously reported to be altered in subjects displaying anxiety²⁵, depression²⁶, PTSD²⁷ and chronic stress²⁸, which further confirmed our findings. However, these studies did not evaluate in depth the relationship between CCL5 and the brain and anxiety. Many studies have indicated that CCL5 is not altered in mental disorders^{29, 30}. Our results showed that the levels of MCP-1, IL-1 β , IL-2, IL-4, IL-6, TGF- α , TNF- α and BDNF in stressed and control individuals were not different. These results were different from those of many other studies^{28, 31}, possibly because we measured cytokines and chemokines one month after a stressful event. In addition, the literature

within the field is inconsistent due to variables such as age, species, and cultural factors, as well as testing conditions such as the time of day and environment.”

7. *In the abstract, the authors claim that "Together, our data suggest that CCL5 might be a novel, stable and practicable stress level biomarker." I would eliminate all claims about biomarkers. There is too much overlap in CCL5 levels between stressed participants and unstressed control participants to serve as a biomarker.*

RESPONSE: We thank the reviewer for this suggestion. We have updated the description in the abstract. (**Lines 41-42**)

8. *fMRI data: I would move these results to the supplement or maybe cut them. The methods aren't very well described (they refer to seven seeds but do not define them) but my understanding is that the main analysis in Fig. 6 compared functional connectivity between the red nucleus and every other voxel in the brain in a relatively small sample (~35-40 subjects per group) and then used a cluster threshold to correct for multiple comparisons and identify significant differences in the stressed group. This approach is well known for yielding false positives (<https://www.pnas.org/doi/10.1073/pnas.1602413113>), and there are a lot of red flags here: the samples are relatively small for performing tens of thousands of tests, and the reported effects are very small as well, both in magnitude and spatial extent. Further complicating things, subcortical / midbrain signals are notoriously noisy in resting state fMRI datasets: e.g. <https://pubmed.ncbi.nlm.nih.gov/31836321/>. One option would be to evaluate the stability of these results in bootstrapped subsamples of the data. If the results are stable, they might be more persuasive. But a better option might just be to remove the fMRI data or at least move it to the supplement and present it as a preliminary finding with a lot of caveats.*

RESPONSE: We thank the reviewer for this comment and apologize for the misunderstanding. We defined two seeds in our analysis: left and right red nuclei, and we have revised it in our paper (**Lines 647-649**). Considering the small sample size, we moved the MRI results to the supplemental materials.

Minor issues:

9. *Fig. S1A: "Our results showed that several inflammatory cytokines, such as IL2 and TGF α , were negatively correlated with the stress level (Fig. S1a)." This analysis needs to be corrected for multiple testing. This claim probably isn't true*

after making these corrections, since the correlations in question are very weak and just barely exceed a threshold for significance without correction. Same goes for the claims about CCL5 correlating with PCL5, PHQ9, and SRQ scores -- the p values should be corrected. Since those p values are smaller, I would guess they might actually be significant.

RESPONSE: We thank the reviewer for this comment. Both Fig. S1A and CCL5 correlating with PCL5, PHQ9 and SRQ scores were Bonferroni corrected. We have updated the description in **Lines 1367 and 1373**.

10. The images in Fig. S6a aren't great in terms of quality. The insets in S6b help to some degree but they're still hard to see. Are better images available? Are the six regions shown in S6b the only regions that showed any labeling? That very hard to appreciate from Fig. S6a because of the high background. The six named regions appear to have some of the lowest levels of red fluorescence, presumably due to background. But this raises questions about false negatives due to high background. Overall, this is a relatively minor issue since there are any other experiments in this paper implicating RN.

RESPONSE: We thank the reviewer for this comment and apologize for the low quality. We have updated the images as shown in **Figs. S7 and S8**. (**Lines 1443-1452**)

11. Minor: there are two panels labeled "r" in Figure 2

RESPONSE: We thank the reviewer for this comment and apologize for the mistake. We have revised Fig. 2.

12. Minor: for the photometry experiments, the authors note that "We implanted an optic fiber (200 μm optical density, 0.37 numerical aperture) above the unilateral RN to record activity changes of the RN neurons." The scale bar in Fig. 4b makes it look like the diameter of the optical fiber is 100 μm . Is the scale bar a typo?

RESPONSE: We thank the reviewer for this comment and apologize for the mistake. We have revised Fig. 4.

*13. For the RNA seq data in Figure 2, the authors note that "While there are no changes in expression of interferon regulatory factor 1 (IRF1) in stressed CLN, suggesting that RS led to translation dysfunction in CLN, which is consistent with the results of RNA-seq." Why do the results suggest *dysfunction* in translation? Why is this not a normal, healthy, adaptive response, just one component of activating the immune system? How do we know it's evidence of "dysfunction"?*

RESPONSE: We thank the reviewer for this important comment. We have revised this in the manuscript. **(Line 230-232)** “There were no changes in expression of interferon regulatory factor 1 (IRF1) in the CLNs of stressed rats, suggesting that RS promotes translation in CLNs, which is consistent with the results of RNA-seq.”

14. The "stressed" group of humans in Figure 1 were psychiatrists who traveled to Wuhan to perform frontline work. The effects on CCL5 levels in this group are attributed to psychosocial stress, but it's also possible that they were exposed to COVID or some other infection. Do the authors have any data to speak to this? If not, they could just comment on it in the Discussion.

RESPONSE: We thank the reviewer for this comment. All human participants in our research underwent COVID-19 PCR/antigen testing, and the results were negative. **(Lines 568-569)**

Reviewer #4 (Remarks to the Author):

In their manuscript, Shi et al. identify CCL5 as a potential marker of stress in stressed individuals. They further support this finding in rats that underwent restraint stress, revealing cervical LN as the main anatomical origin for CCL5 elevation and its underlying cellular mechanisms. Using retrograde neuronal tracing, they identify specific glutamatergic neurons in the Red Nucleus (RN) that innervate the CLNs. They characterize their electrophysiological properties during stress, dissect a neural circuit in the M1 that regulates these neurons' activity and elegantly demonstrate their functional significance to stress-related behavior and CCL5 elevation via chemogenetic tools. Finally, using fMRI, the authors show reduced functional connectivity in RN of stressed human individuals, consistent with the results they showed in rats.

The findings are novel, interesting and potentially impactful, and the authors are providing overall solid experimental evidence for their claims. Yet, there are several issues that I think should be addressed in order to fully support the authors' conclusions.

Major comments:

- 1. None of the controls (both human and rats) went through any kind of novel non-stressful event, not entirely excluding the possibility that the RN neurons identified in this study are linked to a learning or novel experience process. Please refer to this limitation in the text, or preferentially add such control group,*

showing this doesn't elevate CCL5 in the serums/CLNs and CLN CD8 T cells (e.g., novel object or a motor learning paradigm).

RESPONSE: We thank the reviewer for this important point. We have listed this limitation in the discussion. **(Lines 549-552)**

- 2. Did all human participants undergo COVID19 PCR/antigen testing? As the "stressed" group was exposed to a more infected area, I would recommend making sure and showing that the differences between groups were not a result of a prior/recent infection.*

RESPONSE: Thank you for this very important question. All human participants in our research underwent COVID-19 PCR/antigen testing, and the results were negative. **(Lines 568-569)**

- 3. It would be beneficial if the authors show that upon inhibition of RN glutamatergic neurons, there is an increase in noradrenaline (NA) in the CLN (e.g., via Elisa, IHC). It may further strengthen the link between the stress-induced neuronal response, and CLN immune regulation.*

RESPONSE: As suggested by the reviewer, we detected noradrenaline (NA) in CLNs. Norepinephrine (NA) is one of the primary catecholamines of the sympathetic nervous system released during a stress response and plays an important role in modulating immune function². We detected norepinephrine (NA) in CLNs. The results showed that NA was significantly increased both in stressed rats and in RN inhibition rats (Fig. R11). NA preferentially modulates memory CD8 T-cell function, inducing inflammatory cytokine production³. The increase in CCL5 may be fed back to the brain through noradrenergic fibers, thereby affecting anxiety-like behavior. However, this is a very complicated adjustment process, which more in-depth experiments are needed to explore the mechanisms. We discussed this issue in the discussion **(Lines 520-527)**.

Fig. R11 NA increased in both stressed rats and RN inhibition rats. a, The concentration of NA in RS rats was significantly increased compared to that in controls (two-tailed unpaired t test, $t = 2.668$, $p = 0.0257$). b-c, both ablation (two-tailed unpaired t test, $t = 8.511$, $p = 0.0001$) and inhibition (two-tailed unpaired t test, $t = 14.83$, $p < 0.0001$) of RN neurons led to an increase in NA concentration in CLNs. Data are presented as the mean \pm SEM. * $P < 0.05$, ** $P < 0.01$, *** $P < 0.001$.

4. *I would suggest to avoid using the word “Dysfunctions” in the title to describe the reduced activity of RN in times of stress. An editing suggestion for the title: Stress-induced Red Nucleus Attenuation induces Anxiety-like Behavior and Lymph Node CCL5 secretion.*

RESPONSE: We thank the reviewer for this suggestion. We have updated the title to ‘Stress-Induced Red Nucleus Attenuation Induces Anxiety-Like Behavior and Lymph Node CCL5 Secretion’.

5. *Although there is a simultaneous rise in CCL5 and CD8 T cells in CLN of stressed rats, it still doesn’t entirely prove that these cells are the subgroup that secretes CCL5. It will be valuable if the authors intracellularly stain CD8 T cells to CCL5.*

RESPONSE: We thank the reviewer for this comment. As suggested by the reviewer, we performed intracellular cytokine staining in cervical LNs. For intracellular cytokine staining, T cells were obtained immediately from the CLNs of stressed rats and then permeabilized using intracellular staining permeabilization wash buffer (eBioscience, 88-8824-00). Then, intracellular CCL5 was analyzed by flow cytometry. (Lines 758-761) Without additional stimulation, the CD8⁺ T cells from RS rats exhibited a significant

increase in CCL5 expression compared to those from control rats (Fig. R12&Fig. S5c). (Lines 211-213)

Fig. R12 CD8+ T cells were isolated from the CLNs of CTRL and RS rats, and incubated with monensin (1ug/mL) for 4 h before harvest. Flow cytometric analysis of the percentage of CCL5-producing CD8+ T cells (n = 4). Data are presented as the mean \pm SEM. *P < 0.05, **P < 0.01, ***P < 0.001.

6. *As the authors mentioned, the fact that female rats did not show signs of stress and no cortisol level increase, but still presented elevated CCL5 levels, weakens the claim that stress is related to the chemokine dynamics. Are there any stress tests that are known to be more suitable for female rats other than the open field or elevated maze, or professional literature evidence for poor outcomes of stress test in female rodents?*

RESPONSE: We thank the reviewer for this point. In our research, we found that female rats showed better stress resilience than males. The resilience of female rats to stress may be because of the protective effect of estrogen^{13, 14, 15}. Current and previous findings highlight the need to develop animal models and testing protocols that better represent female anxiety-like behavior¹³. Female rats may appear to be less anxious than males, but further investigation into the appropriateness of the standard tests like EPM, open field and social interaction for understanding sex and cycle differences in anxiety disorders is warranted and perhaps the development of female-specific anxiety measures are needed¹³.

7. *Overall, poorly written (also making some of the experimental explanations not entirely clear): Use of both present and past tenses interchangeably, spelling and grammar mistakes, and questionable choices of words (“besides”, “fortunately”, “so on”, “dramatic”). The discussion is also very repetitive and unorganized.*

Would highly recommend the text goes through professional proofreading (including Legends and Materials and Methods section).

RESPONSE: We apologize for the mistakes in the manuscript. We have revised the writing of this manuscript. The manuscript was edited for proper English language, grammar, punctuation, spelling, and overall style by one or more of the highly qualified native English-speaking editors at SNAS.

8. *Figure 6: Are the brain regions that are presented in Fig. 6m, the only areas that showed significant differences between the groups? I would expect additional regions to be involved during stress (e.g., hypothalamus, amygdala, prefrontal cortex).*
- *fMRI data is not in the scope of my expertise, so couldn't fully assess these results.*

RESPONSE: We defined two functional regions of interest (ROIs) in our analysis: the left and right red nucleus (RN), and we used WFU PickAtlas software to generate the ROIs of the RN³². Then, we used the bilateral RN as ROIs to conduct voxelwise rsFC analysis between each seed and the whole-brain voxels by DPABI. Whole-brain analysis did not find differences in these brain regions (e.g., hypothalamus, amygdala, prefrontal cortex), which may be due to the small sample size.

9. *As the authors stated, it is surprising that cortisol levels were diminished in individuals from the stressed group, rendering the stress questionnaire the only measure by which they confirmed these individuals' elevated stress levels. The authors should address this limitation in the text, and- if possible- measure cortisol levels of these participants at an earlier time point (closer than 1 month to the stressful event).*

RESPONSE: We thank the reviewer for this comment. Glucocorticoids are increased during stress, especially in the early phase. Unfortunately, we did not collect blood during or immediately after stress. According to the changes in glucocorticoids after stress, we speculate that glucocorticoids increase first and compensatorily decrease after stress. Thus, one month after subjects left the stressful environment, glucocorticoids were lower in stressed subjects than in controls because of a compensatory decrease. Six months later, the glucocorticoid levels returned to normal (Fig. 1). We have addressed this limitation in the text. (Lines 537-542)

Minor comments:

1. *There are many figure reference mistakes throughout the text, figures and legends. Some of these are listed here:*

- a. Line 150: Fig.S2a - S3a?
- b. Line 152: Fig. S2d - S3d?
- c. Lines 211, 213, 214, 216 - Fig.s-v? (Fig.2- "r" appears twice in the figure)
- d. Line 261: Fig. 3e - 3f?
- e. Line 264: Fig. 3f - 3e?
- f. Lines 274-5: Fig. ref?
- g. Line 279: Fig.S7a?
- h. Line 302: Fig. 5j - 5g?
- i. Line 325: Fig. S5a-b - S6a-b?
- j. Line 599: wrong Table ref.
- k. Line 680: Table S5 - S4?

RESPONSE: We apologize for the mistakes in the manuscript. We have revised these mistakes in this manuscript.

2. *The figures' panels are not in the same order as their presentation in the text, causing quite a confusion throughout the reading (one example: Fig.1g). Recommend changing the order of the plots to the order they appear in the text.*

RESPONSE: We apologize for the mistakes in the manuscript. We have revised these mistakes in this manuscript.

3. *The authors should clarify undefined abbreviations: For example, FC (line 366), EPR (line 531), PND (line 565, where it first appears and not later in the text).*

RESPONSE: We have clarified undefined abbreviations in the manuscript.

4. *The number of human participants is inconsistent throughout the text, figures and tables: Table 1- 88 (n=47, 41), Methods fMRI- 82 (n=44, 39)?*

RESPONSE: We apologize for the misunderstanding. In Table 1, 88 (n = 47, 41) were recruited for blood collection. Eighty-three participants were recruited, including forty-four stressed psychiatrists and thirty-nine demographically matched controls, for fMRI analysis.

5. *How did the authors acclimatized the mice before behavioral tests? The authors should depict the methodological details.*

RESPONSE: Rats have a minimum acclimation period of 3 days prior to performing behavioral tests. For pre-experimentation handling, we removed the rat from its cage and

either simply held or stroked the animal for a duration of no less than one minute at least once per day for 3 days prior to behavioral assays. (**Lines 708-711**)

6. *There is no mentioning of CD45RC in the text and Fig. 1q-r legend. The authors should state why they measured this marker and what its increase indicates.*

RESPONSE: We thank the reviewer for this comment and apologize for the omission. We have added information on CD45RC in the text (**Lines 208-210**) and Fig. 2 legend. CD8+CD45RC-: memory CD8 T cells; CD8+CD45RC+: Naïve CD8 T cells. (**Lines 1248-1249**)

7. *Line 50: What is the “inflammatory system”? Recommend changing to “inflammatory response”.*

RESPONSE: We thank the reviewer for this suggestion. We have changed “inflammatory system” to “inflammatory response”.

8. *Line 68: Add a reference to the statement.*

RESPONSE: We have added Fig. ref.

9. *Line 147: add “Circadian” rhythm.*

RESPONSE: We have added circadian rhythm to the manuscript.

10. *Line 226: “The inhibition effects were confirmed by immunohistochemistry.” Add Fig. ref.*

RESPONSE: We have added Fig. ref.

11. *Line 277: Add a reference.*

RESPONSE: We have added Fig. ref.

12. *Line 290: Why “conditionally”?*

RESPONSE: We used the promoter of CaMKII α , suggesting that only glutamate neurons can express the inhibitory receptor hM4D(Gi).

13. Fig. 2:

- a. 2a: add days units to the scheme.
- b. 2f: Live - Liver?
- c. 2k: problematic to state that there is a difference between groups while it is not significant... The authors can write "a trend toward higher..."
- d. 2l: it seems there is a difference in CCL5 also in the sham group, before, during and after stress. Is't it significant? This suggests that there is another possible source of CCL5, bone marrow (BM) for example, which is known to also be innervated and affected by the sympathetic nervous system (and therefore by stress responses). Would be beneficial if the authors will measure levels of CCL5 in BM as well.

RESPONSE: We thank the reviewer for this comment and apologize for the mistakes in Fig. 2. We have revised Fig. 2a and 2f and updated the statement of Fig. 2k.

14. Fig. 3:

- a. 3a: Glass pipette instead of electrode?
- b. 3b: What do the arrows signify?
- c. The title of the figure is not clear.

RESPONSE: We have revised Fig. 3 as the reviewer suggested. We changed the title to "Identification of a cluster of RN glutamate neurons upstream of CLNs."

15. Fig. 5:

- a. 5i: Missing units for Y axis

RESPONSE: We have added the units for the Y-axis in Fig. 5.

16. Fig.6: no 6l in legends, wrong 6m, n.

RESPONSE: We apologize for the mistake, and we have revised Fig. 6.

17. In Fig. S7, the experimental design and timeline is not clear. The authors should add an experimental scheme.

RESPONSE: We have clarified undefined abbreviations in the manuscript.

Reference

1. Louis T. Giron KAC, James N. Davis. Lymph nodes--a possible site for sympathetic neuronal regulation of immune responses. *Annals of Neurology* **8**, 520-525 (1980).
2. Sanders VM, Straub RH. Norepinephrine, the beta-adrenergic receptor, and immunity. *Brain Behav Immun* **16**, 290-332 (2002).
3. Slota C, Shi A, Chen G, Bevans M, Weng NP. Norepinephrine preferentially modulates memory CD8 T cell function inducing inflammatory cytokine production and reducing proliferation in response to activation. *Brain Behav Immun* **46**, 168-179 (2015).
4. Spitzer RL, Kroenke K, Williams JB, Löwe B. A brief measure for assessing generalized anxiety disorder: the GAD-7. *Arch Intern Med* **166**, 1092-1097 (2006).
5. Plummer F, Manea L, Trepel D, McMillan D. Screening for anxiety disorders with the GAD-7 and GAD-2: a systematic review and diagnostic metaanalysis. *Gen Hosp Psychiatry* **39**, 24-31 (2016).
6. Kroenke K, Spitzer RL, Williams JB. The Patient Health Questionnaire-2: validity of a two-item depression screener. *Med Care* **41**, 1284-1292 (2003).
7. Kroenke K, Spitzer RL, Williams JB. The PHQ-9: validity of a brief depression severity measure. *J Gen Intern Med* **16**, 606-613 (2001).
8. Orley MBaJ. A user's guide to the self reporting questionnaire (SRQ). *World Health Organization*, (1994).
9. Blevins CA, Weathers FW, Davis MT, Witte TK, Domino JL. The Posttraumatic Stress Disorder Checklist for DSM-5 (PCL-5): Development and Initial Psychometric Evaluation. *J Trauma Stress* **28**, 489-498 (2015).
10. Bovin MJ, *et al.* Psychometric properties of the PTSD Checklist for Diagnostic and Statistical Manual of Mental Disorders-Fifth Edition (PCL-5) in veterans. *Psychol Assess* **28**, 1379-1391 (2016).
11. Giron LT, Jr., Crutcher KA, Davis JN. Lymph nodes--a possible site for sympathetic neuronal regulation of immune responses. *Ann Neurol* **8**, 520-525 (1980).
12. Albayram MS, *et al.* Non-invasive MR imaging of human brain lymphatic networks with connections to cervical lymph nodes. *Nat Commun* **13**, 203 (2022).

13. Scholl JL, Afzal A, Fox LC, Watt MJ, Forster GL. Sex differences in anxiety-like behaviors in rats. *Physiol Behav* **211**, 112670 (2019).
14. Díaz-Véliz G, Alarcón T, Espinoza C, Dussaubat N, Mora S. Ketanserin and anxiety levels: influence of gender, estrous cycle, ovariectomy and ovarian hormones in female rats. *Pharmacol Biochem Behav* **58**, 637-642 (1997).
15. Lopez-Aumatell R, *et al.* Fearfulness in a large N/Nih genetically heterogeneous rat stock: differential profiles of timidity and defensive flight in males and females. *Behav Brain Res* **188**, 41-55 (2008).
16. Bradley J, Swanson MM, Thomas C, Mitchell, John Kappler, Phillippa Marrack. RANTES Production by Memory Phenotype T Cells Is Controlled by a Posttranscriptional, TCR-Dependent Process. *Immunity* **17**, 605-615 (2002).
17. Devi S, *et al.* Adrenergic regulation of the vasculature impairs leukocyte interstitial migration and suppresses immune responses. *Immunity* **54**, 1219-1230 e1217 (2021).
18. Clark SM, Song C, Li X, Keegan AD, Tonelli LH. CD8(+) T cells promote cytokine responses to stress. *Cytokine* **113**, 256-264 (2019).
19. Association AP. *Diagnostic and Statistical Manual of Mental Disorders: DSM-5*, 5 edn (2013).
20. Devi S, *et al.* Adrenergic regulation of the vasculature impairs leukocyte interstitial migration and suppresses immune responses. *Immunity* **54**, 1219-1230.e1217 (2021).
21. Denenberg VH. Open-field behavior in the rat: what does it mean? *Ann N Y Acad Sci* **159**, 852-859 (1969).
22. Stanford SC. The Open Field Test: reinventing the wheel. *J Psychopharmacol* **21**, 134-135 (2007).
23. Sturman O, Germain PL, Bohacek J. Exploratory rearing: a context- and stress-sensitive behavior recorded in the open-field test. *Stress* **21**, 443-452 (2018).
24. Pellow S, File SE. Anxiolytic and anxiogenic drug effects on exploratory activity in an elevated plus-maze: a novel test of anxiety in the rat. *Pharmacol Biochem Behav* **24**, 525-529 (1986).

25. Ogłodek EA, Szota AM, Just MJ, Moś DM, Araszkiwicz A. The MCP-1, CCL-5 and SDF-1 chemokines as pro-inflammatory markers in generalized anxiety disorder and personality disorders. *Pharmacol Rep* **67**, 85-89 (2015).
26. Camacho-Arroyo I, *et al.* Chemokine profile in women with moderate to severe anxiety and depression during pregnancy. *BMC Pregnancy Childbirth* **21**, 807 (2021).
27. Hoge EA, Brandstetter K, Moshier S, Pollack MH, Wong KK, Simon NM. Broad spectrum of cytokine abnormalities in panic disorder and posttraumatic stress disorder. *Depress Anxiety* **26**, 447-455 (2009).
28. Polacchini A, *et al.* Distinct CCL2, CCL5, CCL11, CCL27, IL-17, IL-6, BDNF serum profiles correlate to different job-stress outcomes. *Neurobiol Stress* **8**, 82-91 (2018).
29. Roomruangwong C, Sirivichayakul S, Carvalho AF, Maes M. The uterine-chemokine-brain axis: menstrual cycle-associated symptoms (MCAS) are in part mediated by CCL2, CCL5, CCL11, CXCL8 and CXCL10. *J Affect Disord* **269**, 85-93 (2020).
30. Shen Y, *et al.* Altered plasma levels of chemokines in autism and their association with social behaviors. *Psychiatry Res* **244**, 300-305 (2016).
31. Bocchio-Chiavetto L, *et al.* Immune and metabolic alterations in first episode psychosis (FEP) patients. *Brain Behav Immun* **70**, 315-324 (2018).
32. Huang X, *et al.* Altered functional connectivity of the red nucleus and substantia nigra in migraine without aura. *J Headache Pain* **20**, 104 (2019).

Reviewers' Comments:

Reviewer #1:

Remarks to the Author:

The authors have done an adequate job of revising their manuscript. I have no further concerns.

Reviewer #2:

Remarks to the Author:

The authors thoroughly replied to my concerns and revised figures and text to make the manuscript clearer and more understandable. With respect to some minor concerns, the CCL5 imaging data (Fig. 2g) is not convincing and should be taken out. Line 265 – 'fortunately' needs to be rephrased and the acronyms pertaining to the questionnaires need to be spelt out already in the main text.

Reviewer #4:

Remarks to the Author:

In their revised manuscript, Shi et al. further solidified their study's interesting main conclusions regarding the M1-RN neuronal circuit and how it affects anxiety-like behavior by specifically targeting CLNs and elevated CCL5 secretion. Moreover, the clarity of the text itself and the broader noting of the current literature is largely improved.

The authors have addressed most of my technical and theoretical initial concerns. Yet, in my opinion, there are still numerous inaccuracies throughout the text and figures that should be addressed in order to depict the scientific data presented reliably:

1. The abstract is still not very well-written. First, there is a big theoretical jump between the two first sentences, going directly from stress to immune regulation by brain activity. Moreover, the state that "RN glutamatergic neurons triggered anxiety-like behavior after receiving signals through M1 projections" is a bit misleading since, eventually, the authors show that inhibition of this circuit is what reduces stress-related behavior and not that the RN neurons trigger it. Overall, the abstract is not really cohesive, feeling like it is a gathering of fragmented conclusions derived from the study.

2. No reference in the text to Fig. 1a.

3. The authors refer several times to the "acute phase" after stress, yet it is not clear what exactly they are referring to; is it one month after the stressful event? For clarity, they should define it.

4. Line 168: add Fig. reference.

5. Regarding c-Fos increase in RN: If attenuation of RN simulates a stressful event (including the decrease in firing rates showed later on with fiber photometry), how do the authors explain the increase in c-Fos in RN glutamatergic (stimulatory) neurons of stressed rats?

6. Fig. 2:

a. Fig. 2g: The authors should add "CCL5" in green to the images themselves for a clearer and more intuitive figure understanding.

b. Fig. 2l legends: The authors should add statistics to both of the groups. Moreover, the authors did not respond/relate to my previous comment regarding this panel and the fluctuations in CCL5 concentrations in the sham group.

c. In Fig.2o, it seems that the gating is on CD45RC+, meaning naïve CD8 T cells (Ordonez L. et al, PloS ONE, 2009). Opposite to what the authors are stating in the text and showing in Fig.2P.

i. The x and y axes' fluorescence intensity units are small and not clear.

ii. The authors should add percentages in the plot like they added in similar other plots.

d. Not clear why the authors sometimes add statistical details and sometimes don't. There should be more consistency throughout the text, for example, in Legends of Fig.2.

7. Fig. 3:

a. 3b. the authors did not relate to my previous comment regarding this image about clarifying what the arrows signify. If it is pointing to the co-localized cells I would also suggest enlarging the image since it is hard to detect.

8. Line 181: add a reference to Fig. S2.
9. Line 200, 202: "peripheral blood", and not just "peripheral".
10. In lines 226-7, the authors state that they measured concentrations of cytokines and chemokines, suggesting that they measured protein levels (with ELISA, for example), while they are actually referring to the RT-PCR data, measuring RNA levels in CLNs. It is confusing and inaccurate.
11. Fig. S6:
 - a. Fig. S6c: Anisomycin is still misspelled in the scheme.
 - b. Fig. Sj: The image of the EPM is supposed to show that the rats treated with Anisomycin spent more time in the open arms. Yet, if I understand correctly and the open arms are the vertical arms (also something worth mentioning in the figure- which arms are which), it shows the exact opposite trend between the groups.
12. Line 300: The new section's title refers to the inhibition of RN, yet this section also includes a description of the effects of RN activation. Either separate it into two sections (inhibition and activation) or generalize the title, something like "Chemogenetic manipulation of RN glutamatergic neurons affects...".
13. Line 347: As in the abstract, the title of this section is confusing; it suggests M1-derived RN neurons trigger anxiety-like behavior, while in fact, as the authors show, inhibition of the M1-RN circuit induces anxiety-like behavior. Similar to line 504, in the discussion section.
14. Line 351: Add fig. reference.
15. Line 370: There is no Fig. 6l as mentioned in the text.
16. Line 375: DREADD, not DREDD
17. Discussion:
 - a. Throughout this section, the authors rightfully insert statements that are based either on previous studies or the current study. However, it is not always clear when they are referring to which. One example of this is in line 500; after several referenced data, they go back to their own study but do not mention it; in such a case, I would expect them to add "Here, we showed..."/ "In this study, we demonstrated..." and so on.
 - b. Line 467: A reference is needed for this sentence.
 - c. Line 468-9: This sounds like a very significant conclusion, although it is not entirely clear what the authors are basing this statement on.
 - d. Line 472: "electrical" recordings?
 - e. Line 479: Add a reference to this statement about the prefrontal-thalamic circuit.
 - f. The mentioning of CLNs' NA data in the discussion seems quite misplaced. First, it seems more relevant to the result section; second, it is not clear from the text what is known from the literature and what the authors examined in the current study; third, if the authors mention that they checked NA levels in the text I would expect it to be presented, possibly in the supplementary figures (and respectively, how they measured it in the methods section).
 - g. Overall, it feels like a very long discussion with too many repetitions on the study's main conclusions. I would suggest focusing on the interpretation of the authors' main findings (and, as already inserted, limitations of the study) in the context of the scientific literature regarding stress, the relevant brain neuronal circuits, peripheral immunity, and their interconnections.
18. Methods section: The authors are not explaining how they extracted/sorted the T cells before intracellular staining. Did they stain all T cells or specifically CD8 T cells?
19. Methods section: c-Fos staining is not mentioned or described.

Dear reviewers,

We hereby submit a revised manuscript, “Stress-Induced Red Nucleus Attenuation Induces Anxiety-Like Behavior and Lymph Node CCL5 Secretion” (ID: NCOMMS-22-45871A). We appreciate the time and effort that you have dedicated to providing feedback on our manuscript and grateful for the insightful comments on and valuable improvement to our paper.

We have thoroughly examined all the suggestions provided by the reviewer 4. We hope that this revision has addressed all the issues raised by the reviewer 4. We have highlighted the changes within the manuscript. Please find our point-by-point response to the comments of reviewer 2 and reviewer 4 below (reviewer’s comments are shown in italics). We look forward to your subsequent evaluations.

Point-by-point responses to the reviewers' comments:

Reviewer #1

The authors have done an adequate job of revising their manuscript. I have no further concerns.

RESPONSE: We thank the reviewer for his or her positive comments about the importance of our work.

Reviewer #2

The authors thoroughly replied to my concerns and revised figures and text to make the manuscript clearer and more understandable. With respect to some minor concerns, the CCL5 imaging data (Fig. 2g) is not convincing and should be taken out. Line 265 – ‘fortunately’ needs to be rephrased and the acronyms pertaining to the questionnaires need to be spelt out already in the main text.

RESPONSE: We thank the reviewer for this comment. And we have taken out the CCL5 imaging data in Fig. 2g. And we removed the “fortunately”. (**Line 267**)

We have spelt out the acronyms pertaining to the questionnaires in the main text. (**Line 99-105**)

Reviewer #4

In their revised manuscript, Shi et al. further solidified their study’s interesting main conclusions regarding the M1-RN neuronal circuit and how it affects anxiety-like behavior by specifically targeting CLNs and elevated CCL5 secretion. Moreover, the clarity of the text itself and the broader noting of the current literature is largely improved.

The authors have addressed most of my technical and theoretical initial concerns. Yet, in my opinion, there are still numerous inaccuracies throughout the text and figures that should be addressed in order to depict the scientific data presented reliably:

RESPONSE: We thank the reviewer for his or her positive comments about our work. Below, we address the concerns of the reviewer point by point.

1. *The abstract is still not very well-written. First, there is a big theoretical jump between the two first sentences, going directly from stress to immune regulation by brain activity. Moreover, the state that “RN glutamatergic neurons triggered anxiety-like behavior after receiving signals through M1 projections” is a bit*

misleading since, eventually, the authors show that inhibition of this circuit is what reduces stress-related behavior and not that the RN neurons trigger it. Overall, the abstract is not really cohesive, feeling like it is a gathering of fragmented conclusions derived from the study.

RESPONSE: We thank the reviewer for this important comment. We have reorganized the abstract according to the reviewer's suggestions.

“Previous studies have speculated that brain activity directly controls the immune responses in lymphoid organs. However, the upstream brain regions that control lymphoid organs and how they interface with lymphoid organs to produce stress-induced anxiety-like behavior remain elusive. Using stressed humans and rats models, we show that CCL5 levels were increased significantly in stressed individuals compared to controls. Stress-inducible CCL5 is mainly produced from cervical lymph nodes (CLN). Retrograde tracing from CLN identified glutamatergic neurons in the red nucleus (RN), the activities of which were tightly correlated with the CCL5 level and anxiety-like behavior. Ablation or chemogenetic inhibition of RN glutamatergic neurons increased anxiety levels and CCL5 expression in the serum and CLNs, whereas pharmacogenetic activation of these neurons reduced anxiety levels and CCL5 synthesis after restraint stress exposure. RN receives the projection from M1, and chemogenetic inhibition of M1-RN neural circuit elicited anxiety and CCL5 synthesis. These findings provide a mechanistic understanding of the stress-induced anxiety and the inflammatory response. This brain-lymph node axis might provide new insights into lymph node tissue as a stress-responsive endocrine organ and mechanistic insight into targeting this axis in the treatment of neuropsychiatric diseases.” (Line 27-43)

2. *No reference in the text to Fig. 1a.*

RESPONSE: We thank the reviewer for this remind. And we have added the reference in the text to Fig.1a. (Line 95)

3. *The authors refer several times to the “acute phase” after stress, yet it is not clear what exactly they are referring to; is it one month after the stressful event? For clarity, they should define it.*

RESPONSE: We thank the reviewer for this suggestion. According to the definition of DSM-5 (The diagnostic and statistical manual of mental disorders, DSM), the acute phase refers to within one month after stress. And we have explained the acute phase in the text. “Acute phase refers to within one month after stress according to DSM-5 (The diagnostic and statistical manual of mental disorders, DSM)” (Line 103-105)

4. *Line 168: add Fig. reference.*

RESPONSE: Thanks and we have added the reference in the text. (Line 174)

5. *Regarding c-Fos increase in RN: If attenuation of RN simulates a stressful event (including the decrease in firing rates showed later on with fiber photometry), how do the authors explain the increase in c-Fos in RN glutamatergic (stimulatory) neurons of stressed rats?*

RESPONSE: We thank the reviewer for these important comments. Red nucleus comprises a caudal magnocellular and a rostral parvocellular part. The caudal part of red nucleus is positive only for vGluT, whereas the rostral part is positive for both vGluT and GAD67. In our study, we found the increase in c-Fos in RN neurons of stressed rats, but the types of activated neurons were not studied. Thus, we first determine the location of activated neurons. We found that there were no c-Fos expression in caudal part, but c-fos expression was increased in rostral part. It is suggested that stress did not activate glutamatergic neurons in caudal part. Neurons activated in rostral part may be GABAergic neurons. Activation of GABAergic neurons has the potential to inhibit the activity of glutamatergic neurons. In our study, we identified that RN controls CLN, and that chemogenetic manipulation of RN glutamatergic neurons affects anxiety-like behavior and CCL5 secretion. Microcircuits in RN brain regions is complex and require further study. So, we removed this result from the text.

6. *Fig. 2:*
- a. *Fig. 2g: The authors should add “CCL5” in green to the images themselves for a clearer and more intuitive figure understanding.*
 - b. *Fig. 2l legends: The authors should add statistics to both of the groups. Moreover, the authors did not respond/relate to my previous comment regarding this panel and the fluctuations in CCL5 concentrations in the sham group.*
 - c. *In Fig.2o, it seems that the gating is on CD45RC+, meaning naïve CD8 T cells (Ordonez L. et al, PloS ONE, 2009). Opposite to what the authors are stating in the text and showing in Fig.2P.*
 - i. *The x and y axes' fluorescence intensity units are small and not clear.*
 - ii. *The authors should add percentages in the plot like they added in similar other plots.*
 - d. *Not clear why the authors sometimes add statistical details and sometimes don't. There should be more consistency throughout the text, for example, in Legends of Fig.2.*

RESPONSE: We thank the reviewer for these suggestions. And we have revised Fig.2 according to the suggestions from the reviewer.

- a. We have taken out the CCL5 imaging data in Fig. 2g according to the comments of reviewer 2.
- b. We have added statistics to both groups “two-way ANOVA, interaction $F_{4,20} = 4.44$, $P = 0.0099$, effect of time $F_{4,20} = 21.02$, $P < 0.0001$, effect of lymphadenectomy $F_{1,20} = 191.9$, $P < 0.0001$ ” (**Line 1241-1242**). Restraint stress could lead to a sharp increase of CCL5 in rats, thus, CCL5 could be increased significantly after restraint stress in

sham group.

c. In Fig. 2o, “Flow cytometry analysis of CD8 memory cells in the CLNs of control and restraint stress-treated rats. CD8+CD45RC-: memory CD8 T cells; CD8+CD45RC+: Naïve CD8 T cells.”, which is consistent with the results of Fig. 2p.

We have updated Fig. 2o.

d. We have added the statistical details in legends of Fig.2. (**Line 1241-1251**)

7. Fig. 3:

a. *3b. the authors did not relate to my previous comment regarding this image about clarifying what the arrows signify. If it is pointing to the co-localized cells I would also suggest enlarging the image since it is hard to detect.*

RESPONSE: We apologize for the negligence in the manuscript. The arrows in Fig. 3b signify co-localized cells, and we enlarged the image. (**Line 1264**)

8. *Line 181: add a reference to Fig. S2.*

RESPONSE: The reference to Fig. S2 is in **line 148 & line 152**.

9. *Line 200, 202: “peripheral blood”, and not just “peripheral”.*

RESPONSE: Thank you, correction made. (**Line 205, 207**)

10. *In lines 226-7, the authors state that they measured concentrations of cytokines and chemokines, suggesting that they measured protein levels (with ELISA, for example), while they are actually referring to the RT-PCR data, measuring RNA levels in CLNs. It is confusing and inaccurate.*

RESPONSE: We thank the reviewer for this point, and have updated the statement: “Restraint stress also led to an increase in the mRNA concentrations of cytokines and chemokines, such as IFN α , IL2, IL6, and CCL2.” (**Line 230-232**)

11. Fig. S6:

a. *Fig. S6c: Anisomycin is still misspelled in the scheme.*
b. *Fig. Sj: The image of the EPM is supposed to show that the rats treated with Anisomycin spent more time in the open arms. Yet, if I understand correctly and the open arms are the vertical arms (also something worth mentioning in the figure- which arms are which), it shows the exact opposite trend between the groups.*

RESPONSE: We apologize for the mistake and have revised it in the Fig. S6. (**Line**

1415)

12. *Line 300: The new section's title refers to the inhibition of RN, yet this section also includes a description of the effects of RN activation. Either separate it into two sections (inhibition and activation) or generalize the title, something like "Chemogenetic manipulation of RN glutamatergic neurons affects..."*

RESPONSE: We thank the reviewer for this point, and we have revised the title as "Chemogenetic manipulation of RN glutamatergic neurons affects anxiety-like behavior and CCL5 secretion". (Line 302-303)

13. *Line 347: As in the abstract, the title of this section is confusing; it suggests M1-derived RN neurons trigger anxiety-like behavior, while in fact, as the authors show, inhibition of the M1-RN circuit induces anxiety-like behavior. Similar to line 504, in the discussion section.*

RESPONSE: We thank the reviewer for this point, and we have updated the title as: Inhibition of the M1-RN circuit induces anxiety-like behavior. (Line 366) In the discussion section, the statement has been revised as: our results causally demonstrate a new pathway from the M1 directly to the RN for regulating anxiety-like behavior. (Line 508)

14. *Line 351: Add fig. reference.*

RESPONSE: We thank the reviewer for this point, and we have added the fig. reference. (Line 369)

15. *Line 370: There is no Fig. 6l as mentioned in the text.*

RESPONSE: We apologize for the mistake and have revised it. (Line 388)

16. *Line 375: DREADD, not DREDD*

RESPONSE: Thank you, correction made. (Line 393)

17. *Discussion:*

- a. *Throughout this section, the authors rightfully insert statements that are based either on previous studies or the current study. However, it is not always clear when they are referring to which. One example of this is in line 500; after several referenced data, they go back to their own study but do not mention it; in such a case, I would expect them to add "Here, we showed..." / "In this study, we demonstrated..." and so on.*

RESPONSE: We thank the reviewer for this point, and we have revised it. (**Line 430, 436, 507,**)

b. Line 467: A reference is needed for this sentence.

RESPONSE: We have added the reference.

c. Line 468-9: This sounds like a very significant conclusion, although it is not entirely clear what the authors are basing this statement on.

RESPONSE: We thank the reviewer for this point, and we have revised the statement. “In this study, we demonstrated that inhibiting the translational pathways using anisomycin could reduce the stress-induced anxious behavior. Translational regulation of mRNA shows promise as a safe and specific treatment to combat stress-induced anxiety disorders.” (**Line 476-479**)

d. Line 472: “electrical” recordings?

RESPONSE: Electrical recordings including in vivo electrophysiological techniques, Fiber optic recording, Optogenetic techniques and so on. (**Line 482**)

e. Line 479: Add a reference to this statement about the prefrontal-thalamic circuit.

RESPONSE: We have added the reference. (**Line 488**)

f. The mentioning of CLNs’ NA data in the discussion seems quite misplaced. First, it seems more relevant to the result section; second, it is not clear from the text what is known from the literature and what the authors examined in the current study; third, if the authors mention that they checked NA levels in the text I would expect it to be presented, possibly in the supplementary figures (and respectively, how they measured it in the methods section).

RESPONSE: We thank the reviewer for this suggestion. And we have reorganized the CLNs’ NA data. According to the suggestions of the reviewer, we put the CLNs’ NA data to the results section (**Line 349-364; Fig. S11, Line 1475-1483**) and methods section (**Line 742-744**).

g. Overall, it feels like a very long discussion with too many repetitions on the study’s main conclusions. I would suggest focusing on the interpretation of the authors’ main findings (and, as already inserted, limitations of the study) in the context of the scientific literature regarding stress, the relevant brain neuronal circuits, peripheral immunity, and their interconnections.

RESPONSE: We thank the reviewer for this comment. And we have reorganized the discussion section. According to the reviewer’s suggestions, we first present the stress in the second paragraph, and then peripheral immunity, the relevant brain neuronal circuits and their interconnections. (**Line 429-520**)

18. *Methods section: The authors are not explaining how they extracted/sorted the T cells before intracellular staining. Did they stain all T cells or specifically CD8 T cells?*

RESPONSE: We thank the reviewer for this comment and add the information in methods section.

“CD8+T cells were isolated from the lympho-nodes (LNs) of adult rats using anti-CD8 magnetic beads (Miltenyi Biotec).” (Line 751-752)

19. *Methods section: c-Fos staining is not mentioned or described.*

RESPONSE: We thank the reviewer for this comment. And we have removed the results of c-Fos results.

Reviewers' Comments:

Reviewer #4:

Remarks to the Author:

The authors adequately replied to my concerns.

Regarding the cFos data, I think that removing it from the manuscript is a good solution, preventing complex explanations and presumptions of an issue that should probably be tackled elsewhere.

Only one remaining comment; the data regarding CD45RC+CD8+ T cells is still confusing (Fig. 2 n-o), as there was no change in the gating, the graphs, or the description of naïve/memory CD8 T cells in the text (lines 220-221), as the authors stated in their response. The gating in Fig. 2n still indicates CD45RC+CD8, while Fig. 2o and the text refer to CD45RC-CD8 memory T cells. The authors should correct this issue.

Dear reviewer 4,

We hereby submit a revised manuscript, “Stress-Induced Red Nucleus Attenuation Induces Anxiety-Like Behavior and Lymph Node CCL5 Secretion” (ID: NCOMMS-22-45871B). We appreciate the time and effort that you dedicated to providing feedback on our manuscript.

Please find our response to the comments of reviewer 4 below (reviewer’s comments are shown in italics).

Point-by-point responses to the reviewers’ comments:

Reviewer #4

The authors adequately replied to my concerns.

Regarding the cFos data, I think that removing it from the manuscript is a good solution, preventing complex explanations and presumptions of an issue that should probably be tackled elsewhere.

Only one remaining comment; the data regarding CD45RC+CD8+ T cells is still confusing (Fig. 2 n-o), as there was no change in the gating, the graphs, or the description of naïve/memory CD8 T cells in the text (lines 220-221), as the authors stated in their response. The gating in Fig. 2n still indicates CD45RC+CD8, while Fig. 2o and the text refer to CD45RC-CD8 memory T cells. The authors should correct this issue.

RESPONSE: We thank the reviewer for his or her comments about our work. We have updated the gating in Fig. 2n.